# Rapid culture-free diagnosis of clinical pathogens via integrated microfluidic-Raman micro-spectroscopy

Yuetao Li [1,2,8], Jiabao Xu [1,3,8], Xiaofei Yi[4,5], Xiaobo Li[1], Yanjun Luo[4], Andrew Glidle[1], Phil Summersgill[2], Simon Allen[2], Tim Ryan[2], Xiaochen Liu[6], Wei Yu[7], Xiaobing Chu[7], Shiyu Chen[7], Qian Zhang[7], Xiaogang Xu [5], Xiaoting Hua [6], Qiwen Yang[7], Julien Reboud [1], Yunsong Yu [6] ✉, Wei E. Huang [3] ✉, Jonathan M. Cooper [1] ✉ & Huabing Yin [1] ✉

Antimicrobial resistance (AMR) is a critical global health challenge, demanding rapid and accurate diagnostics to guide timely antimicrobial therapy. Current diagnosis is hindered by prolonged culturing and difficulties detecting low pathogen loads. Here, we present a culture-free diagnostic platform that integrates microfluidics, Raman micro-spectroscopy, and deep learning to deliver "sample-to-report" testing within 20 min. The microfluidic enrichment system employs dialysis-dielectrophoresis (DEP) technology to rapidly isolate pathogens directly from clinical samples with a detection limit as low as <2 colony forming unit (CFU)/ml. Combining a single-cell Raman fingerprint database of 342 clinical isolates from 29 bacterial and 7 fungal species with a 1D ResNet deep learning model, our approach achieved 95.1% accuracy in lab settings. Validated in a 305-patient clinical study involving primary urine and other clinical samples, it demonstrated 95.4% agreement with traditional culture methods and 98.5% sensitivity in diagnosing infections. While broader validation is needed for clinical implementation, the integrated, rapid diagnosis pipeline, as well as broad-spectrum detection, offer a promising solution for next-generation diagnostics for combating AMR.

Antimicrobial resistance (AMR) has become one of the top threats to human health. Globally, deaths due to AMR are expected to rise from ~5 million in 2019[1] to ~10 million by 2050 and the GDP loss is estimated to be around $100 trillion by 2050[2]. While the development of AMR is inevitable due to natural selection, its rapid acceleration is primarily driven by the overuse and incorrect prescription of antibiotics. A significant contribution to this crisis is the lack of rapid diagnostic tools, which slows timely identification of microbial infections. Current diagnostic standards, require at least two days to provide results due to lengthy sample culture (> 24 hours) and the need for pure isolates for pathogen identification (e.g., MALDI-TOF mass spectrometry)[3] and further sub-culturing for antibiotic susceptibility testing[4]. Thus, broad spectrum antibiotics are empirically administered to patients in the first instance, which could lead to poor patient outcomes, and foster the development of antibiotic resistance[5].

[1]James Watt School of Engineering, University of Glasgow, Glasgow, UK. [2]Epigem Ltd, Redcar, UK. [3]Department of Engineering Science, University of Oxford, OX1 3PJ, Oxford, UK. [4]Shanghai D-band Medical Technology Co., LTD, Shanghai, P. R. China. [5]Huashan Hospital, Fudan University, Shanghai, P. R. China. [6]Department of Infectious Diseases, Sir Run Run Shaw Hospital, School of Medicine, Zhejiang University, Hangzhou, China. [7]Department of clinical laboratory, Peking Union Medical College Hospital, Peking Union Medical College, Beijing, P. R. China. [8]These authors contributed equally: Yuetao Li, Jiabao Xu. ✉e-mail: yvys119@zju.edu.cn; wei.huang@eng.ox.ac.uk; Jon.Cooper@glasgow.ac.uk; huabing.yin@glasgow.ac.uk

Diagnostic time is crucial for combating AMR infections, especially in life-threatening conditions like sepsis and pneumonia. To shorten turnaround time for diagnosing infections, molecular-based technologies such as polymerase chain reaction (PCR) and whole-genome sequencing (WGS) have emerged[6–10]. While these methods eliminate the need for culturing, the prerequisite of extracting and enriching the genetic materials from the sample adds substantial cost and complexity[11]. Additionally, nucleic acid-based detection is limited to identifying known species using specific genetic markers[6,7]. WGS provides a more universal approach to detect pathogens without prior knowledge[10], but still requires high DNA copy numbers and thus, in practice, still necessitates prior culture and so delays the time-to-result, placing it on par with conventional gold-standard techniques. Metagenomics, although capable of working with uncultured samples, requires high pathogen loads, and is mainly applicable to limited sample types (e.g., non-sterile sputum, faeces)[8,9].

Rapid, untargeted diagnosis of early-stage infections and life-threatening conditions, particularly those with low pathogen loads, remains an unmet clinical challenge. For critical conditions such as sepsis and pneumonia, patients have extremely low bacterial loads (i.e., between 1 and 100 colony-forming units (CFU) per ml) in either bloodstream or pulmonary effusion[12]. Each hour of delay in initiating an appropriate antimicrobial therapy has been associated with a 7.6% decrease in survival for a septic patient[13]. Current technologies fall short of detecting such low pathogen levels[8–10,14,15]

underscoring the urgent need for rapid, culture-free diagnostic solutions.

Raman spectroscopy is a label-free vibrational spectroscopy technique that provides the intrinsic biochemical profiles of pathogens at the single-cell level[16–18]. Each single-cell Raman spectrum (SCRS) provides a unique "Raman phenotype," representing the vibrational frequencies of cellular biomolecules. While recent studies have demonstrated the potential of Raman spectra to identify pathogens by genus, species, and even strain level[19–22], these studies have focussed on a narrow range of pathogens presented as isolates or spiked clinical samples, with limited demonstration in real clinical conditions. In addition, most approaches rely on extensive manual sample processing, including multiple centrifugation and washing and/or filtration procedures[22–26]. Even global collaborative initiatives such as 'MicroBio-Raman', which have illustrated the increasing trend towards large-scale, systematic, and open-access databases for microbial Raman fingerprints, aiming to standardise data collection and improve accessibility[27], do not address the critical need for rapid diagnostics going from patient sample to a clinical decision, as an integrated workflow.

Here, we develop a transformative culture-free approach for rapid "sample-to-result" diagnosis of microbial infections within 20 min (Fig. 1, Supplementary Fig. S1). This platform combines advanced microfluidics technology, the largest database of Raman fingerprints of pathogens, and deep learning algorithms for rapid clinical

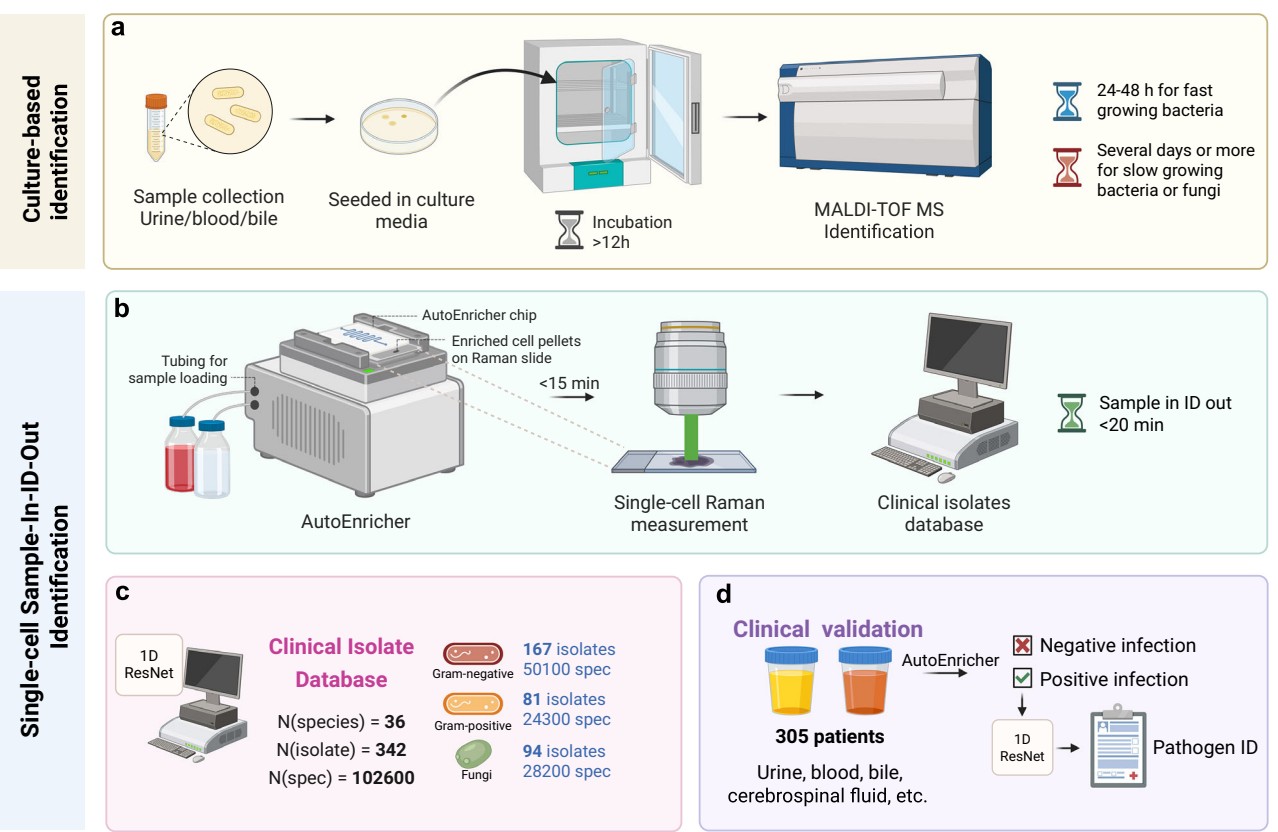

**Fig. 1 | Comparison of conventional culture-based ID and culture-free sample-to-result ID methods. a** Conventional culture-based approach involves sample collection, seeding in culture media, incubation (24-48 hours for fast-growing bacteria, or several days for slow growers), and final identification using MALDI-TOF MS. **b** The rapid sample-to-result ID method isolates pathogens directly from the clinical sample using the microfluidic enrichment system (denoted as Auto-Enricher), followed by single-cell Raman spectroscopic measurement to identify pathogens within 20 min (complete workflow and timeline in Supplementary Fig. S1). **c** A clinical isolate database containing 36 species and 342 clinical isolates

across Gram-negative (167 isolates, 50,100 spectra), Gram-positive (81 isolates, 24,300 spectra), and fungi (94 isolates, 28,200 spectra) was used to train a ResNet convolutional neural network model. **d** Clinical validation using the enrichment system and single-cell Raman identification method. The study involves 305 patients with various sample types (urine, blood, bile, cerebrospinal fluid, etc.). Samples were processed and classified by AutoEnricher as positive or negative for infection, and isolated pathogens were identified using the 1D ResNet algorithm. Created in BioRender. Xu, J. (https://BioRender.com/dmb6g5f).

diagnostics. The innovative dialysis-dielectrophoresis (DEP) enrichment system, denoted as the AutoEnricher[28], effectively isolates and enriches pathogens directly from clinical samples with microbial loads ranging from <2 CFU/ml to >10⁵ CFU/ml) within minutes. This study marks the most comprehensive application of single-cell Raman spectroscopy for clinical pathogen identification, supported by a vast database of 342 clinical isolates, including 36 bacterial and fungal species commonly found in hospital infections. Using an advanced deep learning model (1D ResNet), our platform achieves 95.1% species-level accuracy in laboratory settings. More importantly, we extend our validation to a large-scale clinical study involving 305 patients, demonstrating real-world applicability and achieving 95.4% accuracy and 98.5% sensitivity in diagnosing infections directly from clinical samples. With rapid turnaround, sensitivity, and broad-spectrum capabilities, this platform offers a next-generation diagnostic tool with game-changing potential for rapid diagnosis of microbial infections, addressing a vital challenge in life-threatening AMR.

## Results

### Engineering the microfluidic enrichment system

Rapid phenotypic diagnosis of microbial infections relies on efficiently obtaining potential pathogens from clinical samples, where pathogen species are usually unknown. Ideal diagnostic methods must be broad-spectrum, capable of detecting a wide range of species without prior knowledge. Current methods, however, often depend on affinity binding of biomolecules to specific pathogens (e.g., using magnetic beads, filtration)[5,29,30], and thus limit the detection range[31]. Emerging microfluidic technologies provide label-free alternatives to isolate bacteria using physical forces or barriers[32,33]. Among these, positive dielectrophoresis (pDEP) has demonstrated promising capability of capturing bacteria in flow[34,35]. However, pDEP requires low media conductivity, whereas the conductivity of body fluids ranges from 0.1 S/m to 3.4 S/m[36]. Traditional sample desalting techniques using macroscopic pre-washing[37], often result in significant pathogen loss, and existing on-chip approaches using ion-exchange membrane[38] or H-filters[35] suffer from low desalting efficiency and low process throughput (<0.6 mL/h). These problems prevent those methods from finding clinical applicability. Clinical settings demand processing of millilitre-scale samples containing various biomolecules and cells, not just pathogens.

To overcome these challenges, we developed an automated, integrated dialysis-DEP microfluidic system (Fig. 2), offering high efficiency and high throughput in isolating pathogens, making it well-suited for clinical applications. The core of our enrichment system is the seamless integration of a membrane dialysis unit and a DEP microfluidic device, as detailed in the top and side views (Fig. 2a and b). The sample first undergoes desalting as it passes through the dialysis unit, where the electrolytes diffuse through a porous membrane and are removed by counter-flowing water (Fig. 2a). The initial high conductivity of the sample is significantly reduced as electrolytes are removed, preparing the sample for efficient pathogen capture (Fig. 2b). The desalted sample then enters the DEP chip, which consists of a chevron electrode array and an interdigital transducer (IDT) array. The chevron electrode array focuses pathogens towards the centre of the bottom channel by pDEP forces to facilitate cells being captured in the trap/release area on the IDT array (Fig. 2a). Once the pathogens are captured, the alternating current (AC) voltage is turned off, allowing the trapped cells to be released into the collection window for in situ analysis or off-chip retrieval for downstream processes (Fig. 2b).

To enable sample-to-report operation in clinical settings, a user-friendly instrument was developed (Fig. 2c), which consists of compact benchtop housing pumps, circuit boards, and a customised base that houses the disposable dialysis device and DEP chips. The entire process is automated via control software, enabling streamlined

operation. While the DEP chip selectively captures pathogens (<15 min), other components in the sample (e.g., blood cells) are diverted to the waste outlet, yielding purified and enriched pathogens ready for Raman measurements (<5 min) (Supplementary Fig. S1). This advanced system facilitates rapid pathogen identification, achieving a "sample-to-report" turnaround within ~20 min, thus significantly enhancing the speed and accuracy of clinical diagnostics.

### System characterisation

For rapid processing of large sample volumes, we designed a DEP chip with a large cross-section (1 mm width and 30 µm height) to accommodate high flow rates. To reduce random adhesion of cells onto such a large surface area, the chevron electrode array directs pathogens towards the centre of the bottom channel, where they are trapped on the downstream IDT array. Simulation indicated that a solution conductivity below 10 mS/m is required for effective pDEP (Fig. 3a). Applying an AC signal generates a non-uniform electrical field and corresponding pDEP force (Fig. 3b) that captures pathogen at the electrode edges where the electric field strength is highest.

We tested the desalting efficiencies for a range of sample flow rates and found a relatively high water flow rate can effectively reduce the sample' conductivity below 10 mS/m (Fig. 3c). Even for a high conductivity sample (e.g., LB broth with conductivity of 2.25 S/m) at 1 mL/min, a 2 mL/min water flow can effectively reduce its conductivity to 1.3 ± 0.12 mS/m instantly after the sample flow passed through the dialysis device (Fig. 3c). With this capability, our system can process several millilitres of samples within tens of minutes. Using a 10⁷ CFU/ml E. coli (ATCC 25922) solution, no bacterial colonies were observed after culturing the water channel effluent, showing that no bacterial cells passed through the membrane during dialysis.

While high flow rates allow fast processing of samples, they also increase the hydrodynamic drag force, potentially impacting capture efficiency. To evaluate this, we tested capture efficiency at various flow rates using a 10⁷ CFU/ml E. coli solution. As shown in Fig. 3d, at the voltage of 20 V, 98.3 ± 4.01% of the bacteria can be captured at a flow rate of 10 µL/min. Increasing the voltage permits higher sample flow rate. At the voltage of 40 V (i.e., the allowed operational range for the current hardware), >95% capture efficiency has been achieved at higher flow rates, e.g., 400 µL/min (Supplementary Fig. S2).

### Validation of capture efficiency across major pathogen groups

To demonstrate the generalisability of our system's functionality beyond E. coli, we systematically evaluated capture efficiency across 9 additional species representing major pathogen groups commonly encountered in healthcare settings. These included three Klebsiella species (K. aerogenes, K. oxytoca, and K. pneumoniae), which are known to produce extracellular matrix and can be challenging to handle in microfluidics; three Gram-positive species (Enterococcus faecalis, E. faecium, and Staphylococcus capitis) and three clinically relevant Candida species (C. albicans, C. glabrata and C. parapsilosis), each with distinctive cell wall properties and morphologies (Supplementary Fig. S3).

Despite these differences, the enrichment system demonstrated excellent capture efficiencies across these species between 89.1% and 99.5% (Fig. 3e). Furthermore, time-lapse fluorescence imaging revealed an efficient trapping process across all species: all cells entering the field of view were captured and progressively accumulated at the electrode edges (Supplementary Fig. S4, Supplementary Movies 1–11). This robust performance demonstrates its capability for blinded testing of clinical samples containing unknown pathogen species.

### Capture extremely low bacterial loads

To further evaluate the enrichment system's capture efficiency with extremely low bacterial loads, RFP E. coli was used as a model strain to facilitate cell counting via fluorescence imaging. Using cell

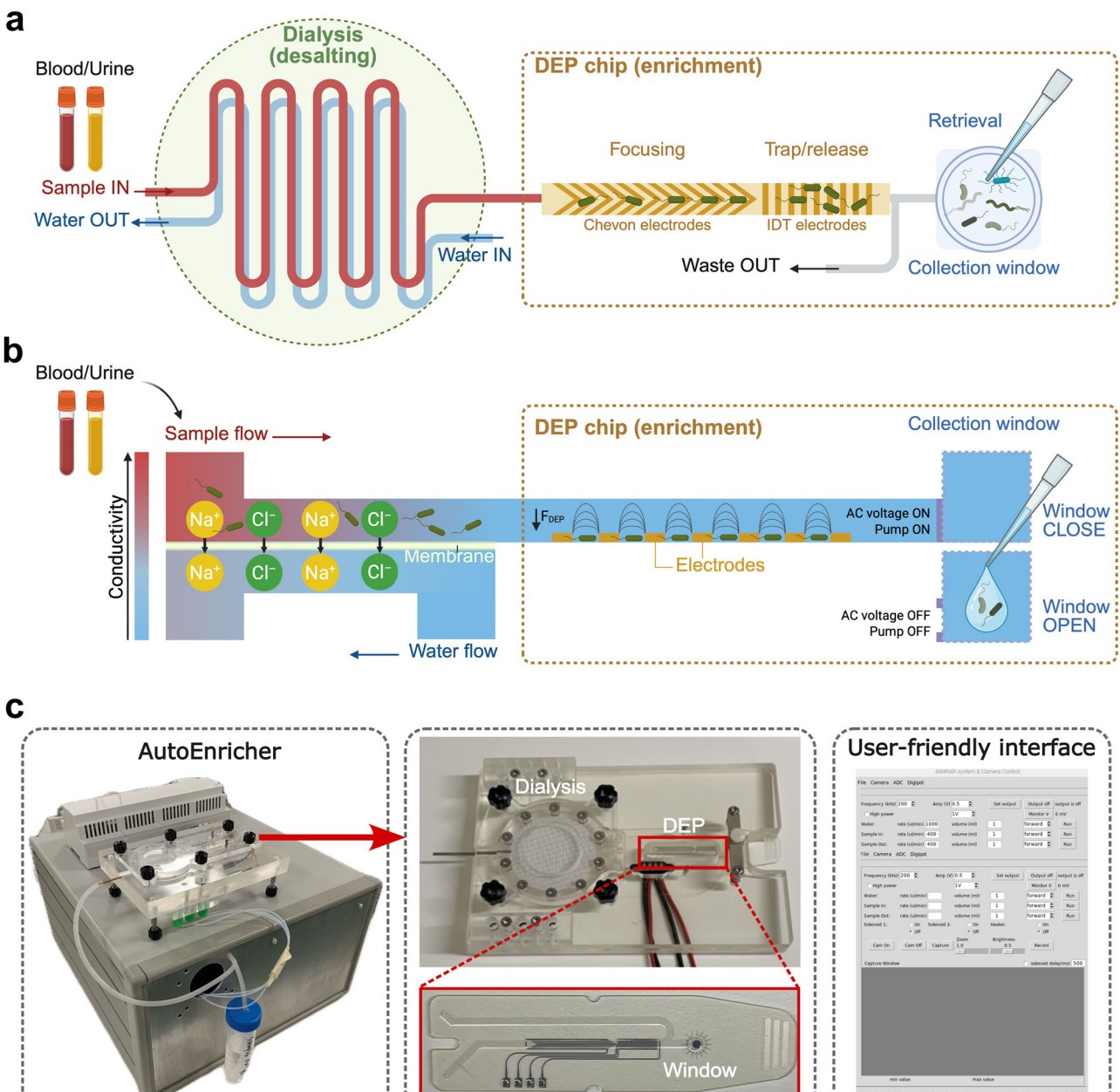

**Fig. 2 | The integrated dialysis-DEP enrichment system. a** Top view and (**b**) Side view of the system. The dialysis unit consists of two layers of microfluidic channels separated by a porous membrane. As the sample flow passes through the upper channel, electrolytes diffuse through the membrane and are removed by the counterflow of water below. This desalted sample flow then enters the DEP chip, where pathogens are focused in the centre of the channel and trapped on the IDT arrays due to positive dielectrophoresis (pDEP) forces. When the AC voltage is turned off and the pump is stopped, the captured cells are released into the collection window for in situ or off-chip analysis. Created in BioRender. Xu, J. (https://BioRender.com/038g1ru). **c** Photo of the enrichment system (AutoEnricher–a compact benchtop instrument that houses all components). The dialysis unit and DEP chip are disposable components housed in a customised plastic base. A user-friendly control software enables automated processing after sample loading.

concentration of ~1000 CFU/ml and tiled fluorescence images, we found that only 17.3 ± 7.8% of the cells were located at the chevron areas, but most of the captured cells (76.5 ± 11.0 %) were concentrated in the front IDT electrode pairs–the designated trapping area (Fig. 4a and Supplementary Fig. S5). Thus, time-lapse fluorescence imaging at this location was employed to monitor real-time trapping processes, providing a direct and rapid means to estimate the capture efficiency of samples with very low pathogen counts.

We first validated this approach using a 5 μm PMMA fluorescent beads solution at a concentration of ~1 bead/mL. As shown in Supplementary Fig. S6, the device consistently trapped one bead every 5 min, which matched very well with the number of the input beads (i.e., 1 bead per 5 min at 200 μl/min flow rate). Next, we tested a range of low bacterial concentrations (2, 4, 8, 23, 41 and 231 CFU/ml) using RFP *E. coli* (Figs. 4b, c). The captured cell numbers correlated linearly with the theoretical input cell numbers across all the concentrations, with an average gradient of 0.84 ± 0.06 (R² > 0.98). Extrapolating these results to the entire chip area indicates a capture efficiency over 84%, demonstrating the excellent capability of the system for capturing pathogens with concentrations as low as ~2 CFU/ml.

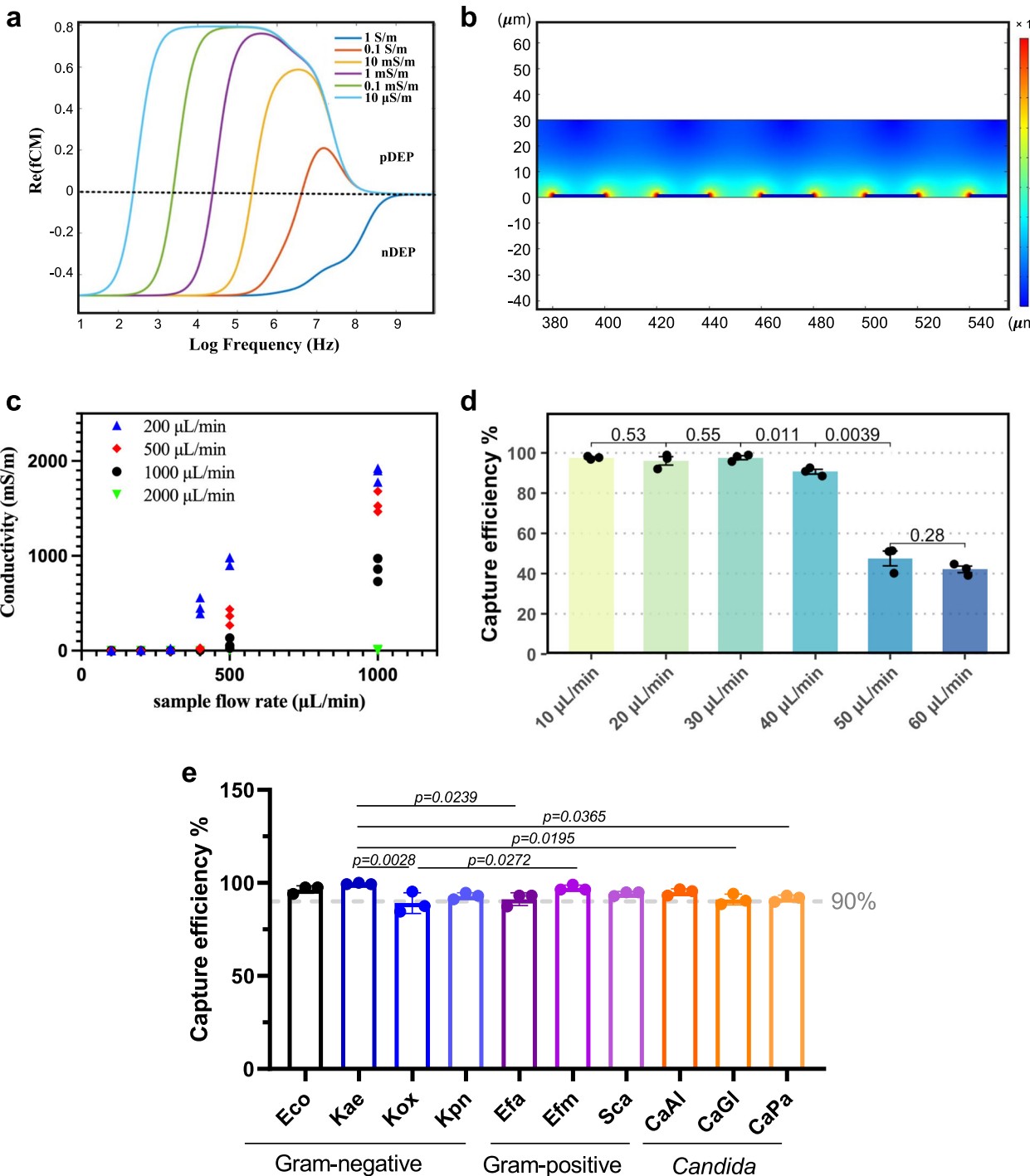

**Fig. 3 | Characterisation of DEP chip for pathogen enrichment and capture efficiency. a** Simulation showing the importance of low conductivity for positive dielectrophoresis (pDEP). **b** Simulation to optimise electrode geometry and channel height to achieve a high trapping force. **c** The relationship between sample and water flow rates for effective desalting of the sample ($n = 3$). **d** The effect of sample flow rates on capture efficiency. AC voltage was fixed at 20 V. **e** Comparative capture efficiency across pathogen groups. Capture efficiency measured for 10 different species representing Gram-negative bacteria, Gram-positive bacteria, and *Candida* species. Dashed line indicates 90% efficiency threshold. Statistical significance was determined by one-way ANOVA, $F_{(9, 20)} = 4.664$, $p = 0.0020$, $R^2 = 0.6773$, followed by Tukey's post-hoc test. Abbreviations: Eco, *E. coli*; Kae, *Klebsiella aerogenes*; Kox, *Klebsiella oxytoca*; Kpn, *Klebsiella pneumoniae*; Efa, *Enterococcus faecalis*; Efm, *Enterococcus faecium*; Sca, *Staphylococcus capitis*; CaAl, *Candida albicans*; CaGl, *Candida glabrata*; CaPa, *Candida parapsilosis*. **d**, **e** Data represent mean ± SD from three independent experiments. Source data for (**c**), (**d**) and (**e**) are provided as a Source Data file.

A distinct feature of the enrichment system is to release and deliver the captured cells directly to the on-chip opening window, which not only facilitates in situ analysis of cells but also enables sample retrieval for further off-chip processes. Using a spiked concentration of ~150 CFU/ml, we found that 95.6 ± 1.1 % of the captured RFP *E. coli* cells were successfully released (Fig. 4d) and delivered to the opening area with flow, forming a < 3 µl droplet. This high capture and release efficiency allows for rapid estimation of pathogen

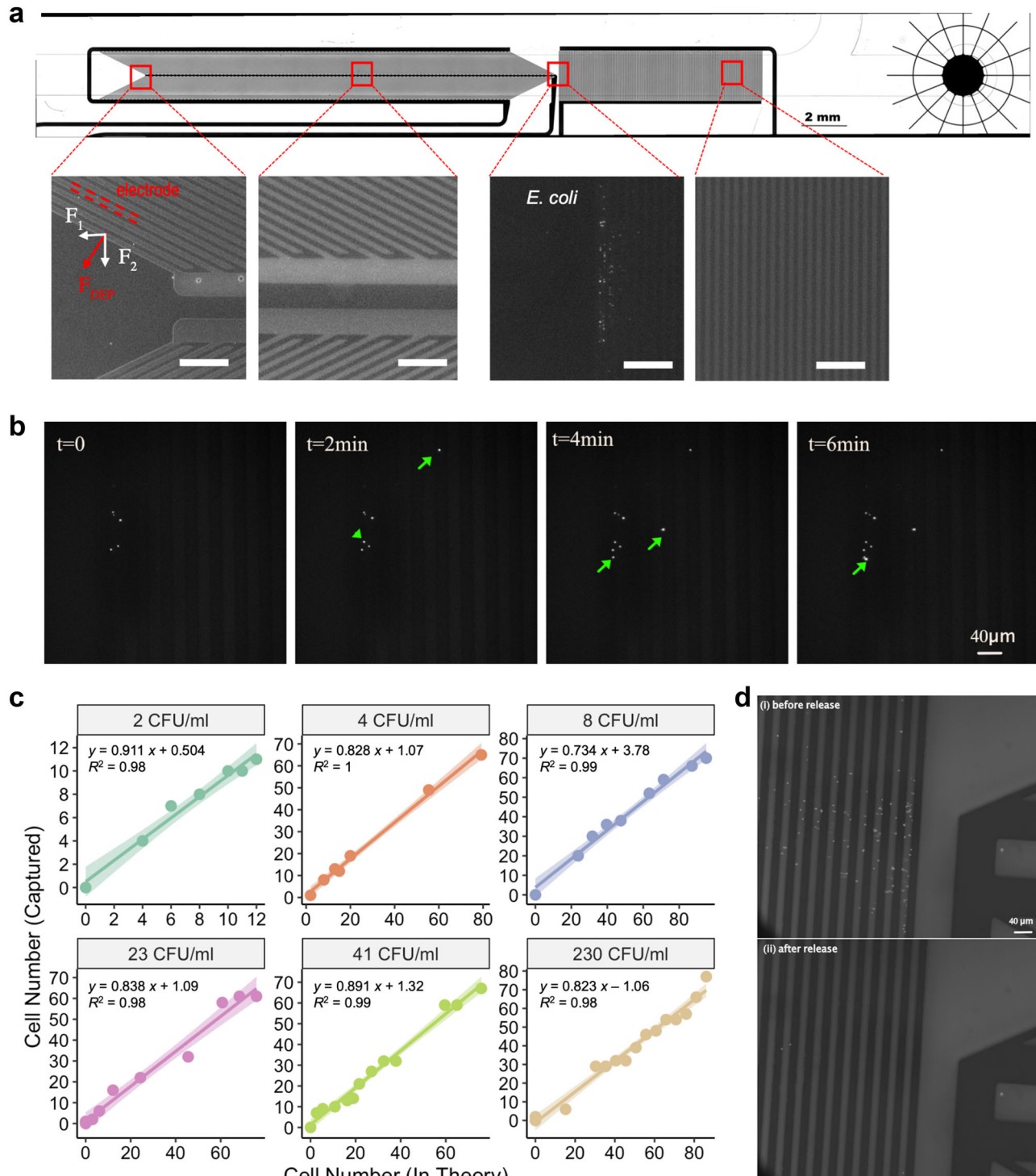

**Fig. 4 | Effective bacteria enrichment and release at low concentrations.**
**a** Location of trapped cells on the chip. Top row shows a bright-field image of the DEP device. Bottom row shows fluorescence images at representative locations along the electrode array, with a full scan available in Supplementary Fig. S5. Scale bar, 200 μm. **b** Representative time-lapse fluorescence images showing the enrichment of a low bacterial load sample ( ~ 5 CFU/ml in DI water). The sample flow rate was set at 200 μl/min, with a voltage of 40 V at 100 kHz. The input rate was calculated at ~1 cell per minute based on the sample volume delivered. Green arrows indicate newly captured fluorescent bacteria at each time point compared to the previous frame, demonstrating the progressive accumulation of pathogens on the electrode surface. **c** The captured cell number plotted against the theoretical input bacterial number for various low bacterial load concentrations (2, 4, 8, 23, 41, and 230 CFU/ml in DI water). Linear regression analysis indicates a strong correlation between the captured and theoretical cell numbers. Shaded areas represent 95% confidence intervals. The captured cell number was counted from the time-lapse images. **d** Fluorescence images showing the DEP chip before and after the release of trapped cells (i.e., small bright dots) upon turning off the DEP power, illustrating the effective release of captured bacteria. Data in (**a**–**d**) represent $n = 3$ independent experiments. Source data for (**c**) and (**d**) are provided as a Source Data file.

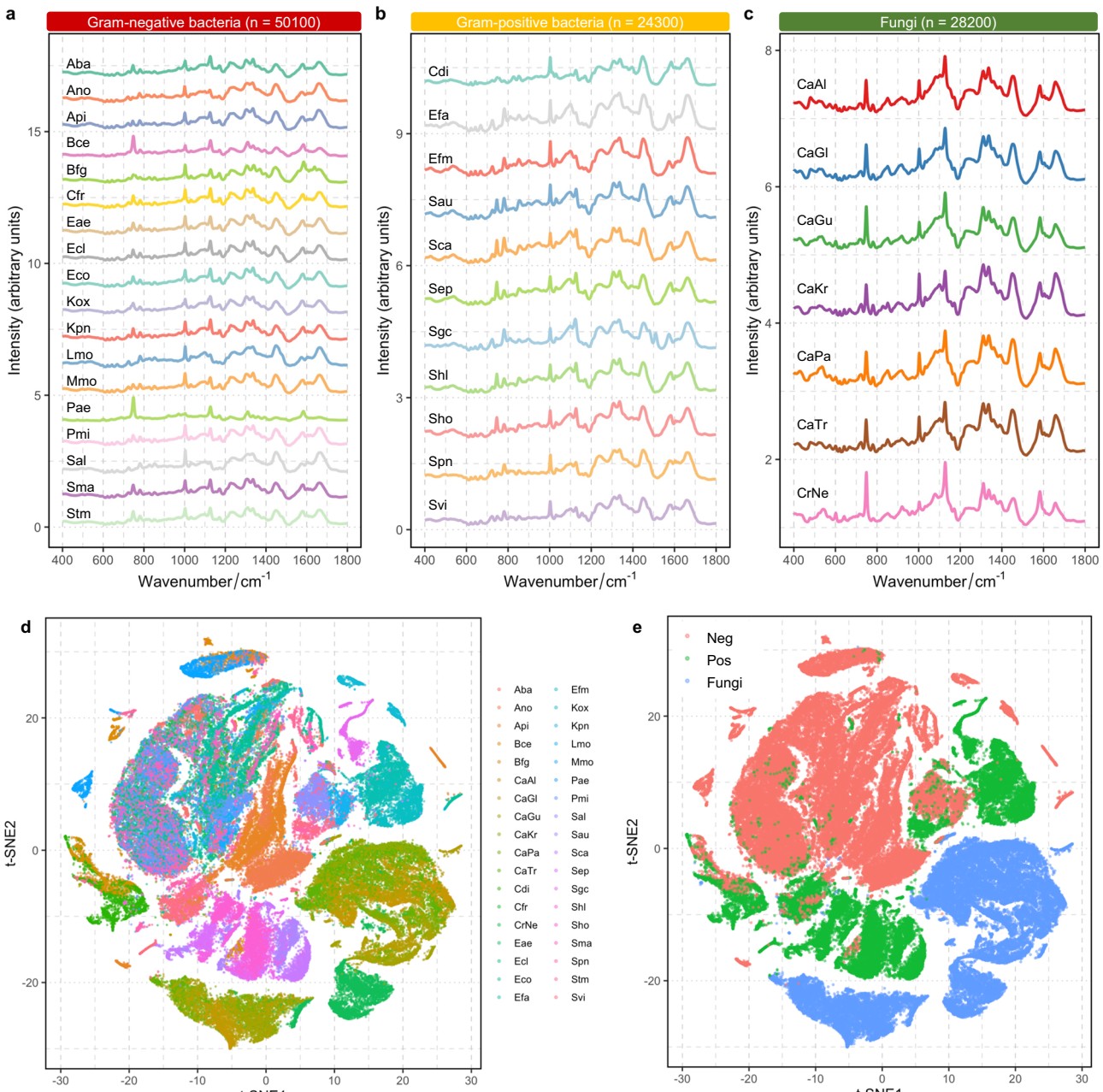

**Fig. 5 | Single-cell Raman database of 342 clinical isolates. a–c** Characteristic Raman spectra of clinical isolates from (**a**) Gram-negative bacteria (*n* = 50,100 spectra), **b** Gram-positive bacteria (*n* = 24,300 spectra), and (**c**) fungi (*n* = 28,200 spectra) used to train the ResNet model. Each coloured line represents the mean spectrum for one species. (**d-e**) t-SNE visualisation showing clustering of (**d**) different 36 species or (**e**) groups of Gram-negative bacteria (Neg), Gram-positive bacteria (Pos) or fungi. Abbreviations for the 36 species can be found in Supplementary Data 1.

load levels in the sample based on optical imaging of bacteria, allowing rapid diagnosis of negative/positive infection status. The captured cells in the droplet can be pipetted out for off-chip analysis (i.e., optical imaging or Raman-based identification). Further validation by agar plate culture verified that the enriched and released *E. coli* cells were viable, as further evidenced in the clinical trial section.

**Establish Raman identification (ID) database and classification model with clinical isolates**

Single-cell Raman spectroscopy has emerged as a non-invasive and rapid technology to classify microorganisms[17–20,37] and antimicrobial resistance[21,22,25,39–41]. For accurate identification of unknown pathogens in clinical samples, it is necessary to establish a Raman database over a wide range of commonly occurring pathogens. Thus, we constructed a single-cell identification database comprising 342 clinical isolates from 36 clinically important species (Fig. 1c; species abbreviations listed in Supplementary Data 1; a comprehensive list of all isolates in Supplementary Data 2). Each isolate includes 300 single-cell Raman spectra (SCRS), resulting in a total of 102,600 SCRS.

Fig. 5 illustrates the characteristic Raman spectral fingerprints and the clustering of clinical isolates within the database. Fig. 5a–c present the mean Raman spectra for 18 species of Gram-negative bacteria (50,100 spectra), 11 species of Gram-positive bacteria (24,300 spectra), and 7 species of fungi (28,200 spectra), respectively. Each line represents the mean spectrum for one species. Comparing bacterial and fungal spectra (Supplementary Fig. 7), fungi show higher intensity of

Raman spectral bands related to cytochrome $c$[42] at 750 cm$^{-1}$ (pyrrole ring breathing) and 1128 cm$^{-1}$ (C–N stretching) and ergosterol backbones[43] at 597 cm$^{-1}$; bacterial cells have higher intensities in bands related to nucleic acids[44] at 780 cm$^{-1}$ (cytosine/uracil ring breathing) and proteins[45] at 1240 (Amide III) and 1650 cm$^{-1}$ (Amide I). These results are consistent with our previous report on fungal identification[18]. This is likely because fungi contain mitochondria with large amounts of cytochrome c, while bacteria do not have mitochondria and only a small proportion of bacterial pathogens have cytochrome $c$ in their periplasmic membrane for respiration (where it is also in low concentration)[46]. Additionally, bacteria replicate faster than fungi, leading to quicker DNA synthesis, which facilitates more rapid transcription activities and protein synthesis[47], explaining the increases in band intensities related to nucleic acids and proteins. Interestingly, clear spectral differences can also be observed between Gram-positive and Gram-negative bacteria (Supplementary Fig. 8), primarily due to the unique cell wall components of each group: Gram-positive bacteria have a thicker peptidoglycan layer, while Gram-negative bacteria possess lipopolysaccharides in their outer membrane.

To visualise the high-dimensional Raman dataset, we used unsupervised t-distributed stochastic neighbour embedding (t-SNE) to project the data into two-dimensional space. Each dot represents an individual spectrum, coloured by different microbial species (Fig. 5d) or groups of Gram-negative, Gram-positive bacteria and fungi (Fig. 5e). Fig. 5d reveals distinct clusters for different species, indicating intrinsic spectral differences between pathogenic species. Multiple clusters within the same species suggest intra-species diversity, potentially due to strain variations or differences in the two measuring instruments. Fig. 5e shows distinct separation between bacterial and fungal spectra, with some overlap between Gram-positive and Gram-negative bacteria, highlighting intrinsic differences between bacteria and fungi.

Among all isolates, 12 *E. coli* (Eco) isolates were collected with their antibiotic resistance profiles characterised, including MEM-R (resistance to meropenem), ESBL+ (extended-spectrum beta-lactamase positive) and ESBL–(extended-spectrum beta-lactamase negative), each with four isolates (Supplementary Data 3; Supplementary Fig. 9a). A supervised linear discriminant analysis clearly shows the separation of *E. coli* cells with different resistance characteristics (Supplementary Fig. 9b), supporting the evaluation of using SCRS in distinguishing resistant and susceptible strains.

## ResNet model training with isolate database

Next, we trained a one-dimensional ResNet model using the constructed database of 342 clinical isolates spanning 36 bacterial and fungal species. The details of the ResNet model architecture can be found in Methods. Training and validation were conducted using stratified group five-fold cross-validation to maintain distribution consistency of the 36 classes and ensure spectra from the same isolate did not appear in both training and validation sets. Hyperparameters were fine-tuned through validation set experimentation. We utilised Monte Carlo simulation with majority voting for validation datasets across all five folds. For each fold, 100 trials involved 10 random selections from the validation set, and majority voting determined the final predicted class for each isolate.

The confusion matrix for the 36 species classification results for the 342 isolates (Fig. 6a), highlights the accuracy of predictions. Each cell contains the percentage of true class/predicted class pairs. The model achieved an overall average accuracy of 95.1% at the species level. 27 out of the 36 species were predicted with 100% accuracy, indicating the model's robust capability to distinguish these pathogens based on their Raman spectral fingerprints. Fig. 6b shows the aggregated confusion matrix for three groups: Gram-negative bacteria, Gram-positive bacteria, and fungi. The model achieved 99% accuracy for Gram-negative bacterial isolates, 100% accuracy for Gram-positive

bacterial isolates, and 100% accuracy for fungi isolates, further demonstrating its strong performance across different pathogen types.

To further validate against biological variance, we conducted Leave-One-Replicate-Out cross-validation using 9 representative species across major pathogen groups. Independent biological replicates achieved 100% accuracy at the isolate level using majority voting and 82.7% mean single-cell accuracy (Supplementary Fig. S10). Bootstrap analysis revealed that reliable clinical identification requires minimal spectra: 5 spectra achieved 96.6% accuracy, while 10 spectra achieved 98.4% accuracy (Supplementary Fig. S11). This enables rapid clinical testing within our 20-min workflow (Supplementary Fig. S1), requiring only 4–5 min for Raman measurements of 10-15 cells.

## Cross-instrument validation on the ResNet approach

To address the challenge of cross-instrument transferability in Raman spectroscopy, we conducted systematic validation using a dual-instrument approach. We evaluated cross-instrument performance using a representative subset of 6 *Candida* species (81 isolates, 24,300 spectra) measured on both HORIBA and WITec Raman instruments.

Single-machine training (training on one instrument, testing on another) showed limited transferability, with HORIBA to WITec achieving 31.8% spectrum-level and 44.4% sample-level accuracy, while WITec to HORIBA showed 8.1% spectrum-level and 4.4% sample-level accuracy (Supplementary Fig. S12). However, our implemented mixed-machine training approach demonstrated robust cross-instrument performance, achieving 84.2% spectrum-level and 90.9% sample-level accuracy for HORIBA spectra, and 89.4% spectrum-level and 100% sample-level accuracy for WITec spectra when leaving one species out for independent validation.

## Adapting ResNet model for clinical samples

To advance towards directly identifying pathogens in complex clinical samples which have high variability that is largely absent in purified cultures, we introduced a fine-tuning dataset as a crucial link, expanding the model's applicability from controlled isolate conditions to the nuanced challenges of clinical settings. Prior to a Larger-Scale clinical study, we processed urine samples from 40 patients, each confirmed by hospital culture-based methods to have single-agent infections (Supplementary Data 4). By capturing Raman spectra from these 40 single-infection samples, we generated a fine-tuning dataset that connects the controlled isolate database with the heterogeneity of real-world clinical samples. To adapt the ResNet model for this diverse dataset, we applied fine-tuning on the pre-trained model with a lower learning rate of 0.00001. This adjustment was critical to maintaining model stability and preventing overfitting on the smaller, more complex fine-tuning dataset. The refined model was then subjected to a rigorous 5-fold stratified grouped cross-validation, mirroring the initial model training process while addressing the unique challenges of clinical data, specifically, the limited spectra per sample (30–50 single-cell spectra). These refinements have prepared the ResNet model for the upcoming clinical study, where the efficacy of the fine-tuned model in identifying pathogens directly from complex patient samples will be evaluated in practical diagnostic contexts.

## Clinical validation of rapid "sample-to-result" diagnosis of 305 patients

Following the development of the culture-free, rapid enrichment system and the establishment and finetuning of a ResNet ID model, we conducted a clinical study involving 305 patients from two hospitals. Various types of clinical samples including urine, bile, cerebrospinal fluid (CSF), drainage fluid, and puncture fluid, were collected (Fig. 1d; Supplementary Data 5). An additional 15 blood culture samples were collected to demonstrate the performance of the enrichment system

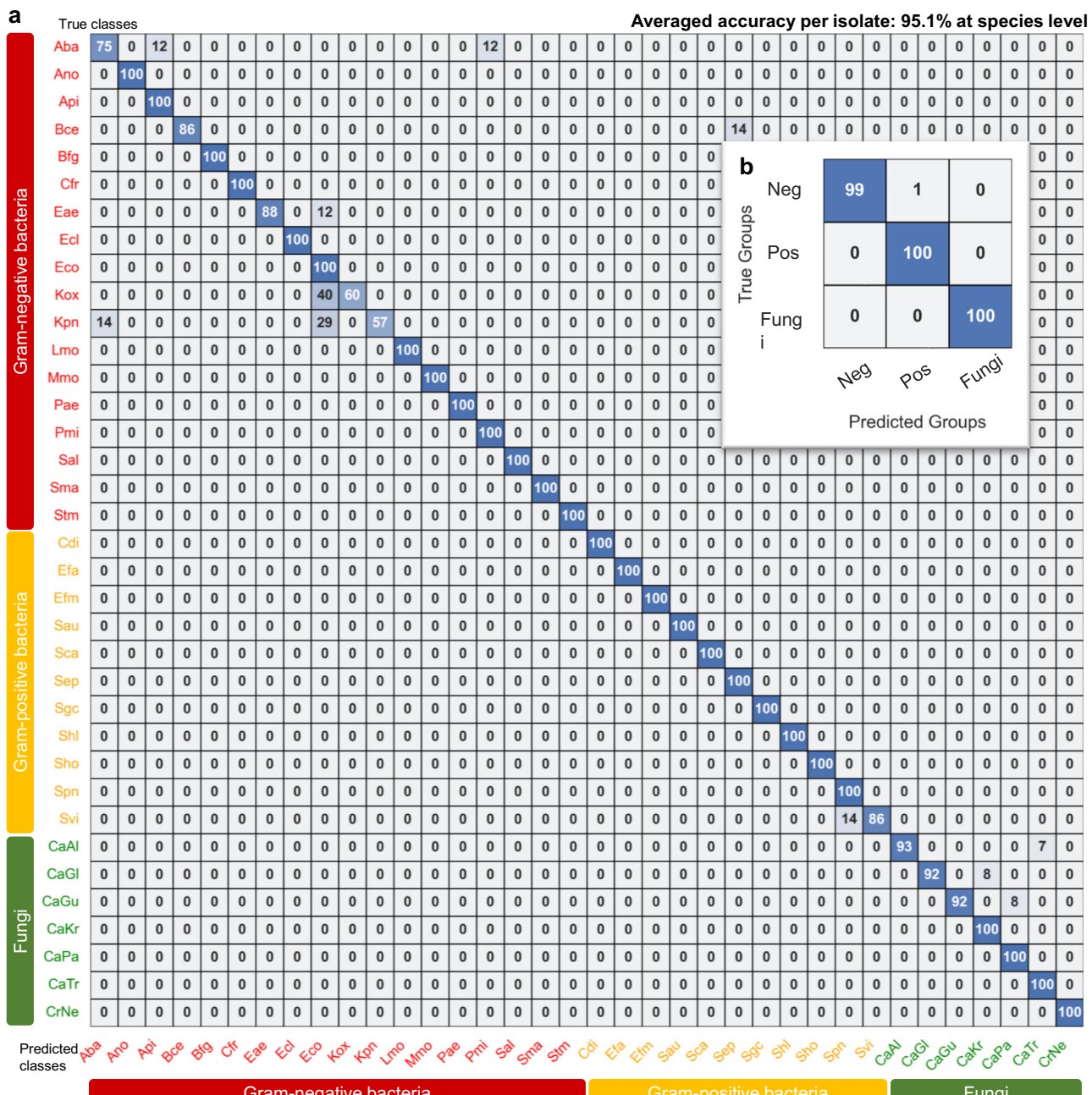

**Fig. 6 | Confusion matrix showing the classification performance of the ResNet model on 342 clinical isolates spanning. a** 36 bacterial and fungal species and (**b**) aggregated into three groups: Gram-negative (Neg), Gram-positive (Pos), and fungi. Each cell contains the percentages for that true class/predicted class pair. The average accuracy per isolate at the species level is 95.1%. Abbreviations for the 36 species can be found in Supplementary Data 1. Source data are provided as a Source Data file.

in processing samples with highly complex matrices (Supplementary Fig. S13). However, these were excluded from the 305-patient cohort as the standard protocol required collection of blood directly into culture medium due to regulatory compliance and therefore did not strictly follow our culture-free workflow.

To diagnose positive or negative infection after pathogen enrichment, we established detection thresholds based on optical images using representative bacterial and fungal pathogens at known concentrations (Supplementary Fig. S14 and Supplementary Fig. S15). Quantitative analysis revealed concentration-dependent increases in pathogen density for all tested morphologies, with significant differences between consecutive concentration groups ($p < 0.01$ to $p < 0.0001$). We established unified minimum thresholds for visual

determination of positive infection status: 300 per mm² for bacteria and 200 per mm² for fungi, corresponding to the clinical threshold (>1000 pathogens/ml) for urine samples. For other normally sterile clinical specimens (CSF, blood, bile, and drainage fluids), any visible pathogen presence was classified as a positive infection.

Our culture-free approach demonstrated remarkable effectiveness for rapid clinical pathogen identification, achieving "sample-to-report" testing within 20 min (Figs. 1b and 7a). Out of 305 samples analysed, our enrichment system achieved a total accuracy of 95.4% (291 out of 305), with a sensitivity of 98.5% and a specificity of 83.6% (Supplementary Data 4; Fig. 7a).

Specifically, the model correctly classified 291 samples and misclassified 14 (4 false negatives and 10 false positives). The high

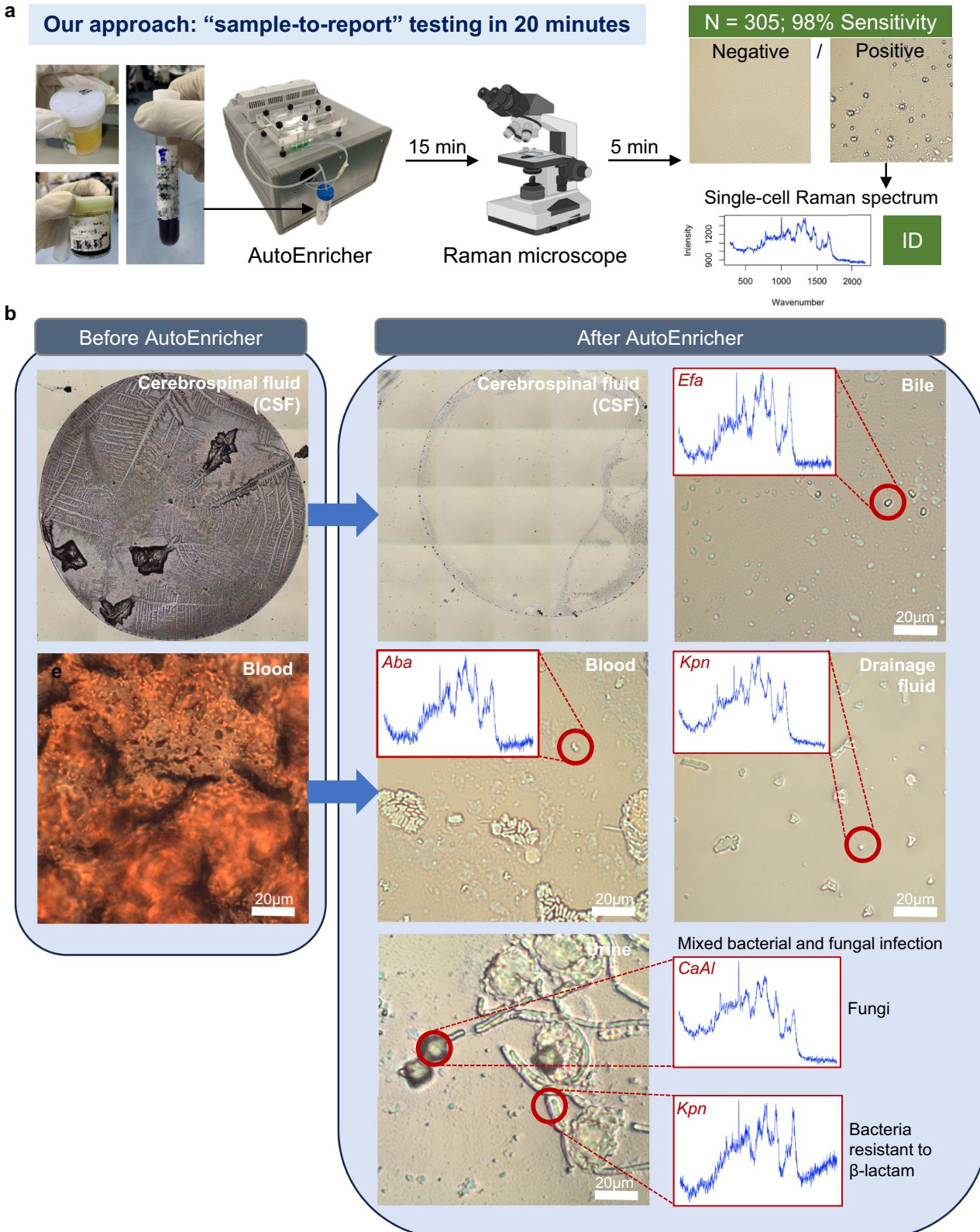

**Fig. 7 | Rapid clinical pathogen identification using the enrichment system and Raman microscopy. a** Our approach achieves "sample-to-report" testing within 20 min for clinical pathogen identification. This method was tested on 305 clinical samples (urine, bile, cerebrospinal fluid (CSF), drainage fluid) with photos of collected sample tubes shown: urine sample (top left); bile sample (bottom left); pancreatic drainage fluid (right). It demonstrated over 90% accuracy and 98% sensitivity in identifying negative and positive infections. Representative images show the difference in microscopic appearance between negative and positive samples, with corresponding single-cell Raman spectra used for pathogen identification. **b** Representative microscopic images from the 305 processed clinical samples, shown before and after processing with the AutoEnricher. Left panels show the original state of the samples, while the right panels illustrate the samples post-enrichment, highlighting the improved visibility and identification of pathogens. CSF and blood culture samples reveal clear pathogen identification after enrichment, with corresponding Raman spectra for pathogens such as *E. coli* (Eco) and *Klebsiella pneumoniae* (Kpn). A mixed infection in a urine sample shows distinct morphologies of fungal cells (CaAl) and elongated bacterial cells, indicating beta-lactam resistance.

sensitivity of 98.45% demonstrates our system's robust capability to identify true positive cases effectively, which is critical in clinical diagnostics, especially for early and accurate pathogen detection. The lower specificity of 83.6% can be attributed to the inherently smaller number of negative samples ($n = 51$), as well as the fact that single-cell method is likely more sensitive than traditional culture-based methods, which may miss certain pathogens due to nutritional deficiencies or other medium-related issues, leading to negative results that might not fully represent the true infection status.

To examine the effect of complex matrices of the clinical samples on the capture efficiency of the enrichment system, we tested samples with known infections of *Staphylococcus capitis* and *E. coli* at a series of enriching duration (between 0.5 to 10 min) and collected the released droplets for agar culture (Supplementary Fig. S16). Negative controls (i.e., the initial priming of the system using deionised water) showed no cell colonies, while both enriched samples exhibited substantial cell growth, confirming that the enrichment process did not affect cell viability. The enriched cell number increased linearly with capturing duration (Supplementary Fig. S16c), further proving the excellent capturing efficiency of the system. Our system significantly improved pathogen visibility by enriching pathogen cells directly from complex clinical matrices and yielded 'clean' Raman spectra with less background signals (Fig. 7b). The enriched pathogens were effectively purified, and individual bacterial cells were clearly observable under the microscope in the processed bile, blood culture, and drainage fluid samples. High-quality spectra with clean background were observed for key pathogens like *Enterococcus faecalis* (Efa), *Acinetobacter baumannii* (Aba), *Klebsiella pneumoniae* (Kpn) (Fig. 7b), highlighting the system's ability to handle clinically significant samples with typically low pathogen counts. This capacity to process these precious samples without culture and accurately diagnose infections represents a significant advancement in rapid diagnostics.

Notably, our approach successfully identified polymicrobial infections that could be missed by traditional culture methods. A mixed-infection case from a urine sample (Fig. 7b) exemplified this, where both *Candida albicans* (fungal cells) and elongated *K. pneumoniae* (bacterial cells) were directly observed under the microscope. The elongated morphology of *K. pneumoniae* suggested beta-lactam resistance, highlighting the enrichment system's unique ability to provide detailed phenotypic information critical for patient management.

## Clinical ID of 120 patients

Out of the 291 concordant samples, the causative pathogens of 120 positive samples were identified by MALDI-TOF and were in the established single-cell Raman database. These 120 samples were subsequently blinded and sent for Raman measurements to assess the system's ID performance further (Fig. 8a; and Supplementary Data 5). These included 97 single infections and 23 mixed infections. These samples were subjected to single-cell Raman spectroscopy for rapid, culture-free pathogen identification using the constructed 1D ResNet model.

The ResNet model was adapted to handle mixed infection samples by encoding class labels with a MultiLabelBinarizer, using majority voting for final predictions across single cells per sample. The ResNet model classified 99 out of 120 patients identically to the gold-standard culture and MALDI-TOF MS results, achieving an overall agreement of 82.5% [74.7%, 88.3%] with a margin of error of ±6.8% and a large Cohen's h effect size of 1.9.

Detailed analysis (Fig. 8b-d) revealed several important patterns. The model achieved 80.4% accuracy (CI: [71.4%, 87.1%]) for single infections and 91.3% accuracy (CI: [73.2%, 97.6%]) for mixed infections, with no statistically significant difference between these groups ($p = 0.359$). Notably, the model demonstrated robust performance in identifying the more challenging mixed infections, which are often

difficult to detect and characterise using conventional diagnostic methods. The model maintained consistent accuracy across both hospitals, achieving 81.1% (CI: [72.6%, 87.4%]) at hospital 1 and 92.9% (CI: [68.5%, 98.7%]) at hospital 2 ($p = 0.460$), indicating the robustness and generalisability of the method.

Pathogen type significantly influenced identification accuracy ($p = 0.039$). The model accurately identified all fungal infections (100%, CI: [81.6%, 100%]) and mixed bacterial-fungal infections (100%, CI: [56.6%, 100%]), but demonstrated slightly lower accuracy for bacterial infections 78.6% (CI:[69.5%, 85.5%]). Fungal infections, in particular, can be challenging to diagnose and are often overlooked due to their slower growth rates, lack of specific symptoms, and the difficulty of culturing fungi from clinical samples compared to bacteria[48]. Species-specific analysis (Fig. 8e) revealed that performance varied considerably among individual bacterial species, with two poor performers as *Klebsiella pneumoniae* (7/21, 33.3%) and *Proteus mirabilis* (1/5, 20.0%), emphasising the need for future targeted refinement of the bacterial isolate database to enhance diagnostic performance.

## Discussion

This study presents a culture-free, transformative approach to clinical diagnosis of microbial infection by integrating the microfluidic enrichment system, single-cell Raman spectroscopy, and a deep learning model for rapid, accurate pathogen identification. Our platform is efficient (< 20 min), sensitive (< 2 CFU/ml), accurate (95.1% at species level for 342 clinical isolates), and scalable (validated in a clinical validation study involving 305 patients). This diagnostic platform can identify pathogens directly from clinical samples without the need for traditional culture-based methods, making it a significant advancement in clinical microbiology.

Our approach offers a substantial leap forward compared to existing diagnostic technologies (Table 1). Its capability of untargeted detection of unknown pathogens, short turnaround time and low limit of detection contrasts sharply with existing approaches that require lengthy sample preparation (e.g., the gold standard workflow)[48], and molecular approaches (e.g., PCR and WGS)[8–10] that provide faster identification but are inherently limited by the need for prior DNA isolation and targeted detection. Other emerging single-cell diagnostic platforms (e.g., 16S rRNA and digital PCR) still typically require $>10^3$ cells and are often complex to operate and prone to blockage[33,49]. By achieving rapid "sample-to-result" pathogen identification without the need for culture or complex preparatory steps and covering a wide dynamic range of pathogen load (<2 CFU/ml to $>10^8$ CFU/ml), our approach represents a significant advancement and a new capability for speed, sensitivity, and broad-spectrum applicability in clinical microbiology.

The microfluidic enrichment system, accompanied by visual inspection of enriched pathogens to determine positive or negative infection, achieved 98.5% sensitivity and 83.6% specificity. The observed specificity could be linked to the well-documented limitations of the gold-standard culture. Approximately one-third of sepsis cases have been shown to be culture-negative[50,51], with contributing factors including the fact that only about 1% of environmental bacteria are currently culturable. Additionally, culture specificity varies dramatically by patient population, from 80–90% in healthy outpatients to nearly 0% in patients with chronic catheters[52]. While additional validation methods such as either PCR confirmation of discordant cases or clinical outcome tracking, would strengthen future studies, our approach addresses the critical clinical need for rapid infection triage.

While studies[17,19–21] highlight the potential of Raman spectroscopy for rapid pathogen identification and diagnosis, these have often relied on cultured strains, labour-intensive sample preparation protocols[22,24,25], or pathogen concentrations higher than clinically relevant levels[26,53]. In contrast, our automated microfluidic system

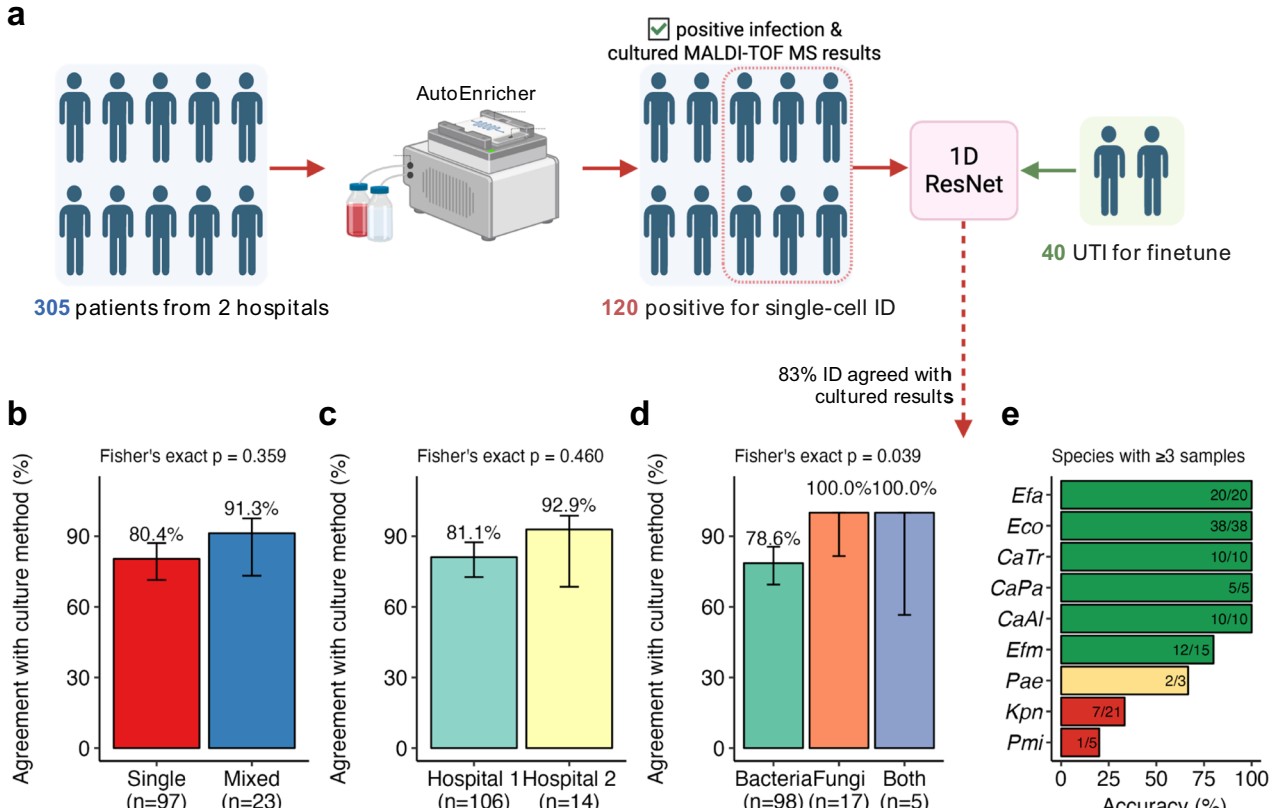

**Fig. 8 | Evaluation of single-cell ID on 120 patients. a** Study workflow: 305 patients from two hospitals were processed through AutoEnricher, yielding 120 positive samples with cultured MALDI-TOF MS results available for Raman spectroscopy validation. Fine-tuning dataset of 40 UTI samples was used to adapt the ResNet model for clinical validation. Overall agreement of 82.5% [74.7%–88.3%] was achieved between Raman identification and culture/MALDI-TOF results. Created in BioRender. Xu, J. (https://BioRender.com/qa6ykqf) (**b-d**) Bar plots showing averaged agreement with culture method further grouped into (**b**) single (80.4% [71.4%–87.1%]) or mixed infection (91.3% [73.2%–97.6%]), p = 0.359; **c** hospital 1 (81.1% [72.6%–87.4%]) or hospital 2 (92.9% [68.5%–98.7%]), p = 0.460; **d** samples containing bacteria (78.6% [69.5%–85.5%]), fungi (100.0% [81.6%–100.0%]) or both bacteria and fungi (100% [56.6%–100%]), p = 0.039. P values are from two-sided Fisher's exact tests. Bar heights indicate the mean accuracy for each group with error bars represent 95% confidence intervals calculated using Wilson score intervals. **e** Species-specific performance analysis for species with ≥3 clinical samples. Species are ranked by accuracy and colour-coded: green (≥ 80%), yellow (50–79%), red (< 50%). Source data for (**b**) to (**e**) are provided as a Source Data file.

eliminates manual intervention while maintaining detection sensitivity at clinically relevant concentrations. Our study is the largest of its kind, presenting a comprehensive dataset built directly from clinical isolates rather than standard ATCC strains. We established a robust database of 342 clinical isolates encompassing 36 bacterial and fungal species commonly found in hospital infections. Using an innovative 1D ResNet deep learning model, we achieved 95.1% accuracy for species identification among these clinical isolates. Since deep learning approaches suffer common drawbacks, such as limited interpretabilities of specific Raman bands[54] and a tendency to overfit[55], future improvements should focus on developing more interpretable models and incorporating uncertainty quantification to enhance clinical trust and adoption[56].

Uniquely, our study also moved beyond the laboratory, conducting a large-scale clinical study with 305 patients to validate the approach in real-world settings. The high accuracy in classifying pathogens from clinical samples (82.6% overall and 91.3% for mixed infections) compared to traditional culture methods underscores its effectiveness and readiness as a promising rapid diagnostic tool. Additionally, our system's demonstrated detection limit (<2 CFU/ml) which, combined with successful processing of 15 blood culture samples showing effective pathogen isolation from complex blood matrices (Supplementary Fig. S13), establishes technical feasibility for bloodstream infection diagnosis.

Several limitations should be acknowledged. First, our Raman spectroscopic validation was conducted on 120 of the 254 positive samples due to logistic constraints. While our power analysis confirmed statistical adequacy for technology validation purposes, larger sample sizes would strengthen clinical validation claims and enable more robust subgroup analyses. Additionally, our validation was predominantly based on urine samples (n = 298/305), with limited representation of other sample types. While we demonstrated technical feasibility with blood culture samples and other sterile site specimens, broader validation across diverse clinical samples would enhance generalisability and clinical impact.

Second, the lower accuracy observed in bacterial identifications (78.6%) compared to fungal identifications (100%) (Fig. 8d) indicates a need for further refinement of the bacterial isolate database with expanded strain diversity, particularly resistance variants. Additionally, the use of traditional culture as a reference standard in our clinical trial may introduce bias, as culture methods are inherently selective and often fail to detect fastidious or low-abundance pathogens, which may contribute to the discordant cases in the clinical study. Mixed infections, where culture often identifies only one pathogen, and instances of negative cultures but positive single-cell results due to low pathogen abundance or fastidious organisms, can add uncertainty to the gold standard results. Not all the microbial pathogens present can grow due to the selective nature of culture media, for example, in the

**Table 1 | Comparison of our approach with existing technologies**

| Approaches | Sample culture | Measured subjects | Pathogen identification | ID speed | Limit of detection |
|---|---|---|---|---|---|
| Our approach | No | Individual cells | Unknown pathogens | < 20 min | < 2 CFU/ml |
| Gold standard workflow (MALDI-ToF)[42] | Yes | Pure isolates, population | Unknown pathogens | > 2 days | ~$10^5$ CFU/ml |
| Polymerase Chain Reaction (PCR)[6,7] | No, need DNA isolation | DNA, to detect specific genes | Targeted pathogens | > 2 h | > 100 CFU/ml |
| Whole genome sequencing[10] | No, need DNA isolation | DNA, for genome sequencing | Known pathogens and AMR genes | > 6 h | > $10^5$ CFU/ml |
| Metagenomics[8,9] | No, need DNA isolation | DNA, for genome sequencing | Known pathogens and AMR genes | > 6 h | > $10^5$ CFU/ml |
| Other single-cell approach[27] | No | Individual cells (e.g., digital PCR, 16S rRNA) | Targeted pathogens | > 1 h | > $10^3$ CFU/ml |

polymicrobial infections involving both fungi and bacteria. Additionally, the accuracy of MALDI-TOF for identifying microbial isolates varies between different species (e.g., 74.5% to 98.8%)[48,57]. Future work of establishing a more comprehensive gold standard using techniques like next-generation sequencing will help better validate the single-cell method's performance and address selectivity-driven biases.

While preliminary data suggest some capability to distinguish resistance profiles among *E. coli* isolates (Supplementary Fig. S9), detailed antimicrobial susceptibility testing (AST) is essential for comprehensive clinical decision-making. With enriched pathogens, our system can be readily integrated with established rapid AST technologies, including $D_2O$-based Raman approaches[25,39,58] and microfluidic platforms[30,32,59–61], enabling pathogen identification and susceptibility testing within a few hours.

Nevertheless, as demonstrated in the clinical study, our approach can effectively address these real-world challenges: the high capturing efficiency of our enrichment system reveals a comprehensive profile of pathogens within the clinical samples, which can reduce the false-negative cases due to unsuccessful culture of certain pathogens. A distinct advantage of our single-cell Raman classification is its ability to detect mixed infections, as demonstrated with the urine sample containing both *C. albicans* and *K. pneumoniae*. In the clinical study, we found > 50% positive cases had more than one morphological type of bacteria in the Raman scan. This ability to detect complex infections and potential resistance phenotypes, such as β-lactam resistance, offers a substantial advantage over conventional diagnostic methods and is vital for guiding effective antimicrobial therapy and combating antibiotic resistance.

For microfluidic systems to be adopted in clinical settings, automation and workflow integration are essential for use by non-specialists[62,63]. Our system demonstrates seamless integration potential: negative samples enable immediate antibiotic stewardship decisions (<12 min), while positive samples guide species-directed therapy selection (<20 min), replacing prolonged broad-spectrum empirical treatment. The visual inspection protocol requires only standard microscopy skills to determine pathogen density above/below established thresholds, as demonstrated by routine laboratory technicians during our clinical validation without specialised training. The microfluidic enrichment system benefits from automated processing and disposable dialysis-DEP chips, which reduces contamination risks, maintenance burden and variability due to manual operations.

Importantly, as we demonstrate, both the disposable dialysis and DEP chips can be manufactured industrially, allowing for scalable mass production at low cost. The performance of successfully testing 305 clinical samples demonstrated the usability of our system in two distinct hospital environments. Following further device optimisation, studies will be needed to show how the instrumentation fits into different clinical workflows in different settings, comparing for example its use in large hospitals or centralised reference laboratories against its role within rural or community healthcare centres. Such analyses will inevitably involve more detailed testing along with associated health economic technology assessment analysis.

In conclusion, our approach is simple to operate, requiring no culture or destructive processing, and offers a rapid "sample-to-report" workflow within 20 min, meeting the urgent diagnostic need for rapid identification in ICUs and in the primary care where timely detection of bacterial infections is critical. It represents a significant advancement in clinical microbiology. Its ability to rapidly and accurately identify pathogens, detect mixed infections, and provide insights into antibiotic resistance has the potential to revolutionise infectious disease management, improve patient outcomes, and contribute to the global fight against antibiotic resistance. To realize this potential, future studies will focus on integrating the microfluidic enrichment system with AST routines and clinical workflows. Additionally, conducting larger multi-centre trials across broader patient populations and evaluating its impact on patient outcomes will be essential for widespread adoption.

## Methods

### Inclusion and Ethics

This study was approved by the ethics committee of Peking Union Medical College Hospital (No. S-K676), Huashan Hospital (No. 2020–907), and Sir Run Run Shaw Hospital (No. 20200316-33) in China. Only anonymised analytical data was used, and data produced in this study was not used for the treatment or management of patients. No sex- and gender-based analyses have been performed.

### Design and fabrication of the Dialysis and DEP devices

The geometry of the DEP chip was first simulated using a COMSOL model. The design of both the dialysis chip and DEP chip were then transferred to operation files using AutoCAD. The dialysis chip consists of two layers with mirrored serpentine channels, which were engraved using CNC milling on 5 mm PMMA (Supplementary Fig. S17). A 0.45 μm porous membrane (Advantec A045H047A, Japan) was placed between the two channel layers, and the dialysis chip was assembled by screwing the layers together. The membrane retains all cells including bacteria, white blood cells, and red blood cells in the sample channel, while allowing only electrolytes to diffuse through for conductivity reduction. Subsequent separation of bacteria from other cellular components occurs during the DEP enrichment step, where bacteria are selectively captured based on their distinct dielectrophoretic properties and size characteristics. It should be noted that sample solutions flow parallel to the dialysis membrane, minimising the risk of clogging. No membrane failures or channel blockages were observed during the clinical validation study.

The DEP chip features a 20 μm wide chevron electrode array over a 20 mm length to concentrate pathogens toward the middle and bottom of the channel. This is followed by a 20 μm wide interdigitated electrode (IDT) array extending over a length of 10 mm for capturing bacteria. The DEP electrodes and opening area were made of nickel using electroplating. The channel was formed with SU-8 using a standard photolithography method. The DEP chip was bonded by thermally pressing two pieces of 500 μm PMMA sheet at 100 °C. All chips were manufactured using industry high-volume production processes at Epigem Limited, offering a significant advantage in future scale-up to reduce the cost.

## Control hardware of the enrichment system

In the system's setup, a PicoScope 2206 was used to generate a 2 Vpp, 100 kHz sine signal and monitor the output signal. An AD621 instrument amplifier with a non-inverter amplification circuit was used to amplify the original waveform with a gain of 20, resulting in a final signal of 40 Vpp, 100 kHz. A relay system (Elegoo 8-channel relay board) was used to control the on/off states of the amplification circuit. Two stepper motors were used to drive the MasterFlex pump heads, functioning as water in and sample out. Another stepper motor was used to drive the MasterFlex EasyLoad pump head, functioning as sample in. A set of power supplies was used to provide voltages at 24 V, −24 V, 12 V and 5 V. A customised PCB was designed and manufactured to drive stepper motors and amplification circuit. A customised program written in Python was run on Raspberry Pi 4b to control the PicoScope, stepper motors, and relays.

## Automated workflow of the enrichment system

A user-friendly software was developed for the automated processing of clinical samples. The automated workflow operates at low electrical power (40 V at 100 kHz) with brief processing times (6–10 minutes), minimising potential heating effects. After inserting the "sample-in" tubing into the sample, the entire workflow can be automatically processed with pre-set parameters. The workflow consists of the following steps (Supplementary Fig. S1): (1) Sample loading and System Priming (2 minutes): Tubing is inserted into a sample and sterile DI water is delivered into the system at 2 mL/min to remove air. (2) Pathogen Enrichment (~5 min): Sample solution is pumped through the disposable dialysis-DEP device where pathogens are captured within the DEP chip. The duration is pre-set based on sample volume and flow rate (e.g., 5 min for 1 ml at 200 μl/min). (3) Cell Release (1 min): Captured pathogens are released through automated voltage shutdown and collected via hydraulic flow into a ~3 μl concentrated droplet. All operations are seamlessly synchronized with customised software (<10 min in total for steps 1-3). (4) Cell collection & Transfer (2 min): 2 μl of the enriched droplet is pipetted onto an aluminium-covered glass slide and air-dried for 30-60 seconds. (5) Microscopic Inspection & Triage (~2 min): Optical inspection determines positive/negative infection status, creating workflow branching. For negative samples (6 A): workflow stops (~12 min total). For positive samples (6B): Workflow continues with Raman spectroscopy (5 min for cell targeting and spectral acquisition of 15 cells) followed by automated ResNet classification (<1 min) for complete species identification (<20 min total).

## Bacterial culture and characterising capture efficiency

The *E. coli* 25922 strain was initially used for characterising the capture efficiency of the enrichment system. *E. coli 25922* cells were cultured at 37 °C at 200 rpm overnight in LB broth. To demonstrate the generalisability of the system across different pathogen types, capture efficiency was further validated using nine clinically isolated species: three Gram-negative strains (*Klebsiella aerogenes*, *Klebsiella oxytoca*, and *Klebsiella pneumoniae*), three Gram-positive strains (*Enterococcus faecalis*, *Enterococcus faecium*, and *Staphylococcus capitis*), and three

Candida species (*Candida albicans*, *Candida glabrata*, and *Candida parapsilosis*). These strains were cultured in TSB medium at 37 °C overnight, then harvested by centrifugation and resuspended in 500 μl DI water.

Prior to testing, the bacterial solution in DI water was diluted to $10^7$ CFU/mL to standardise the input concentration ($C_{input}$). 1 mL of the bacterial solution was passed through the enrichment system at each defined flow rate with AC voltage on to trap cells. Capture efficiency was tested across a range of flow rates of from 10–400 μl/min with 20 V or 40 V AC voltage. The outlet solution was collected and plated on LB agar plates after 24-h incubation at 37 °C for colony counting to determine the remaining cell concentration ($C_{outlet}$). The capture efficiency was calculated as the percentage of cells removed from the input solution using the equation:

$$\text{Efficiency} = \frac{(C_{input} - C_{outlet})}{C_{input}} \times 100\%$$

Where $C_{input}$ = Initial bacterial concentration (CFU/ml) determined by serial dilution and plate counting; $C_{outlet}$ = Final bacterial concentration (CFU/ml) in the outlet after enrichment processing. Each measurement represents the average of three independent experiments.

## Fluorescence imaging

The variations and difficulties of using agar plates culturing low numbers of bacteria cells are well-known. Therefore, a kanamycin-resistant and RFP-expression *E. coli* strain (MG1655::PLacUV5-mRFP) was used as a model strain to prepare low bacteria load samples, which allows fluorescence imaging-based characterisation. The RFP *E. coli* strain was cultured in LB Broth with 50 μg/ml kanamycin overnight. Series dilution of the overnight cultured RFP *E. coli* was performed to achieve different bacterial concentrations prior to testing. The RFP *E. coli* solution was pumped at defined flow rate (e.g., 200 μl/min). Time-lapse fluorescence images were recorded once the AC voltage was turned on. The concentrations of all the input bacterial solutions were verified using agar plate counting (*n* = 3 for each solution). LB Agar was mixed with 50 μg/ml kanamycin to prevent contamination. The input bacteria number was estimated using the input RFP *E. coli* concentration, flow rate, and running time (e.g., N = 10 CFU/ml × 5 min × 0.2 mL/min). At least three independent tests were conducted for each condition using a new dialysis unit and DEP chip.

For fluorescence visualization of the above nine bacterial species, 1 ml of bacterial solutions were stained with 1 μl BactoView™ Live Red dye (Biotium; catalogue number: 40101), incubated at 37 °C for 30 min in the dark, then washed and resuspended in DI water. Prior to DEP testing, all bacterial solutions were diluted to ~5 × $10^3$ CFU/ml and processed through the enrichment system at 200 μl/min with 40 V AC voltage.

Optical images of DEP chips were acquired using a Zeiss Axio microscope (Carl Zeiss AG, Germany) with various objectives (5×/NA = 0.3, 20×/NA = 0.5 and 63×/NA = 0.7). Texas Red Fluorescence filter set (559 ± 34 nm excitation and 630 ± 69 nm emission) was used for fluorescence imaging of the RFP-expression *E. coli* strain and Bacto-View™ Live Red-stained bacteria. Morphological characterisation images were captured using a 100× EC Epiplan-Neofluar objective lens (NA = 0.9). All fluorescence images were analysed using ImageJ, where the images were binarized and the number of cells were counted using the built-in "particle counting" function.

## Positive/negative infection threshold determination by optical imaging

*E. coli*, *E. faecalis*, and *C. auris* were cultured in TSB medium at 37 °C overnight, washed six times with DI water, and counted microscopically. Serial dilutions were prepared and 2 μL aliquots were deposited on aluminium-coated glass slides and air-dried to achieve

specific cell numbers per spot ( ~ 20, 500, 1000, 2000, 5000, 10000, and 20000 cells/spot). For each bacterial density, 10 randomly selected areas from two independent replicates were analysed using a 20× objective lens. Pathogen counts were determined using ImageJ[64] software to establish visual detection thresholds for rapid assessment of infection loads. Mean minus standard deviation values were used as threshold criteria. Based on these calibrations, unified thresholds of 300 per mm² for bacteria and 200 per mm² for fungi were established as minimum criteria for positive infection classification. Positive infection status required pathogen density exceeding these thresholds, corresponding to > 1000 pathogens/ml in the analysed samples.

## Database of 342 clinical isolates

The clinical isolate database includes 342 isolates from 36 species, each with 300 single-cell Raman spectra (SCRS). In total, the database consists of a total of 102,600 SCRS, specifically, 167 Gram-negative bacteria (50,100 SCRS), 81 Gram-positive bacteria (24,300 SCRS) and 94 fungi isolates (28,200 SCRS). The abbreviations of the 36 species are in Supplementary Data 1 and full information for the 342 clinical isolates can be found in Supplementary Data 2. Among all isolates, 12 *E.coli* (Eco) isolates were collected with their antibiotic resistance profiles characterised, including MEM-R (resistance to meropenem), ESBL+ (extended-spectrum beta-lactamase positive) and ESBL– (extended-spectrum beta-lactamase negative), each with four isolates (Supplementary Data 3).

In this study, all fungal clinical isolates were sourced from Peking Union Medical College Hospital, while all bacterial clinical isolates were obtained from Huashan Hospital in Shanghai, China. The fungal strains were cultured on yeast extract–peptone–dextrose (YPD) agar plates at 35 °C for 16 h. A single colony was then transferred to 5 mL of YPD medium and incubated at 35 °C for 16 h with shaking at 180 rpm. Bacterial isolates were cultured on tryptone soya agar (TSA) plates at 37 °C for 24 h. A single colony from each plate was transferred to 5 mL of tryptone soy broth (TSB) medium and incubated at 37 °C for 16 h with shaking at 180 rpm. Once the bacterial or fungal cells reached the stationary phase, 1 mL of the sample was washed three times with sterile water. The washed cells were resuspended in 1 mL of sterile water, and 2 µL of each sample was placed onto an aluminium-coated Raman microscopic slide to dry at room temperature. For each isolate, three independent batches were prepared.

Additional Leave-One-Replicate-Out cross-validation experiments combined with comprehensive bootstrap analysis were done on nine representative species: Gram-negative bacteria (*Acinetobacter baumannii* [Aba], *Escherichia coli* [Eco], *Pseudomonas aeruginosa* [Pae]), Gram-positive bacteria (*Enterococcus faecalis* [Efa], *Staphylococcus aureus* [Sau], *Streptococcus pneumoniae* [Spn]), and fungi (*Candida albicans* [CaAl], *Candida glabrata* [CaGl], *Candida parapsilosis* [CaPa]). For each selected species, an additional independent biological replicate was prepared by culturing a single colony from the same clinical isolate used in the original database but grown independently on a different day under identical culture conditions.

## Clinical validation of 305 patients for "sample-to-result" diagnosis

A finetuning dataset was obtained from 40 positively infected urine samples with their Raman spectra used for ResNet model tuning (Supplementary Data 4). A clinical validation study was conducted with samples from 305 patients admitted to Huashan Hospital in Shanghai and Sir Run Run Shaw Hospital in Zhejiang, China. The collected samples included urine, bile, cerebrospinal fluid, drainage fluid, and puncture fluid (Supplementary Data 5).

All samples were processed using the established culture-free enrichment system. ~1 mL of patient sample was directly processed through the system at 200 µL/min flow rate. Captured pathogens were concentrated into a ~ 3 µL droplet (333-fold volume concentration), of which 2 µL was deposited onto aluminium-coated glass slides for Raman analysis and 1 µL was serially diluted and cultured on agar plates for validation of CFU counting. In the hospitals, these samples were cultured to obtain gold standard results, indicating either positive or negative culture outcomes.

Of the 254 positive samples, only 120 were selected for Raman spectroscopic analysis due to logistic constraints. These samples contained pathogens present in our Raman database and were analysed for species identification (Supplementary Data 5). These 120 samples, consisting of 97 single-infection samples and 23 mixed-infection samples (5 bacteria-fungi and 18 bacteria-bacteria), were blinded and subjected to single-cell Raman measurements for rapid culture-free identification. Mixed infections were identified by single-cell Raman analysis and the ResNet model when multiple species each represented ≥3 cells per species in the sample population. The performance was then compared with MALDI-TOF MS results. The 120-sample validation achieved adequate statistical precision ( ± 6.8% margin of error) and large effect size (Cohen's h = 1.94), though individual species representation varied. Post-hoc analysis confirmed robust statistical significance for primary endpoints.

## Blood culture samples

Fifteen blood culture samples were processed using the enrichment system to validate the technical potential for bloodstream infection detection (Supplementary Fig. S13). However, these samples were excluded from the culture-free "sample-to-result" validation study due to standard clinical collection protocols. In routine hospital practice, blood samples are collected directly into blood culture bottles containing pre-formulated culture media (BacT/ALERT or BACTEC systems) to prevent pathogen death during transport and ensure regulatory compliance. Including these cultured samples in our "culture-free" validation would have been methodologically inconsistent. However, these experiments demonstrated effective pathogen isolation from complex blood matrices and maintained detection sensitivity, confirming the technical feasibility of bloodstream infection diagnosis.

## MALDI-TOF mass spectrometry (MS) identification of positive results

Single colonies from positive cultures were sent to MALDI-TOF MS for pathogen identification as the gold standard (Supplementary Data 4 and 5), which later served to assess the accuracy of Raman-based identification results. A total of 305 clinical bacterial isolates were analysed. Each sample was measured once (n = 1). *Escherichia coli* ATCC 8739 was used as the quality control strain and was analysed at the beginning of each batch to verify system performance. For sample preparation, a single colony from a Columbia blood agar plate (incubated overnight at 35 °C) was directly smeared onto a target plate (MALDI-TOF MS target, bioMérieux). Each spot was then overlaid with 0.5 µL of 70% formic acid and allowed to dry at room temperature. Subsequently, 0.7 µL of matrix solution was applied and dried again before analysis. Mass spectrometry analysis was performed using a MALDI-TOF mass spectrometer (bioMérieux VITEK MS, model: IVD). Spectra were acquired in linear positive ion mode over a mass range of 2,000–20,000 Da. Each spectrum was generated by accumulating data from 100 laser shots per sample location. The instrument was calibrated using the VITEK MS Test Standard (bioMérieux) according to the manufacturer's instructions, as detailed in the VITEK MS Manual, Version 3. Raw spectral data were processed and analysed using the MYLA software (version 1.6.0) with the default KB 3.2 knowledge base. Spectral quality was automatically assessed by the software based on peak intensity and number. Bacterial identification was performed by comparing the acquired spectra with the reference library. A confidence score ≥ 9.0 was used as the threshold for reliable species-level identification, as defined by

the software's algorithm. No additional statistical tests were applied to the identification process.

## Single-cell Raman measurements, preprocessing and analysis

Single-cell Raman spectra (SCRS) were collected using either a HORIBA HR Evolution 800 (HORIBA, France) or a WITec Alpha300R (WITec, Germany) Raman microscope spectrometer by Shanghai Hesen Bio-tech Co./Shanghai D-Band Co. For the WITec spectrometer, a 532-nm laser with a power of ~8–11 mW was focused onto the sample using a 100× objective lens (100×/NA = 0.9, ZEISS, Germany). Measurements were taken with a 1200 mm/g grating, covering a spectral range of 331–1991 cm⁻¹, centred at 1200 cm⁻¹. For the HORIBA spectrometer, a 532-nm laser with a 75-mW power was directed onto the sample through a 100× objective lens (100×/NA = 0.9, Olympus, Japan) using a 25% neutral density filter. This setup used a 600 mm/g grating, with a spectral range of 279–2197 cm⁻¹, also centred at 1200 cm⁻¹. The acquisition time for each cell was 2–5 seconds. For larger fungal cells, the laser spot was slightly defocused to cover a larger cell area. The use of two Raman microscopes allowed for the collection of a large number of spectra and enabled the evaluation of instrument effects on classification results. Single-cell Raman spectra were acquired for 300 cells per clinical isolate for 342 isolates and 100 cells per replicate for the 9 representative species.

All preprocessing was performed under an R 4.2.2 environment. The initial quality control involved removal of abnormal or burnt high-intensity spectra and cosmic ray correction. Baseline fitting and sub-traction were performed for autofluorescence removal using the "baseline" package with the rolling ball algorithm[65] (ws = 100, wm = 100). The spectra were min-max normalised from 0 to 1, mitigating instrumentation variability and sample or experimental factors without altering biological content. To build a unified database, all spectra from two instruments were adjusted to a range of 400–1800 cm⁻¹, interpolating to 1000 wavenumbers. For visualisation of the high-dimensional Raman dataset, an unsupervised method of t-distributed stochastic neighbour embedding (t-SNE) was used to embed the dataset into a two-dimensional space by minimising the Kullback-Leibler divergence between the probability distributions in respective dimensional spaces.

## ResNet architecture and model training with clinical isolate database

A one-dimensional ResNet (ResNet-1D) model was constructed upon a previous report[19]. The initial layer consisted of a 1D convolution with 100 filters, each of size 7, followed by batch normalisation and ReLU activation. The network included 6 residual blocks, each containing 2 convolutional layers. Each convolutional layer had 100 filters with a kernel size of 3. A dropout layer with a dropout rate of 0.2 was applied after the first ReLU activation in each residual block. A global average pooling layer was applied to reduce each feature map to a single value. The final layer was a dense layer with softmax activation.

The model training and validation process was conducted using a stratified group five-fold cross-validation method to ensure that the class distribution is approximately the same in each fold and that spectra from the same isolate do not appear in both training and validation sets. For each fold, the model was compiled using the Adam optimiser with a learning rate of 0.0001, the categorical cross-entropy loss function, and accuracy as the evaluation metric. The model was trained for 50 epochs with a batch size of 32. During training, early stopping was employed if the validation loss did not improve for 10 consecutive epochs. The hyperparameters were decided after experimenting on the validation set.

After training, the model was evaluated on the validation set by a Monte Carlo simulation combined with majority voting across all five folds. For each fold, we conducted 100 trials, where in each trial, 10 random selections were made from the validation set. The selected spectra were used to make predictions, and the majority vote from these predictions determined the final predicted class for each isolate. After completing all trials for a fold, the final predicted class for each isolate was determined by the majority vote from all trials. The accuracy for each fold was calculated by comparing the final predicted classes with the true labels. This process was repeated for all five folds. The overall metrics, including the confusion matrix, mean accuracy, and standard deviation of the accuracy, were computed to assess the model's performance.

The Leave-One-Replicate-Out validation was performed by testing each new replicate against the pre-trained ResNet model, with final species identification determined by majority voting across all cells in the replicate. This approach validates model performance against biological variance between independent culture replicates. To determine the minimum number of spectra required for robust majority voting in clinical applications, we performed bootstrap analysis on each replicate. For spectrum counts ranging from 1 to 90, we performed 100 independent bootstrap iterations where spectra were randomly selected without replacement, subjected to ResNet model predictions, and final species identification determined by majority voting.

## Model fine-tuning and testing in clinical validation

A finetuning dataset generated from 40 clinical samples were used to fine-tune the model to bridge the gap between the isolate database and complex clinical samples (Supplementary Data 4). The pre-trained model was loaded and compiled with a lower learning rate (0.00001) for fine-tuning. The fine-tuning set was subjected to another round of 5-fold stratified grouped cross-validation similar to the initial model training process.

In the clinical validation, the 120 positive clinical samples processed via the enrichment system containing 23 mixed infections were measured by Raman spectroscopy and used to evaluate the model's performance (Supplementary Data 5). The class labels, which may contain mixed infections, were encoded using a MultiLabelBinarizer to handle multiple classes per sample. For each clinical sample, the majority vote determined the final predicted classes from all measured single cells. The proportion of correct predictions was calculated, and the overall accuracy of the model was determined by comparing with the cultured MALDI-TOF results. A confidence interval for the accuracy was computed using standard error and Z-scores for a 95% confidence level. This structured approach ensured the model's applicability to real-world clinical samples, highlighting its potential for rapid, culture-free pathogen identification.

## Statistics & Reproducibility

Statistical analyses were performed in R (4.2.2) and GraphPad Prism (8.0.2). Tests are detailed in the figure legends and include one-way ANOVA with Tukey's post hoc test, one-tailed unpaired t-tests, Fisher's exact test, and linear regression. Confidence intervals for proportions were calculated using the Wilson score method. All in vitro technology-validation experiments were repeated independently at least three times (typically n = 3 per condition), and no a priori sample-size calculation was applied for these assays. For the clinical validation cohort, the total sample size (n = 305) was determined by consecutive sample collection during the study period. The subset analysed by Raman identification (n = 120) was calculated using power analysis, requiring a minimum of 61 samples to estimate 80% target accuracy with 95% confidence and a 10% margin of error, and expanded to 120 to include a safety buffer. The reference database comprised 342 clinical isolates across 36 species to ensure representative coverage of major hospital pathogens. Samples were processed prospectively and were not randomized. Investigators performing Raman measurements and ResNet classification were blinded to the gold-standard (culture and MALDI-TOF) results. No data were excluded from analyses.

## Reporting summary

Further information on research design is available in the Nature Portfolio Reporting Summary linked to this article.

## Data availability

The datasets generated and analysed during this study are publicly available in the Open Science Framework (OSF) repository at https://osf.io/784pz/ (DOI: 10.17605/OSF.IO/784PZ). This includes the single-cell Raman spectra database for the 342 clinical isolates and the processed clinical validation dataset for 120 patients. Source data are provided with this paper. The anonymised analytical data used in this study are available in the OSF repository and in Supplementary Data 2-5. Source data are provided with this paper.

## Code availability

All relevant code and models are publicly available in the Open Science Framework (OSF) repository at https://osf.io/784pz/ (DOI: 10.17605/OSF.IO/784PZ). This repository includes: (1) The pre-trained 1D ResNet model and label encoder for direct inference; (2) Python scripts; and (3) A standalone graphical user interface (GUI) application (for macOS and Windows) to allow users to test the model without requiring a programming environment.

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

## Acknowledgements

This work was supported by Innovate UK AMRAR project (File reference 104984 to H.Y.), National Key R&D Program of China (MOST, 2018YFE0101800 to Y.Y.), and international collaboration project between University of Oxford and Suzhou Institute of Biomedical Engineering and Technology, Chinese Academy of Sciences. We also thank support from EPSRC (EP/M002403/1 and EP/ M02833X/1 to W.E.H.) and from MRC (MR/Z50628X/1 to H.Y.). National High Level Hospital Clinical Research Funding (2022-PUMCH-B-028 and 2022-PUMCH-C-060 to Q.Y.). We thank the support from Richard Traynor, John Gardner, Paul Havenhead, Dave Currie, Carl Jameson at Epigem Limited, technical team at D Band and technical team at James Watt Nanofabrication Centre at the University of Glasgow. We are grateful to Dr. Xin Guo and Mr. Jisen Chen for their assistance in testing the reproducibility of the code and model. We also thank Sue Cardy for her constructive suggestions on the AIMRAR project.

## Author contributions

Conceptualisation: Y.L., H.Y., W.E.H., Y.Y., J.M.C. Methodology: all authors. Investigation: Y.L., J.X., X.Y., X.L., Y.L., A.G., X.L., W.Y., X.C., S.C., Q.Z. Formal analysis: J.X., Y.L., X.L., H.Y. Writing – Original Draft Preparation: Y.L., J.X., H.Y. Writing – Review & Editing: all authors.

## Competing interests

Both Epigem and Shanghai D-Band were partially funded through by Innovate UK AMRAR project (File reference 104984), National Key R&D

Program of China (MOST, 2018YFE0101800). The remaining authors declare no competing interests.
