## [Peer Review file · Nature Communications]

Rapid culture-free diagnosis of clinical pathogens via integrated microfluidic-Raman micro-spectroscopy

Corresponding Author: Professor Huabing Yin

Version 0:

Reviewer comments:

Reviewer #1

(Remarks to the Author)

Summary: The study developed a novel, automated, integrated dialysis-dielectrophoresis (DDEP) microfluidic system for rapid pathogen detection. This system is designed to isolate and enrich pathogens directly from clinical samples, addressing the need for rapid and accurate diagnosis of infections. With this approach, the authors addressed a very important issue in the field of Raman-based biology, as the isolation of single bacterial cells from a complex matrix is a crucial step for subsequent Raman measurements. Additionally, the desalting step is essential for preparing the sample for efficient pathogen capture and analysis.

This work is of high interest to the community of clinical microbiologists, as it significantly enhances the speed and accuracy of clinical diagnostics.

After careful review, I would like to make a number of comments that should be considered for publication in a high-impact journal. The results were not always put into the right context. It would be helpful to structure the data more clearly and to make the correlations clearer. Some presentations appear over-motivated. A more objective presentation of the results would increase the credibility of the work. I recommend a major revision. I would like to express that I see the sample preparation strategy presented here as a highlight that definitely deserves to be published in a technological journal, considering the comments mentioned in point 1. However, I find the Raman spectroscopic study conducted on bacteria to be very weak and not in line with the established practices in the community.

1. Auto-enricher

The results relating to the auto-enricher are the highlight of the manuscript. The structure and working methods are clearly and comprehensibly presented. The evaluation with *E. coli* is very well explained. The recovery rate for *E. coli* is outstanding, and the experiments with fluorescent bacteria also demonstrate the potential of the method.

However, I take a critical view of the exemplary presentation of the catchability of *E. coli*. In order to prove the claimed generalizability of the auto-enricher, the recovery rate should be shown for at least two other gram-negative bacteria. Of particular interest would be *Klebsiella*, which often show difficulties in microfluidics due to their extracellular matrix. The recovery rate should also be demonstrated for three Gram-positive bacteria. The different cell wall structure often results in different behavior in the DEP field. Thirdly, the recovery rate should also be shown for at least three different *Candida* species. *Candida* cells are much larger than bacteria. Their behavior in fluidics should also be shown in a way that is representative for the readership. The validation with the clinical samples gives no information about the recovery rate.

2. Establish Raman identification (ID) database and classification models.

When establishing the Raman database, I have two points of criticism. In Raman spectroscopic identification of bacteria, there are two major challenges: firstly, the biological variance. When building an ID database from Raman spectra, this should be taken into account. The classification model should be trained with a Leave-One-Replicate-Out cross-validation. This means that at least three replicates drawn on independent days should be measured for each species. Training is done with two replicates, and the prediction is made with the third. The authors have only performed a 10-fold cross-validation here, where the predicted spectrum was not part of the training, but the training dataset only included spectra from the same biological replicate. The authors should conduct additional experiments with three representative species each of gram-positive, gram-negative, and *Candida*, and demonstrate the performance of the ResNet model with a Leave-One-Replicate-Out cross-validation.

What surprises me the most is that the authors measured their training dataset with two different devices with different optical

parameters (e.g., different gratings and thus different spectral resolutions). The transferability of Raman classification data from single-cell bacteria is still unresolved, both for devices from the same manufacturer and especially for devices from different manufacturers. The authors should demonstrate to the reader how reproducible the ResNet model can predict data recorded with device A using data recorded with device B.

3. Raman measurements of single cells

The authors fail to address a speed-limiting phase in Raman analysis concerning the measurement of individual cells, which impacts the overall speed of the approach. I am familiar with the two Raman instruments utilised for the investigation, the HORIBA HR Evolution 800 and the WITec Alpha300R. Both instruments require the manual identification, approach, and measurement of bacteria. What is the size of the area occupied by the cells? What is the number of fields of view that require scanning? Based on my experience, manually measuring more than 100 individual cells requires several hours. What was the required number of cells per patient sample that the authors needed to assess to achieve statistically robust results from their viewpoint?

Reviewer #2

(Remarks to the Author)

I read this article very carefully and found it to be an interesting attempt at developing a relatively automated workflow for the rapid identification of uropathogens. It's an intriguing idea and has some merit, but as I went through it, I couldn't help but have a few major concerns.

1. On the Novelty of the Article

The detection of pathogens in this study is achieved through two main steps: first, the development of a low-concentration pathogen enrichment device based on the principle of dialysis; and second, the establishment of a Raman spectroscopy database combined with a classification algorithm.

The first part can be considered a preprocessing step for Raman spectroscopy data acquisition, and I'll admit, this approach is relatively novel. Unlike traditional culture-based or nucleic acid detection methods, the Raman spectroscopy method used here requires the removal of impurities that interfere with single-cell Raman spectroscopy. This is an area that hasn't been explored much, so I can see the value here.

However, the second part—the combination of Raman spectroscopy and deep learning for pathogen identification—feels far less innovative. This has already been extensively reported in the literature. For example:

- 1) Ho, C.S., et al., Rapid identification of pathogenic bacteria using Raman spectroscopy and deep learning. *Nat. Commun.*, 2019;
- 2) Ogunlade, B., et al., *Proceedings of the National Academy of Sciences*, 2024;
- 3) X. Wang, Robust Spontaneous Raman Flow Cytometry for Single-Cell Metabolic Phenome Profiling via pDEP-DLD-RFC. *Adv. Sci.* 2023;
- 4) Lu, W., et al., Combination of an Artificial Intelligence Approach and Laser Tweezers Raman Spectroscopy for Microbial Identification. *Analytical Chemistry*, 2020.
- 5) Lu, W., et al., Identification of pathogens and detection of antibiotic susceptibility at single-cell resolution by Raman spectroscopy combined with machine learning. *Frontiers in Microbiology*, 2023. 13.

The authors claim in Line 79: "While recent studies have demonstrated the potential of Raman spectra in model studies, its real-world application to clinical practice remains limited, with current research often characterised by small-scale and unsystematic studies for a very narrow range of pathogen species, such as ATCC control strains." I have to say, this statement is simply not accurate. Just from the five articles I listed above, some have already established large Raman databases and applied them to clinical samples, achieving even higher accuracy than what's reported in this paper. Additionally, some studies have already used pDEP for sample manipulation combined with Raman spectroscopy. For example, group from the No.3 paper have directly performed Raman spectroscopy detection on microfluidic chips using pDEP. This approach is a step ahead of what's proposed in this paper, where the authors enrich the samples first and then transfer them for further Raman testing. Compared to their work, the system described here feels more like a prototype. To be fair, the authors have increased the database size and analyzed more real samples, but to me, this feels more like an accumulation of workload rather than a true methodological innovation or improvement.

2. On the Scientific and Clinical Significance of the Article

Reducing the detection time for pathogenic microorganisms is undoubtedly important in clinical practice. However, I feel that the scientific and clinical significance of this study hasn't been emphasized enough.

- Lack of Antibiotic Susceptibility Testing: The study focuses solely on pathogen identification but doesn't address antibiotic susceptibility testing, which is critical for guiding clinical treatment and combating antimicrobial resistance. Identification alone has limited value, especially since other culture-free methods can also achieve rapid identification. Throughout the paper, the authors repeatedly refer to "diagnosis," but this is misleading. Without antibiotic susceptibility testing, this process should be referred to as "identification," not "diagnosis."

- Limited Sample Scope: The study analyzed 120 clinical samples using Raman spectroscopy, but in several parts of the paper, the authors claim "over three hundred clinical samples," which is misleading. Furthermore, these 120 samples are almost entirely urine samples, with no blood samples included. Clinically, urinary tract infections (UTIs) are less significant than bloodstream or respiratory infections. Moreover, UTIs don't require the same level of diagnostic speed as bloodstream infections. By focusing on a single type of infection with relatively low urgency in clinical practice, the study's overall significance is greatly reduced. The title, abstract, and conclusions should clearly state that the method is currently effective only for urine samples, despite the inclusion of a few non-urine samples that lack representativeness.

- Unproven Effectiveness for Bloodstream Infections: The authors emphasize in the introduction that bloodstream infections involve low pathogen concentrations and require advanced preprocessing, but they don't demonstrate the method's effectiveness for such cases. Uropathogens typically exist in higher concentrations, requiring less stringent enrichment, which further diminishes the clinical significance of this technical approach.
- Due to the aforementioned issues with novelty, the scientific significance of this paper is limited.

3. On the Reliability of Data and Results

There are several inconsistencies in the data and descriptions that make me question the reliability of the results:

- 1) Blood Sample Results in Fig. 7: The figure provides results for blood samples, yet the 120 clinical samples analyzed via Raman spectroscopy don't include blood samples. Were blood samples actually tested? Does this method work for bloodstream infections? Given that white and red blood cells in blood wouldn't pass through the dialysis membrane and their quantities far exceed those of pathogens, this would inevitably affect pathogen detection.
- 2) Sample Selection: The study includes 305 patients, but only 120 samples were analyzed via Raman spectroscopy. Why? The authors state in Line 364 that these 120 samples were selected because their pathogens were included in the Raman database, but Table S5 suggests otherwise. Were poor data excluded?
- 3) Flow Rate and Retention Efficiency: The authors claim that retention efficiency remains high at a flow rate of 200 $\mu\text{L}/\text{min}$ (Line 195), but no supporting figures or tables are provided. In fact, the results in Fig. 3D seem to contradict this claim.
- 4) Enrichment Efficiency: The final concentration factor is unclear. How many bacteria are present in the 3 μL sample volume to be analyzed via Raman spectroscopy?
- 5) Definition of False Positives/Negatives: What constitutes a false positive or false negative in Line 329? How is positivity determined? Is it based on microscopic observation of bacterial morphology? How are live bacteria distinguished from dead bacteria?
- 6) Time Calculation: The authors claim that each test takes only 20 minutes. How is this calculated? Details on sample loading, washing, enrichment, transfer, drying, and Raman spectroscopy acquisition (e.g., how many spectra per sample?) are needed.
- 7) Low-Concentration Detection: The study demonstrates detection at a minimum concentration of 2 CFU/mL. Can lower concentrations be detected?
- 8) Mixed Infections: Are all mixed infections found in this study fungal-bacterial infections? If mixed bacterial infections are included, how were they detected? The paper doesn't provide enough details.

Minor Concerns

1. Dialysis membranes are prone to clogging. How did the authors address this issue?
2. The authors emphasize the advantages of the 1D ResNet network but don't discuss its limitations or drawbacks.
3. The calculation process and basis for capture efficiency (Lines 204–205) are unclear.
4. What do the green arrows in Fig. 4 represent?

In summary, while this paper proposes an interesting approach to uropathogen identification, it suffers from significant issues related to novelty, clinical significance, and data reliability.

Reviewer #3

(Remarks to the Author)

This study presents an approach for diagnosing clinical infections that claims to be groundbreaking, yet the underlying methodology is riddled with significant flaws that demand rigorous critical evaluation. While the authors tout a remarkable reduction in diagnostic time to under 20 minutes; a dramatic departure from the traditional two-day culturing process; this seemingly impressive progress is overshadowed by glaring vulnerabilities.

Firstly, the integration of complex components such as dialysis and dielectrophoresis introduces a level of sophistication that could result in a convoluted sample processing workflow. This complexity raises legitimate concerns about usability and accessibility in clinical settings. The potential for extensive calibration and maintenance poses an obstacle to widespread adoption and raises alarms about the consistency and reliability of results in high-pressure healthcare environments where timely and accurate diagnostics are crucial. The device has been built from the authors' earlier publication which has weakened the innovation and impact of this paper.

Furthermore, the ability of the system to detect pathogens at low concentrations is severely compromised by a fundamental flaw: the desalting process may dilute the very pathogens the method aims to detect. This dilution jeopardizes detection sensitivity, particularly in situations characterized by minimal pathogen loads.

This oversight raises profound questions about the dependability of such a diagnostic tool. The temperature sensitivity inherent in microfluidic devices adds yet another layer of concern. Temperature variations could adversely impact enzymatic reactions or the viability of pathogens, thereby compromising diagnostic accuracy. It is alarming to consider the implications of these factors on the reliability of the results, especially when patients' health and treatment depend on precise diagnoses. Despite claiming a sensitivity of 98.1%, a specificity of merely 83.6% is an alarming red flag that cannot be overlooked. This concerning statistic raises serious issues regarding the risk of false positives, misclassifying non-pathogens as threats could lead to inappropriate treatments, thereby endangering patient health.

The authors may need to confront this critical issue with comprehensive evidence; otherwise, they run the risk of foisting an unreliable diagnostic tool on the healthcare community. Additionally, discrepancies in diagnostic accuracy between

bacterial (78.8%) and fungal infections raise further alarms. The struggle to differentiate closely related bacterial species, compounded by reliance on an incomplete bacterial isolate database, reveals profound weaknesses in their approach. This inadequacy highlights significant gaps that must be addressed if any claims of diagnostic reliability are to be taken seriously. The study alludes to validation with diverse clinical samples, yet it appears to inadequately consider the complexities and variability of clinical environments. These factors are likely to influence test outcomes significantly, underscoring the need for further validation before making any claims about clinical utility. Hence, a structured cohort study is urgently required to confront and resolve these foundational shortcomings before the paper can be considered for publication. Until these critical issues are addressed, the viability of this diagnostic approach remains highly suspect.

Version 1:

Reviewer comments:

Reviewer #1

(Remarks to the Author)

I am happy with the revisions, can be accepted as it is.

(Remarks on code availability)

I was not able to load the raw data. I got the following error "Failed to fetch file metadata from WaterButler. Received response:"

Reviewer #2

(Remarks to the Author)

The authors have made extensive revisions to the manuscript, but significant shortcomings persist, and several of my prior questions remain unanswered.

Firstly, inconsistencies between the results and the supporting data raise serious concerns. Some claims about the outcomes are not backed by adequate or rigorous evidence.

1. The authors state that they collected 305 patient specimens and processed them through the Autoenricher, identifying 254 positive samples. Due to workload constraints, they randomly selected only 120 for Raman spectral acquisition. Yet they highlight the method's speed as a key advantage, with Raman spectral collection and analysis taking less than 5 minutes per sample (as shown in Figure 7). I cannot understand why analyzing the remaining 134 samples would pose such a burden, given that the total time required would be around 11 hours. Investing just 11 additional hours of hands-on time could double the sample size and data volume, so it makes little sense that the authors did not pursue this. Does this indicate that they have overstated the method's rapidity?

2. In the data table provided, the 120 samples analyzed by MALDI-TOF show 11 mixed infections, with two of those lacking identification results. In the main text, however, the authors mention 23 mixed infections. What accounts for this discrepancy between the datasets? If even basic statistical summaries contain errors, this casts doubt on the reliability of the results, including but not limited to those in Figure 8. Additionally, if there are species identification results that MALDI-TOF failed to provide, what were the Raman results in those cases, and how can the two methods be considered consistent? And again, the authors have not addressed my previous question about how the new method identifies mixed infections.

3. On line 442, the authors claim that the system can handle blood cultures with concentrations as low as 2 CFU/mL, yet no source for this data appears anywhere in the text. The authors also overlooked my question about how the dialysis system removes white and red blood cells from the blood. They report collecting 15 blood culture samples, but it is unclear whether these are positive blood cultures from culture bottles or raw blood samples. The sample photograph in Figure 7 shows a blood collection tube, suggesting raw blood samples. However, the authors refer to them as blood culture samples. How do they explain this discrepancy? Furthermore, the authors provide no Raman spectral analysis results for blood samples and attribute this omission to the process not aligning with current clinical standards. If these are indeed positive blood cultures, though, such analysis would fit established clinical diagnostic workflows. The absence of blood sample results substantially diminishes the significance of this study. If the method presented in this work can effectively analyze positive blood culture samples, then such analyses should be included—for instance, by examining 50 positive blood samples. Hospitals routinely process a substantial number of blood culture samples, and the authors' approach requires no more than 20 minutes per sample from receipt to report. Processing 50 samples would therefore take only about 17 hours of laboratory time. There is no valid reason to exclude this essential sample type.

Secondly, the authors' portrayal of their work's innovative contributions does not hold up under scrutiny.

4. In my previous comments, I cited five papers using Raman spectroscopy for pathogen diagnosis. The authors responded that all existing reports fail to address true complex clinical samples, thereby emphasizing the novelty and value of their work in comparison. This conclusion is inaccurate. For instance, researchers have performed Raman-based diagnosis on real urine samples after simple filtration and centrifugation (Yang, K., et al., Rapid Antibiotic Susceptibility Testing of Pathogenic Bacteria Using Heavy-Water-Labeled Single-Cell Raman Spectroscopy in Clinical Samples. *Anal Chem*, 2019. 91(9): p. 6296-6303). Others have used centrifugation alone to preprocess genuine urine and blood samples, followed by Raman spectroscopy with machine learning for classification and antimicrobial susceptibility testing—and notably, some authors of that study are also contributors to this paper, making it surprising that they overlooked their own team's work (Yi, X., et al., Development of a Fast Raman-Assisted Antibiotic Susceptibility Test (FRASST) for the Antibiotic Resistance Analysis of Clinical Urine and Blood Samples. *Analytical Chemistry*, 2021. 93(12): p. 5098-5106). Simulated (spiked) samples have

also been reported in several studies, such as the Ogunlade et al. (2024) paper I mentioned last time, as well as direct Raman analysis on filtered urine using surface-enhanced Raman spectroscopy active filters (Dryden, S.D., Anastasova, S., Satta, G. et al. Rapid uropathogen identification using surface enhanced Raman spectroscopy active filters. Sci Rep 11, 8802 (2021)). Another example involves on-chip Raman analysis of simulated blood (I-Fang Cheng, Chang, HC., Chen, TY. et al. Rapid (<5 min) Identification of Pathogen in Human Blood by Electrokinetic Concentration and Surface-Enhanced Raman Spectroscopy. Sci Rep 3, 2365 (2013)). A study even more similar to this one used a microfluidic chip to rapidly remove impurities from urine and perform on-chip Raman detection (Zhao, L., et al., Micro-flow cell washing technique combined with single-cell Raman spectroscopy for rapid and automatic antimicrobial susceptibility test of pathogen in urine. Talanta, 2024. 277) . These works not only analyze complex clinical samples but also demonstrate straightforward preprocessing methods. As a result, the present paper represents only an incremental advance over prior research. **Finally, I previously recommended that the authors highlight the limitations of their sample types in the title and abstract, but no such changes have been made.**

(Remarks on code availability)

Reviewer #3

(Remarks to the Author)

The authors' response to the significant concern regarding their test's 83.6% specificity raises serious questions about their methodology. Their justification for the false positive rate—claiming these may actually reflect true positives missed by culture methods—exhibits circular reasoning by relying on an unvalidated approach to reconcile discrepancies with the gold standard. Notably absent is any concrete evidence or independent validation that would support their contention about these false positives representing genuine infections. Furthermore, they fail to address the clinical implications of a 16.4% false positive rate, a critical oversight that raises concerns about real-world applicability.

The stark disparity in accuracy between bacterial (78.8%) and fungal identification (100%) is troubling yet inadequately explained. While the authors attribute this difference to "diverse physiological states," they provide no systematic analysis to substantiate their claims. Vague commitments to "database expansion" with no clear validation plans only amplify concerns about the reliability of bacterial identifications. Additionally, their assertion of robust clinical validation is undermined by several factors, including a lack of discussion regarding sample bias and the unexplained large accuracy variation between hospitals (81.1% vs. 93.3%). The absence of controls in comparing their method against multiple reference standards beyond cultures and inadequate power analysis further diminish the credibility of their performance claims given the small sample size of 305 patients.

The absence of systematic error analysis, overgeneralization of "significant advancements," and the lack of quantitative assessments of accuracy variations reflect a troubling trend of evasion. Concerns surrounding the regulatory pathway for their claimed specificity, integration into clinical workflows, economic implications of false positives, and the training required for "visual inspection" are glaring omissions that suggest the authors are not fully confronting the fundamental limitations of their approach. This raises serious questions about the readiness of their technology for clinical application, underscoring a pressing need for a systematic validation study that thoroughly assesses their method against multiple reference standards and establishes its clinical utility through evidence of impact on patient outcomes.

(Remarks on code availability)

Version 2:

Reviewer comments:

Reviewer #2

(Remarks to the Author)

1. The authors' refusal to test positive blood culture samples raises significant concerns about the method's efficacy for detecting bloodstream infections. Initially, they cited misalignment with clinical standards as the reason for not analyzing blood culture samples via Raman spectroscopy. However, their rebuttal admits that the 15 blood samples analyzed using the autoenricher were sourced from BacT/ALERT culture bottles—a . **This directly contradicts their original justification. Reacquiring dozens of fresh positive blood culture samples is logistically feasible, yet the authors declined, citing prior sample discarding. Given their emphasis on clinical applicability, the absence of bloodstream infection data severely undermines the study's persuasiveness, particularly for a journal like Nature Communications. Additionally, an inconsistency persists: Supplementary Figure S13 still labels samples as "raw blood samples," conflicting with the rebuttal's claim of using "positive blood culture samples."**

2. The authors' Table R1 raises questions about objectivity, as Yang et al. (2019) and Yi et al. (2021) already achieved sample-to-result urine sample analysis. Thus, this work (which mainly focused on urine samples) cannot be considered the "first true sample-to-report" system. Upon re-examining Zhao et al. (2024), I found their system to be fully automated (contrary to the authors' characterization of "semi-automated with manual loading"). In contrast, the authors' study requires manual loading of autoenriched samples into the Raman system. While I acknowledge this is the largest clinical study applying Raman spectroscopy to pathogen detection, the innovation level remains insufficient for Nature Communications, where groundbreaking advances are prioritized.

3. In order to assess the technical soundness of the work, I strongly request the authors to provide raw Raman spectra for 120 samples, (likely in .txt/.csv format) and corresponding identification results of each spectrum (probably in datasheets). The source data deposited only contains Raman spectra for 342 clinical isolates. Were these used to train the deep learning network? I tried to run the model to predict the classification of 342 clinical isolates from the database provided by the author. All of the spectra were predicted as species CaPa.

(Remarks on code availability)

I tried to run the model to predict the classification of 342 clinical isolates from the database provided by the author. All the spectra were predicted as species CaPa.

Reviewer #3

(Remarks to the Author)

The article has been significantly improved and I would recommend it to be considered for publication.

(Remarks on code availability)

Response to Reviewers

We thank all reviewers for their constructive comments. We have now conducted new experiments and data analysis to provide additional evidence. Below are point-by-point responses to the reviewers' comments.

Reviewer #1:

We thank the reviewer for their introductory comments, emphasizing the importance of sample preparation. We hope that we have now addressed the reviewer's general comments about how we performed the Raman spectroscopic study. We note that we have added significant new data to support our analysis.

1. Auto-enricher

The results relating to the auto-enricher are the highlight of the manuscript. The structure and working methods are clearly and comprehensibly presented. The evaluation with E. coli is very well explained. The recovery rate for E. coli is outstanding, and the experiments with fluorescent bacteria also demonstrate the potential of the method.

However, I take a critical view of the exemplary presentation of the catchability of E. coli. In order to prove the claimed generalizability of the auto-enricher, the recovery rate should be shown for at least two other gram-negative bacteria. Of particular interest would be Klebsiella, which often show difficulties in microfluidics due to their extracellular matrix. The recovery rate should also be demonstrated for three Gram-positive bacteria. The different cell wall structure often results in different behavior in the DEP field. Thirdly, the recovery rate should also be shown for at least three different Candida species. Candida cells are much larger than bacteria. Their behavior in fluidics should also be shown in a way that is representative for the readership. The validation with the clinical samples gives no information about the recovery rate.

Response: We thank the reviewer for their positive assessment of the AutoEnricher as the highlight of the manuscript and for recognizing the clear presentation of our methods and outstanding *E. coli* recovery rates. We fully agree with the reviewer's critical point regarding the need to demonstrate generalizability beyond *E. coli*, and we have conducted additional experiments on **capture efficiency of 10 species** to address each specific concern raised.

The new results show that AutoEnricher exhibited excellent capture efficiencies for these pathogen groups despite of their different cell wall properties and morphologies. In brief, the capture efficiencies are: Gram-negative (3 *Klebsiella* species: 89.1-99.5%), Gram-positive (3 species: 91.2-97.3%), and fungi (3 *Candida* species: 91.0-95.1%).

We have incorporated these results into the revision with **new Fig. 3e, Supplementary Fig S3 and Supplementary Fig S4, Supplementary Videos 2-11** and integrated text in **Methods and Results** sections.

Results Section (Page 6, Line 165)

“Validation of capture efficiency across pathogen groups

To demonstrate the generalizability of our system's functionality beyond E. coli, we systematically evaluated capture efficiency across 9 additional species representing major pathogen groups commonly encountered in healthcare settings. These included three Klebsiella species (K. aerogenes, K. oxytoca, and K. pneumoniae), which are known to produce extracellular matrix and can be challenging to handle in microfluidics; three Gram-positive species (Enterococcus faecalis, E. faecium, and Staphylococcus capitis), and three clinically relevant Candida species (C. albicans, C. glabrata and C. parapsilosis), each with distinctive cell wall properties and morphologies (Supplementary Figure S3).

Despite these differences, the enrichment system demonstrated excellent capture efficiencies across these species between 89.1% and 99.5% (Fig. 3e). Furthermore, time-lapse fluorescence imaging revealed an efficient trapping process across all species: all cells entering the field of view were captured and progressively accumulated at the electrode edges (Supplementary Figure S4, Supplementary Videos 2–11). This robust performance demonstrates its capability for blind testing of clinical samples containing unknown pathogen species.”

Methods Section (Page 19, Line 556)

“**Bacterial culture and characterising capture efficiency.** ... To demonstrate the generalisability of the system across different pathogen types, capture efficiency was further validated using nine clinically isolated species: three Gram-negative strains (*Klebsiella aerogenes*, *Klebsiella oxytoca*, and *Klebsiella pneumoniae*), three Gram-positive strains (*Enterococcus faecalis*, *Enterococcus faecium*, and *Staphylococcus capitis*), and three *Candida* species (*Candida albicans*, *Candida glabrata*, and *Candida parapsilosis*). These strains were cultured in TSB medium at 37°C overnight, then harvested by centrifugation and resuspended in 500 µL DI water.”

(Page 20, Line 590) For fluorescence visualization of the above nine bacterial species, 1 mL of bacterial solutions were stained with 1 µL BactoView™ Live Red dye (Biotium; catalogue number: 40101), incubated at 37°C for 30 minutes in the dark, then washed and resuspended in DI water. Prior to DEP testing, all bacterial solutions were diluted to approximately 5×10^3 CFU/mL and processed through the enrichment system at 200 µL/min with 40V AC voltage.”

Fig. 3 Characterisation of DEP chip for pathogen enrichment and capture efficiency. (e)

Comparative capture efficiency across pathogen groups. Capture efficiency measured for 10 different species representing Gram-negative bacteria, Gram-positive bacteria, and *Candida* species. Dashed line indicates 90% efficiency threshold. Data represent mean \pm SD from three independent experiments. Statistical analysis by one-way ANOVA with Tukey's post-hoc test. Abbreviations: Eco, *E. coli*; Kae, *Klebsiella aerogenes*; Kox, *Klebsiella oxytoca*; Kpn, *Klebsiella pneumoniae*; Efa, *Enterococcus faecalis*; Efm, *Enterococcus faecium*; Sca, *Staphylococcus capitis*; CaAl, *Candida albicans*; CaGl, *Candida glabrata*; CaPa, *Candida parapsilosis*.

Supplementary Fig. S3 Morphological characterization of test organisms. Bright-field images of representative bacterial and fungal species used for capture efficiency testing. *Klebsiella* species display rod-shaped morphology, Gram-positive strains show cocci-shaped morphology, and *Candida* species exhibit ovoid shapes and mother-daughter budding patterns. Scale bars: 5 µm. Abbreviations: Kae, *Klebsiella aerogenes*; Kox, *Klebsiella oxytoca*; Kpn, *Klebsiella pneumoniae*; Efa, *Enterococcus faecalis*; Efm, *Enterococcus faecium*; Sca, *Staphylococcus capitis*; CaAl, *Candida albicans*; CaGl, *Candida glabrata*; CaPa, *Candida parapsilosis*.

Supplementary Fig. S4 Time-lapse fluorescence imaging of pathogen capture. Representative sequences showing progressive accumulation of fluorescently labelled (a) 3 Gram-negative bacteria, (b) 3 Gram-positive bacteria, and (c) 3 *Candida* species on DEP electrodes over 120 seconds. AC voltage: 40V, flow rate: 200 μ L/min. Scale bars: 50 μ m. Abbreviations: Kox, *Klebsiella oxytoca*; Kae, *Klebsiella aerogenes*; Kpn, *Klebsiella pneumoniae*; Efa, *Enterococcus faecalis*; Efm, *Enterococcus faecium*; Sca, *Staphylococcus capitis*; CaAl, *Candida albicans*; CaGl, *Candida glabrata*; CaPa, *Candida parapsilosis*.

Supplementary Videos 3 to 11 recorded real-time trapping of each of the nice species on the IDT array, revealing cell behaviour in fluidics.

2. Establish Raman identification (ID) database and classification models.

2.1. When establishing the Raman database, I have two points of criticism. In Raman spectroscopic identification of bacteria, there are two major challenges: firstly, the biological variance. When building an ID database from Raman spectra, this should be taken into account. The classification model should be trained with a Leave-One-Replicate-Out cross-validation. This means that at least three replicates drawn on independent days should be measured for each species. Training is done with two replicates, and the prediction is made with the third. The authors have only performed a 10-fold cross-validation here, where the predicted spectrum was not part of the training, but the training dataset only included spectra from the same biological replicate. The authors should conduct additional experiments with three representative species each of gram-positive, gram-negative, and *Candida*, and demonstrate the performance of the ResNet model with a Leave-One-Replicate-Out cross-validation.”

Response: We appreciate the reviewer's concern about biological variance. We respectfully clarify our original design to avoid any misunderstanding and provided **additional validation data and tests** as requested.

Firstly, to clarify our original methodology employed **stratified group five-fold cross-validation**, not simple 10-fold cross-validation as suggested. Specifically, we ensured that "*spectra from the same isolate do not appear in both training and validation sets*" (Methods section). This approach is actually more stringent than traditional Leave-One-Replicate-Out validation because **inter-isolate variance > intra-isolate variance**. We classified at the species level using different clinical isolates of the same species, which introduces greater biological variance than culture replicates from a single isolate. Our approach better reflects real-world scenarios where new clinical isolates (not just culture replicates) need to be identified.

Secondly, to directly address the reviewer's specific concern, we conducted **additional experiments with independent biological replicates**. We selected 9 representative species: 3 Gram-negative bacteria (*Acinetobacter baumannii* [Aba], *Escherichia coli* [Eco], *Pseudomonas aeruginosa* [Pae]), 3 Gram-positive bacteria (*Enterococcus faecalis* [Efa], *Staphylococcus aureus* [Sau], *Streptococcus pneumoniae* [Spn]), and 3 fungi (*Candida albicans* [CaAl], *Candida glabrata* [CaGl], *Candida parapsilosis* [CaPa]). We cultured independent replicates from the same clinical isolates used in our original database.

We used the same approach of random sampling of 10 spectra from each of the replicate for testing the trained ResNet model presented in the manuscript. We have achieved **100% accuracy** at the isolate level using majority voting (new **Supplementary Fig. S10**) - perfect species identification across all tested replicates.

In conclusion, both our **original inter-isolate validation** and the **additional Leave-One-Replicate-Out validation** demonstrate the robustness of our ResNet model against biological variance. The

perfect accuracy achieved in the Leave-One-Replicate-Out validation, combined with our comprehensive inter-isolate validation, provides strong evidence for the clinical reliability of our approach.

These new results are included in the revision and copied below.

Results Section (Page 10, Line 276)

“To further validate against biological variance, we conducted Leave-One-Replicate-Out cross-validation using 9 representative species across major pathogen groups. Independent biological replicates achieved 100% accuracy at the isolate level using majority voting and 82.7% mean single-cell accuracy (Supplementary Fig. S10). Bootstrap analysis revealed that reliable clinical identification requires minimal spectra: 5 spectra achieved 96.6% accuracy, while 10 spectra achieved 98.4% accuracy (Supplementary Fig. S11). This enables rapid clinical testing within our 20-minute workflow (Supplementary Fig. S1), requiring only 4-5 minutes for Raman measurements of 10-15 cells.”

Methods Section (Page 21, Line 637)

*“Additional Leave-One-Replicate-Out cross-validation experiments combined with comprehensive bootstrap analysis were done on nine representative species: Gram-negative bacteria (*Acinetobacter baumannii* [Aba], *Escherichia coli* [Eco], *Pseudomonas aeruginosa* [Pae]), Gram-positive bacteria (*Enterococcus faecalis* [Efa], *Staphylococcus aureus* [Sau], *Streptococcus pneumoniae* [Spn]), and fungi (*Candida albicans* [CaAl], *Candida glabrata* [CaGl], *Candida parapsilosis* [CaPa]). For each selected species, an additional independent biological replicate was prepared by culturing a single colony from the same clinical isolate used in the original database but grown independently on a different day under identical culture conditions.”*

(Page 25, Line 743)

“The Leave-One-Replicate-Out validation was performed by testing each new replicate against the pre-trained ResNet model, with final species identification determined by majority voting across all cells in the replicate. This approach validates model performance against biological variance between independent culture replicates. To determine the minimum number of spectra required for robust majority voting in clinical applications, we performed bootstrap analysis on each replicate. For spectrum counts ranging from 1 to 90, we performed 100 independent bootstrap iterations where spectra were randomly selected without replacement, subjected to ResNet model predictions, and final species identification determined by majority voting.”

Supplementary Fig. S10 Leave-One-Replicate-Out Cross-Validation Results. Confusion matrix for Leave-One-Replicate-Out cross-validation using 9 independent biological replicates spanning representative Gram-negative bacteria (Aba, Eco, Pae), Gram-positive bacteria (Efa, Sau, Spn), and fungi (CaAl, CaGl, CaPa).

2.2. What surprises me the most is that the authors measured their training dataset with two different devices with different optical parameters (e.g., different gratings and thus different spectral resolutions). The transferability of Raman classification data from single-cell bacteria is still unresolved, both for devices from the same manufacturer and especially for devices from different manufacturers. The authors should demonstrate to the reader how reproducible the ResNet model can predict data recorded with device A using data recorded with device B.

Response: We thank the reviewer for raising this critical concern about cross-instrument transferability in Raman spectroscopy. This is indeed a fundamental challenge in the field, and we have designed our study specifically to address this limitation.

Our decision to use two different Raman instruments (HORIBA HR Evolution 800 and WITec Alpha300R) was based on two key considerations: (1) **practical necessity** to establish a comprehensive database of 342 clinical isolates (102,600 spectra) within a reasonable timeframe, parallel data collection was essential; (2) **real-world applicability**; as our ultimate goal is clinical implementation across diverse healthcare settings, demonstrating cross-instrument robustness was a primary objective from the study's inception.

To address the reviewer's concern, we conducted **additional systematic cross-instrument validation** using a representative subset of 6 *Candida* species (81 isolates, 24,300 spectra). We chose this subset because: (1) it contains balanced numbers of HORIBA-only, WITec-only isolates (**Response Fig. R1**), (2) *Candida* species within the same genus represent a challenging test case for species differentiation, and (3) full dataset retraining across multiple models requires substantial computational resources.

Example single-machine training

Species	Both	HORIBA_only	WITec_only	Total
1 Candida albicans	6	Train 6	Test 2	14
2 Candida glabrata	7	4	2	13
3 Candida guilliermondii	5	5	3	13
4 Candida krusei	7	3	4	14
5 Candida parapsilosis	7	1	5	13
6 Candida tropicalis	6	6	2	14

Example mixed-machine training

Species	Both	HORIBA_only	WITec_only	Total
1 Candida albicans	6	Train 6	Test 2	14
2 Candida glabrata	7	4	2	13
3 Candida guilliermondii	5	5	3	13
4 Candida krusei	7	3	4	14
5 Candida parapsilosis	7	1	5	13
6 Candida tropicalis	6	6	2	14

Species	Both	HORIBA_only	WITec_only	Total
1 Candida albicans	6	Train 6	2	14
2 Candida glabrata	7	4	Test 2	13
3 Candida guilliermondii	5	5	3	13
4 Candida krusei	7	3	4	14
5 Candida parapsilosis	7	1	5	13
6 Candida tropicalis	6	6	2	14

Response Fig. R1 Cross-instrument validation design for assessing machine transferability.

Example of single-machine and mixed-machine training approaches using 6 *Candida* species (81 isolates, 24,300 spectra). Numbers represent the number of isolates present in each group. Isolates measured by both instruments were excluded from the validation to assess machine transferability. This additional training-testing approach is demonstrated in **Response Fig. R1**, split into **single-machine training** to demonstrate direct transfer and **mixed-machine training** to demonstrated smarter transfer.

- 1) *Single-machine training*: Models were trained exclusively on data from one instrument (HORIBA or WITec) and tested on data from the other instrument. This approach assessed the direct transferability of models across different instrumental configurations.

- 2) *Mixed-machine training*: Models were trained on combined datasets from both instruments using Leave-One-Species-Out cross-validation. In this approach, all isolates from one species were held out for testing while the model was trained on isolates from all other species, regardless of the instrument used. This approach simulates real-world clinical scenarios where the system encounters a previously unseen species that needs to be added and tested in the existing database.

Performance was evaluated at both spectrum-level (individual spectra classification) and sample-level (majority voting from multiple spectra per isolate) accuracy.

Key findings are summarised and included in new **Supplementary Fig. S12**. Single-machine training (training on one instrument, testing on another) showed poor transferability, with HORIBA→WITec achieving 31.8% spectrum-level and 44.4% sample-level accuracy, while WITec→HORIBA showed even lower performance at 8.1% spectrum-level and 4.4% sample-level accuracy. These results confirm the reviewer's assertion that direct transfer between instruments represents a fundamental challenge in Raman classification. However, mixed-machine training (our implemented approach) demonstrated robust cross-instrument performance, achieving 84.2% spectrum-level and **90.9% sample-level accuracy for HORIBA spectra**, and 89.4% spectrum-level and **100% sample-level accuracy for WITec spectra**. This demonstrates that our ResNet model, when trained with data from both instruments, can successfully **generalize the spectral characteristics across different instrumental configurations** and make accurate classifications even when encountering previously unseen species.

In our main clinical database of 342 isolates, approximately one-third were measured on both instruments, one-third exclusively on HORIBA, and one-third exclusively on WITec. Importantly, our original validation (Figure 6) demonstrated consistent high performance (95.1% accuracy) across all isolates regardless of the measuring instrument, confirming that our mixed-machine training approach successfully addresses cross-instrument variability.

We have added detailed methodology and results to the revised manuscript to ensure transparency in our cross-instrument validation approach:

Results Section (Page 10, Line 284)

“Cross-instrument validation on the ResNet approach.

To address the challenge of cross-instrument transferability in Raman spectroscopy, we conducted systematic validation using a dual-instrument approach. We evaluated cross-instrument performance using a representative subset of 6 Candida species (81 isolates, 24,300 spectra) measured on both HORIBA HR and WITec instruments.

Single-machine training (training on one instrument, testing on another) showed limited transferability, with HORIBA to WITec achieving 31.8% spectrum-level and 44.4% sample-level accuracy, while WITec to HORIBA showed 8.1% spectrum-level and 4.4% sample-level accuracy (Supplementary Fig. S12). However, our implemented mixed-machine training approach demonstrated robust cross-instrument performance, achieving 84.2% spectrum-level and 90.9% sample-level accuracy for HORIBA spectra, and 89.4% spectrum-level and 100% sample-level accuracy for WITec spectra when leaving one species out for independent validation.”

Supplementary Fig. S12 Cross-instrument performance comparison between single-machine and mixed-machine training approaches. Spectrum-level accuracy represents the percentage of individual Raman spectra correctly classified, while sample-level accuracy represents the percentage of isolates correctly identified based on majority voting from multiple spectra per isolate.

3. Raman measurements of single cells

The authors fail to address a speed-limiting phase in Raman analysis concerning the measurement of individual cells, which impacts the overall speed of the approach. I am familiar with the two Raman instruments utilised for the investigation, the HORIBA HR Evolution 800 and the WITec Alpha300R. Both instruments require the manual identification, approach, and measurement of bacteria. What is the size of the area occupied by the cells? What is the number of fields of view that require scanning? Based on my experience, manually measuring more than 100 individual cells requires several hours. What was the required number of cells per patient sample that the authors needed to assess to achieve statistically robust results from their viewpoint?

Response: We thank the reviewer for this important practical concern about measurement throughput. We address this through both clarification of our approach and new experimental validation of minimum spectra requirements.

Firstly, we would like to highlight the key distinction between **database training vs. clinical testing**. The reviewer correctly notes that our database construction involved measuring 300 spectra per isolate, which indeed required substantial time investment. However, this is a **one-time database construction phase** that differs fundamentally from **routine clinical testing**. For clinical testing, our original approach only **used 15 spectra** for rapid pathogen identification.

Secondly, to further **quantify the minimum number of spectra needed** for reliable clinical identification, we conducted a bootstrap analysis using our Leave-One-Replicate-Out validation data. For each replicate, we applied random sampling of 1-90 spectra per replicate across 100 bootstrap iterations per spectrum count, with majority voting determining the final classification for each iteration.

The results are presented in the new *Supplementary Fig. S11* and summarised as below:

- **3 spectra:** 92.9% mean accuracy
- **5 spectra:** 96.6% mean accuracy
- **10 spectra:** 98.4% mean accuracy
- **15 spectra:** 99.4% mean accuracy

- **≥20 spectra:** >99.7% mean accuracy

Thirdly, regarding the specific technical questions raised by the reviewer, bacterial cells typically occupy areas of 1-2 μm in diameter, while fungal cells are larger at 3-8 μm . For identifying 10-15 cells in positive samples with concentrations above 1000 CFU/mL (new **Supplementary Fig. S14**), only 1-2 fields of view are typically required, given the post-enrichment cell density of $>10^3$ cells in the 2 μL samples. While the reviewer correctly notes that measuring 100 individual cells requires several hours during database construction, clinical testing requires only 10-15 cells, which can be completed in **approximately 4-5 minutes total measurement time**. This includes cell targeting and focusing (10-15 seconds per cell) plus spectral acquisition (2-5 seconds per cell), resulting in 15-20 seconds per cell overall, making the entire measurement process highly practical for routine clinical implementation within our 20-minute sample-to-report workflow (new **Supplementary Fig. S1**).

These new results are included in the revision and pasted below.

Results Section (Page 10, Line 276)

“To further validate against biological variance, we conducted Leave-One-Replicate-Out cross-validation using 9 representative species across major pathogen groups. Independent biological replicates achieved 100% accuracy at the isolate level using majority voting and 82.7% mean single-cell accuracy (Supplementary Fig. S10). Bootstrap analysis revealed that reliable clinical identification requires minimal spectra: 5 spectra achieved 96.6% accuracy, while 10 spectra achieved 98.4% accuracy (Supplementary Fig. S11). This enables rapid clinical testing within our 20-minute workflow, requiring only 4-5 minutes for Raman measurements of 10-15 cells.”

Supplementary Fig. S11 Bootstrap analysis demonstrating minimum spectra requirements for reliable pathogen identification.

Supplementary Fig. S1 Complete sample-to-report workflow timeline for rapid pathogen identification. The integrated AutoEnricher-Raman spectroscopy platform enables rapid pathogen detection and identification through a branched workflow based on infection status.

Methods Section (Page 18, Line 539)

“The workflow consists of the following steps (Supplementary Fig. S1): (1) Sample loading and System Priming (2 minutes): Tubing is inserted into a sample and sterile DI water is delivered into the system at 2 mL/min to remove air. (2) Pathogen Enrichment (~ 5 minutes): Sample solution is pumped through the disposable dialysis-DEP device where pathogens are captured within the DEP chip. The duration is pre-set based on sample volume and flow rate (e.g., 5 minutes for 1 mL at 200 $\mu\text{L}/\text{min}$). (3) Cell Release (1 minute): Captured pathogens are released through automated voltage shutdown and collected via hydraulic flow into a $\sim 3\mu\text{L}$ concentrated droplet. All operations are seamlessly synchronized with customized software (< 10 minutes in total for steps 1-3). (4) Cell collection & Transfer (2 minutes): 2 μL of the enriched droplet is pipetted onto an aluminium-covered glass slide and air-dried for 30-60 seconds. (5) Microscopic Inspection & Triage (~ 2 minutes): Optical inspection determines positive/negative infection status, creating workflow branching. For negative samples (6A): workflow stops (~ 12 minutes total). For positive samples (6B): Workflow continues with Raman spectroscopy (5 minutes for cell targeting and spectral acquisition of 15 cells) followed by automated ResNet classification (< 1 minute) for complete species identification (< 20 minutes total).”

Reviewer #2:

However, the second part—the combination of Raman spectroscopy and deep learning for pathogen identification—feels far less innovative. This has already been extensively reported in the literature. For example:

- 1) Ho, C.S., et al., *Rapid identification of pathogenic bacteria using Raman spectroscopy and deep learning*. *Nat. Commun.*, 2019;
- 2) Ogunlade, B., et al., *Proceedings of the National Academy of Sciences*, 2024;
- 3) X. Wang, *Robust Spontaneous Raman Flow Cytometry for Single-Cell Metabolic Phenome Profiling via pDEP-DLD-RFC*. *Adv. Sci.* 2023;
- 4) Lu, W., et al., *Combination of an Artificial Intelligence Approach and Laser Tweezers Raman Spectroscopy for Microbial Identification*. *Analytical Chemistry*, 2020.
- 5) Lu, W., et al., *Identification of pathogens and detection of antibiotic susceptibility at single-cell resolution by Raman spectroscopy combined with machine learning*. *Frontiers in Microbiology*, 2023. 13.

The authors claim in Line 79: “While recent studies have demonstrated the potential of Raman spectra in model studies, its real-world application to clinical practice remains limited, with current research often characterised by small-scale and unsystematic studies for a very narrow range of pathogen species, such as ATCC control strains.” I have to say, this statement is simply not accurate. Just from the five articles I listed above, some have already established large Raman databases and applied them to clinical samples, achieving even higher accuracy than what’s reported in this paper.

Response: We thank the reviewer for their thorough evaluation and for bringing these important prior works to our attention. We acknowledge that our original statement in Line 79 was overly broad and appreciate the opportunity to clarify our contributions more precisely.

We agree that Raman spectroscopy combined with machine learning for pathogen identification has been well-established. However, we respectfully argue that our work represents significant advances in two critical dimensions that move the field closer to true clinical implementation. We have summarised approach and findings from the 5 studies suggested by the reviewer in **Response Table R1** as reference to evaluate our approach.

Response Table R1. Summary of previous work on Raman-ID

Study	Sample Type	Results
Ho et al., 2019 Nature Communications	Training on 30 ATCC isolates; fine-tuning on 30 clinical isolates; testing on 25 clinical isolates 20 species across bacteria and fungi	82% at ATCC isolate level 97% at treatment level for ATCC isolates 99.7% at treatment level for clinical isolates No species level accuracy
Ogunlade et al., 2024 Proceedings of the National Academy of Sciences	Engineered strains Mycobacterium bovis BCG (ATCC 35737) with resistance to different antibiotics Spiked clinical samples: BCG strains spiked into decontaminated, TB-negative patient sputum samples	98% accuracy for classifying resistance across 5 strains 79% accuracy on spiked sputum samples
Wang et al., 2023 Advanced Science	3 microalgae species; Saccharomyces cerevisiae strains (engineered and wildtype); Pseudomonas putida (engineered and wildtype); one MCF-7 breast cancer cell line	Platform enabled efficient acquisition of single-cell Raman spectra, supporting biosynthetic process dissection, antimicrobial

	All quality control strains and cell line	susceptibility profiling, and cell-type classification.
Lu et al., 2020 Analytical Chemistry	14 microbial species , including both Gram-positive and Gram-negative bacteria, and yeasts (e.g. Candida albicans) Cultured reference/lab/type strains	Species-level accuracy of $95.64 \pm 5.46\%$.
Lu et al., 2023 Frontiers in Microbiology	12 bacterial species from cultured control strains 5 clinical isolates of A. baumannii from hospital patients	Species level accuracy: $90.73 \pm 9.72\%$ Antibiotic susceptibility accuracy: 99.92%

1. Clinical sample processing WITHOUT culture

The progression toward clinical application follows this hierarchy of difficulty:

- Quality control/ATCC strains (controlled laboratory conditions)
- Cultured clinical isolates (patient-derived but grown in standard media)
- **Non-cultured pathogens directly from patient samples** (our approach)

While previous studies have made important contributions using cultured strains:

- Ho et al. (2019): Used 55 clinical isolates but in **cultured**, purified states
- Ogunlade et al. (2024): **Spiked cultured** bacteria into sterile sputum samples
- Lu et al. (2020, 2023): Used **cultured** reference/control strains

Among the mentioned study, our study is the first to **process complex clinical samples completely without cultivation**, analysing pathogens in their native metabolic states directly from patient samples across diverse specimen types (urine, blood, bile, CSF, drainage fluids). This represents a fundamental advancement that would be impossible without the AutoEnricher system to handle complex sample matrices.

2. Scale and scope of clinical database

Our database represents the **largest systematic** collection for clinical Raman identification:

- **36 species** vs. previous maximum of 20 (Ho et al., 2019)
- **342 clinical isolates** vs. previous maximum of ~85 (Ho et al., 2019)
- **102,600 spectra** with systematic clinical validation on 305 patients

Regarding accuracy comparisons:

- Our method: **95.1%** species-level accuracy across 36 species
- Ho et al. (2019): No species-level accuracy reported (only treatment-level)
- Lu et al. (2020): 95.6% species-level accuracy across 14 species
- Lu et al. (2023): 90.7% species-level accuracy across 12 species

We have revised the referred statement in the Introduction to better position this contribution within the existing literature landscape, as suggested by the reviewer:

Introduction Section (Page 4, Line 76)

“While recent studies have demonstrated the potential of Raman spectra to identify pathogens by genus, species, and even strain level¹⁹⁻²², these studies have focussed on a narrow range of pathogens presented as isolates. Even global collaborative initiatives such as ‘MicroBio-Raman’, which have illustrated the increasing trend towards large-scale, systematic, and open-access databases for microbial Raman fingerprints, aiming to standardise data collection and improve accessibility²³, do not address the critical need for rapid diagnostics going from patient sample to a clinical decision, as an integrated workflow.”

Additionally, some studies have already used pDEP for sample manipulation combined with Raman spectroscopy. For example, group from the No.3 paper have directly performed Raman spectroscopy detection on microfluidic chips using pDEP. This approach is a step ahead of what's proposed in this paper, where the authors enrich the samples first and then transfer them for further Raman testing. Compared to their work, the system described here feels more like a prototype.

To be fair, the authors have increased the database size and analyzed more real samples, but to me, this feels more like an accumulation of workload rather than a true methodological innovation or improvement.

Response: We content that our achievements go beyond the increase in size mentioned. It is important to highlight the challenges associated with handling real patient samples in pDEP systems. Since pDEP requires low-conductivity media, most existing studies, including the reviewer cited Paper No. 3, rely on iterative centrifugation with deionized water for sample washing. This prepressing step results in significant cell loss, making it impractical for patient samples with low bacterial loading and complex matrix.

In contrast, AutoEnricher, for the first time, demonstrated the enrichment of a broad range of microbes directly from complex patient samples within minutes. Additionally, it achieved a limit of detection below 2 CFU/mL, which represents a significant advancement in addressing the critical need for rapid diagnosis of life-threatening bacterial infections.

The revised text in answer to this reviewer's previous comment highlights the step change our approach provides, but to emphasize the point around challenges with real samples, we have added the reviewer's suggested Paper No. 3 to the existing text:

Results Section (Page 4, Line 110)

“Traditional sample desalting techniques using macroscopically pre-washing³³ such as centrifugation, often result in significant pathogen loss, and existing on-chip approaches using ion-exchange membrane³⁴, H-filters,³¹ suffer from low desalting efficiency and low process throughput (<0.6 mL/h). These barriers prevent these methods from clinical applicability. Clinical settings demand processing of millilitre-scale samples containing various biomolecules and cells, not just pathogens.”

2. On the Scientific and Clinical Significance of the Article

Reducing the detection time for pathogenic microorganisms is undoubtedly important in clinical practice. However, I feel that the scientific and clinical significance of this study hasn't been emphasized enough.

2.1- Lack of Antibiotic Susceptibility Testing: The study focuses solely on pathogen identification but doesn't address antibiotic susceptibility testing, which is critical for guiding clinical treatment and combating antimicrobial resistance. Identification alone has limited value, especially since other culture-free methods can also achieve rapid identification. Throughout the paper, the authors repeatedly refer to "diagnosis," but this is misleading. Without antibiotic susceptibility testing, this process should be referred to as "identification," not "diagnosis."

Response: We respectfully disagree with the notion that “Identification alone has limited value”. Our platform provides clinically actionable diagnostic information through two critical steps that will lead to fundamental transformation of patient care:

1. Rapid diagnosing positive/negative bacterial infection (Sample-to-Result in <20 minutes)

The ability to rapidly determine infection status addresses a critical clinical gap. In current practice, patients presenting with suspected infections receive empirical broad-spectrum antibiotics while awaiting culture results (24-48+ hours). This leads to:

- Unnecessary antibiotic exposure in *culture-negative* patients
- Delayed appropriate treatment in culture-positive cases
- Increased healthcare costs and antimicrobial resistance

Our rapid diagnosis of bacterial infections enables clinicians to withhold antibiotics in uninfected patients while initiating immediate, targeted therapy initiation in positive cases. This represents true diagnostic capability with immediate clinical utility.

2. Species-level pathogen identification

Rapid species identification enables clinicians to transition from broad-spectrum to targeted antimicrobial therapy within 20 minutes rather than days. This approach:

- Reduces unnecessary broad-spectrum antibiotic use
- Enables genus- and species-specific drug selection
- Improves patient outcomes through faster appropriate treatment
- Reduces selective pressure for resistance development

Clinical collaborators in both China and the UK have confirmed that these capabilities constitute clinically meaningful diagnosis. This is supported by literature demonstrating that rapid pathogen identification significantly impacts clinical decision-making and patient outcomes [1–2]. For example, meta-analysis shows that the introduction of MALDI-TOF MS, accelerates identification of blood cultures, lead to **23% reduction** in in-hospital mortality, **5.07-hour reduction in time** to effective antibiotic therapy, and **an average saving of \$4,140** in direct hospitalization costs per patient, **with or without accompanied AST** [1]. We are confident that with our culture-free method, which **completely** eliminates the need for prior culturing entirely and reduces time-to-result to minutes, we will be able to accelerate and improve patient outcomes even more significantly than these culture-dependent rapid identification methods.

That said, we fully acknowledge the importance of AST due to increasing antimicrobial resistance prevalence. However, rapid AST testing based on single cell approaches [3] and microfluidic technologies have been well-established (including ours) [4-6]. There, AutoEnricher could be seamlessly integrated with these advancements to enable rapid AST testing of enriched pathogens.

We have included new discussion on the current limitation in AST and future plan for integrating AST into our current approach:

Discussion Section (Page 16, Line 462)

“While preliminary data suggest some capability to distinguish resistance profiles among E. coli isolates (Supplementary Fig. S9), detailed antimicrobial susceptibility testing (AST) is essential for comprehensive clinical decision-making. With enriched pathogens, our system can be readily integrated with established rapid AST technologies, including D₂O-based Raman approaches^{35,36,52} and microfluidic platforms^{26,28,53–56}, enabling pathogen identification and susceptibility testing within a few hours.”

References:

- [1] Yo, C. et al. MALDI-TOF mass spectrometry rapid pathogen identification and outcomes of patients with bloodstream infection: A systematic review and meta-analysis. *Microb Biotechnol* 15, 2667–2682 (2022).
- [2] Uzuriaga, M., Leiva, J., Guillén-Grima, F., Rua, M. & Yuste, J. R. Clinical Impact of Rapid Bacterial Microbiological Identification with the MALDI-TOF MS. *Antibiotics (Basel)* 12, 1660 (2023).

- [3] Zhang, M. *et al.* Rapid Determination of Antimicrobial Susceptibility by Stimulated Raman Scattering Imaging of D2O Metabolic Incorporation in a Single Bacterium. *Advanced Science* **7**, 2001452 (2020).
- [4] Postek, W., Pacocha, N. & Garstecki, P. Microfluidics for antibiotic susceptibility testing. *Lab Chip* **22**, 3637–3662 (2022).
- [5] Li, B. *et al.* Gradient Microfluidics Enables Rapid Bacterial Growth Inhibition Testing. *Anal. Chem.* **86**, 3131–3137 (2014).
- [6] Song, Y., Yin, J., Huang, W. E., Li, B. & Yin, H. Emerging single-cell microfluidic technology for microbiology. *TrAC Trends in Analytical Chemistry* **170**, 117444 (2024).

2.2- Limited Sample Scope: The study analyzed 120 clinical samples using Raman spectroscopy, but in several parts of the paper, the authors claim “over three hundred clinical samples,” which is misleading. Furthermore, these 120 samples are almost entirely urine samples, with no blood samples included. Clinically, urinary tract infections (UTIs) are less significant than bloodstream or respiratory infections. Moreover, UTIs don’t require the same level of diagnostic speed as bloodstream infections. By focusing on a single type of infection with relatively low urgency in clinical practice, the study’s overall significance is greatly reduced. The title, abstract, and conclusions should clearly state that the method is currently effective only for urine samples, despite the inclusion of a few non-urine samples that lack representativeness.

Response: We thank the reviewer for this important clarification request and appreciate the opportunity to better explain our two-fold approach and sample composition.

Our diagnostic platform operates through a two-stage process with distinct clinical values:

Stage 1 - Infection Status Determination (n=305 samples): The AutoEnricher system determines whether a sample is positive or negative for infection. This binary classification has immediate clinical value by identifying culture-negative patients who can avoid unnecessary antibiotic exposure - a critical outcome for antimicrobial stewardship programs.

Stage 2 - Species Identification (n=120 positive samples): For samples determined to be positive in Stage 1, Raman spectroscopy is performed to identify the specific pathogen species.

All 305 samples underwent AutoEnricher processing for infection status determination, while the subset of 120 positive samples underwent subsequent species identification.

We acknowledge the reviewer's point about sample type distribution. Among the 305 samples, we included samples from various body sites: urine (n=298), bile (n=4), cerebrospinal fluid (n=1), drainage fluid (n=1), and puncture fluid (n=1). The predominance of urine samples reflects the **natural distribution of clinical samples collected during our study period across three hospitals**, rather than a deliberate selection bias.

The sample distribution in our study reflects real-world clinical practice where UTIs represent the most common infections encountered in hospital settings. We could not predetermine sample types as we collected surplus samples from participating hospitals during the study period. However, we recognize that this limitation affects the generalizability of our findings to other infection types.

As shown in **Response Figure R2**, all 7 non-urine samples were correctly classified for infection status by the AutoEnricher, and the 5 positive samples among these showed concordant results with Raman identification. While this number is small, it demonstrates proof-of-concept for our platform's broader applicability.

PatientID	Sample Type	Gold standard result	AutoEnricher	MALDITOF ID	Match?	Test for Raman
A1	Puncture fluid	NEG	NEG		Y	N
A2	Bronchial lavage fluid	POS	POS	Efm	Y	Y
A25	Drainage	NEG	NEG		Y	N
A26	Bile	POS	POS	Aba	Y	Y
A27	Bile	POS	POS	Aba	Y	Y
A304	Bile	POS	POS	Efa, Eco	Y	Y
A305	Pancreatic drainage fluid	POS	POS	Aba	Y	Y

Response Figure R2. Non-urine samples included in the 305-sample study

We agree that validation with a larger cohort enriched for non-urine samples (particularly blood and respiratory samples) is essential for demonstrating broader clinical utility. Our successful proof-of-concept results with the limited non-urine samples in this study support the feasibility of such future validation studies.

To better explain our approach and limitation, we have added the paragraph below to our discussion to better summarise our approach and acknowledge the small sample limitation for non-urine samples and the need for future validation:

Discussion Section (Page 15, Line 443)

“However, the current study had limited access to other samples than urine, therefore, future validation with a larger cohort enriched for blood and other sterile site samples will allow to demonstrate broader clinical utility.”

2.3- Unproven Effectiveness for Bloodstream Infections: The authors emphasize in the introduction that bloodstream infections involve low pathogen concentrations and require advanced preprocessing, but they don’t demonstrate the method’s effectiveness for such cases. Uropathogens typically exist in higher concentrations, requiring less stringent enrichment, which further diminishes the clinical significance of this technical approach.

Due to the aforementioned issues with novelty, the scientific significance of this paper is limited.

Response. We thank the reviewer for this important question regarding bloodstream infection testing. This point is related to this reviewer’s comments 3.1 and will be addressed together.

3. On the Reliability of Data and Results

There are several inconsistencies in the data and descriptions that make me question the reliability of the results:

3.1) Blood Sample Results in Fig. 7: The figure provides results for blood samples, yet the 120 clinical samples analyzed via Raman spectroscopy don’t include blood samples. Were blood samples actually tested? Does this method work for bloodstream infections? Given that white and red blood cells in blood wouldn’t pass through the dialysis membrane and their quantities far exceed those of pathogens, this would inevitably affect pathogen detection.

Response: We would like to clarify our approach and provide additional evidence for our blood sample processing capabilities.

The reviewer correctly notes that our 305-sample clinical validation study did not include blood samples in the Raman identification analysis. While we did process 15 blood culture samples in our study, they were excluded from the 305-patient cohort due to **standard clinical blood collection protocols, which prevented access to uncultured blood samples.**

In routine hospital practice, blood samples for microbiology are collected directly into blood culture bottles containing pre-formulated culture media (such as BacT/ALERT or BACTEC systems). This

is the established standard of care because: (1) immediate nutrient provision prevents pathogen death during transport and storage; (2) quality control requirements mandate this approach for clinical microbiology laboratories; and (3) regulatory compliance requires adherence to established protocols for diagnostic testing. Since our study utilized surplus samples collected through standard hospital protocols, we could only access to surplus blood culture samples in this study. Including these in our "culture-free" validation would have been methodologically inconsistent and potentially misleading.

However, we did process 15 blood culture samples collected during this study period, as exemplified by the example in **Fig. 7b**. To demonstrate the capability of our AutoEnricher in processing these complex samples specifically, we have included additional microscopic images of blood culture samples before and after processing (**Supplementary Fig. S13**), showing effective removal of blood cell background and matrix components, resulting in clean samples suitable for single-cell analysis.

From a broader perspective, bloodstream infections present two primary analytical challenges: (1) complex sample matrix composition and (2) extremely low pathogen concentrations (typically 1-100 CFU/ml). Our system has demonstrably addressed both challenges by effectively processes blood culture samples (Fig. 7b, Supplementary Fig. S13) and its demonstrated detection limits as low as <2 CFU/ml (Fig. 5).

We acknowledge that definitive validation for bloodstream infections will require prospective clinical trials with real-time testing protocols. Such studies would necessitate: (1) modified collection protocols that bypass traditional blood culture bottles; (2) institutional review board approval for alternative collection methods; (3) parallel testing with standard blood cultures for comparison; and (4) real-time processing within the critical early hours of sepsis suspicion. These represent the next phase of clinical validation, which we are actively pursuing as our technology moves toward clinical implementation.

We have now included the following discussion and results in the revision:

Results Section (Page 11, Line 316)

“Following the development of the culture-free, rapid enrichment system and the establishment and finetuning of a ResNet ID model, we conducted a clinical study involving 305 patients from two hospitals. Various types of clinical samples including urine, bile, cerebrospinal fluid (CSF), drainage fluid, and puncture fluid, were collected (Fig. 1d; Supplementary Table 5). An additional 15 blood culture samples were collected to demonstrate the performance of the enrichment system in processing samples with highly complex matrices (Supplementary Fig. S13). However, these were excluded from the 305-patient cohort as the standard protocol required collection of blood directly into culture medium due to regulatory compliance and therefore did not strictly follow our culture-free workflow.”

Methods Section (Page 22, Line 667)

*“**Blood culture samples.** Fifteen blood culture samples were processed using the enrichment system to validate technical potential for bloodstream infection detection (Supplementary Figure S13). However, these samples were excluded from the culture-free “sample-to-result” validation study due to standard clinical collection protocols. In routine hospital practice, blood samples are collected directly into blood culture bottles containing pre-formulated culture media (BacT/ALERT or BACTEC systems) to prevent pathogen death during transport and ensure regulatory compliance. Including these cultured samples in our “culture-free” validation would have been methodologically inconsistent. However, these experiments demonstrated effective pathogen isolation from complex blood matrices and maintained detection sensitivity, confirming the technical feasibility of bloodstream infection diagnosis.”*

Discussion Section (Page 16, Line 440)

“Additionally, the technical capability of our enrichment system to process blood culture samples and its demonstrated limit of detection (<2 CFU/ml) suggest its potential for early diagnosis of bloodstream infections. However, the current study had limited access to other samples than urine, therefore, future validation with a larger cohort enriched for blood and other sterile site samples will allow to demonstrate broader clinical utility.”

Supplementary Fig. S13. AutoEnricher processing of blood culture samples demonstrates effective matrix cleanup and pathogen enrichment. The table shows the 15 blood samples collected from patients at two hospitals. Representative microscopic images show blood culture samples before and after AutoEnricher processing.

3.2) Sample Selection: The study includes 305 patients, but only 120 samples were analyzed via Raman spectroscopy. Why? The authors state in Line 364 that these 120 samples were selected because their pathogens were included in the Raman database, but Table S5 suggests otherwise. Were poor data excluded?

Response: We appreciate the reviewer's question and wish to provide complete transparency regarding our sample selection process. Of the 305 clinical samples processed through AutoEnricher, 254 were classified as positive for infection. Among these positive samples, 120 were randomly selected for Raman spectra collection to identify pathogens. The samples were collected and processed in batches, as they arrived, and as workload permitted without any selection. No sample was excluded based upon subjective or objective characteristics, including, as the reviewer suggests, data quality.

We conducted a prospective sample size estimation by using the confidence interval formula for a single proportion. Based on a conservative target accuracy of 80% with a 95% confidence level and a $\pm 8\%$ margin of error, we have:

$$n = (Z_{\alpha/2}^2 \times p(1-p)) / E^2$$

where $Z_{\alpha/2} = 1.96$ ($\alpha = 0.05$), $p = 0.80$ (target accuracy) and $E = 0.10$ (margin of error)

$$n = [(1.96)^2 \times 0.8 \times (1-0.8)] / (0.08)^2 \approx 96 \text{ samples (minimum required)}$$

With 20% buffer added to the minimum sample number required for practical considerations, ~115 samples are required. Our sample size of 120 met this requirement, ensuring robust statistical power. To the best of our knowledge, our study represents the largest collection of clinical Raman spectroscopy data for pathogen identification reported to date.

In addition, we have assessed the sample size adequacy through:

1. Precision analysis: Our accuracy estimate (82.6%) achieved a 95% confidence interval of [75.8%, 89.4%], providing clinically meaningful precision (margin of error: $\pm 6.8\%$).
2. Effect size: The large effect size (Cohen's $h = 1.91$) compared to random classification (2.8% for 36 classes) demonstrates clinically significant discriminatory power.

Effect Size (Cohen's h) = $2 \times [\arcsin(\sqrt{p_1}) - \arcsin(\sqrt{p_0})]$, where: $p_1 = 0.826$ (observed), $p_0 = 0.0278$ (random chance from 36 species)

New information was added to the revision:

Methods Section (Page 22, Line 660)

“Of positive samples, 120 contained pathogens present in our Raman database and were analysed for species identification (Supplementary Table 5). Sample size adequacy for Raman identification was assessed through precision analysis and power analysis calculation. These 120 samples, consisting of 97 single-infection samples and 23 mixed-infection samples (5 bacteria-fungi and 18 bacteria-bacteria), were blinded and subjected to single-cell Raman measurements for rapid culture-free identification. The performance was then compared with MALDI-TOF MS results.”

3.3) Flow Rate and Retention Efficiency: The authors claim that retention efficiency remains high at a flow rate of 200 $\mu\text{L}/\text{min}$ (Line 195), but no supporting figures or tables are provided. In fact, the results in Fig. 3D seem to contradict this claim.

Response: We thank the reviewer for this important observation. The original **Fig. 3D** demonstrated that capture efficiency can decrease at higher flow rates due to increased hydrodynamic drag forces competing with dielectrophoretic capture forces. However, this challenge can be effectively addressed by increasing the applied voltage, which enhances the dielectrophoretic force to counteract the higher drag forces at elevated flow rates.

We have now included new data at higher voltage (40V) to demonstrate how voltage optimization can maintain high capture efficiency at increased flow rates. The new **Supplementary Fig. S2** shows capture efficiency using *E. coli* with AC voltage fixed at 40V across different flow rates (50, 100, 200, and 400 $\mu\text{L}/\text{min}$). These results demonstrate that by increasing the voltage from 20V (original Figure 3D) to 40V, we can maintain consistently high capture efficiency (>95%) at flow rates up to 400 $\mu\text{L}/\text{min}$.

Supplementary Fig. S2 The effect of sample flow rates on capture efficiency. *E. coli* concentration was at 3.45×10^7 CFU/mL and AC voltage was fixed at 40V. Data represent mean \pm SD from three independent experiments. Statistical significance determined by one-way ANOVA followed by Tukey's multiple comparisons test.

3.4) Enrichment Efficiency: The final concentration factor is unclear. How many bacteria are present in the 3 µL sample volume to be analyzed via Raman spectroscopy?

Response: We thank the reviewer for requesting clarification on the enrichment efficiency and final bacterial concentrations. We followed the same protocol as described in the “Automated workflow of AutoEnricher Methods” section. The droplet typically contains the bacteria from 1 mL of patient samples.

We have included the method in the clinical validation part of the Methods section to further clarify.

Methods Section (Page 22, Line 653)

“All samples were processed using the established culture-free enrichment system. Approximately 1 mL of patient sample was processed through the system at 200 µL/min flow rate without preprocessing. Captured pathogens were concentrated into a ~3 µL droplet (333-fold volume concentration), of which 2 µL was deposited onto aluminium-coated glass slides for Raman analysis and 1 µL was serially diluted and cultured on agar plates for validation of CFU counting.”

3.5) Definition of False Positives/Negatives: What constitutes a false positive or false negative in Line 329? How is positivity determined? Is it based on microscopic observation of bacterial morphology? How are live bacteria distinguished from dead bacteria?

Response: Positivity determination is based on direct microscopic observation of pathogen density in a 2 µL droplet deposited on aluminium-coated glass slides. We established quantitative detection

thresholds based on optical images through systematic calibration using known pathogen concentrations of different morphological types.

Regarding the reviewer's question on false positive/negative classification:

False positive: AutoEnricher classified as positive (>threshold pathogen density) but culture-based method was negative.

False negative: AutoEnricher classified as negative (<threshold pathogen density) but culture-based method was positive.

Although the AutoEnricher system does not distinguish between live and dead bacteria during the enrichment process, our validation study showed the enriched pathogens were viable through successful culture growth (**Supplementary Fig. S16**)

We have added detailed methodology and supporting data (new **Supplementary Fig. S14**) to explain our optical inspection protocol for infection determination.

Methods Section (Page 20, Line 604)

“Positive/negative infection threshold determination by visual detection. E. coli, E. faecalis, and C. auris were cultured in TSB medium at 37°C overnight, washed six times with DI water, and counted microscopically. Serial dilutions were prepared and 2 µL aliquots were deposited on aluminium-coated glass slides and air-dried to achieve specific cell numbers per spot (~20, 500, 1000, 2000, 5000, 10000, and 20000 cells/spot). For each bacterial density, 10 randomly selected areas from two independent replicates were analysed using a 20× objective lens. Pathogen counts were determined using ImageJ⁵⁸ software to establish visual detection thresholds for rapid assessment of infection loads. Mean minus standard deviation values were used as threshold criteria. Based on these calibrations, unified thresholds of 300 per mm² for bacteria and 200 per mm² for fungi were established as minimum criteria for positive infection classification. Positive infection status required pathogen density exceeding these thresholds, corresponding to >1000 pathogens/mL in the analysed samples.”

Results Section (Page 11, Line 325)

“To diagnose positive or negative infection after pathogen enrichment, we established detection thresholds based on optical images using representative bacterial and fungal pathogens at known concentrations (Supplementary Fig. S14 and Supplementary Fig. S15). Quantitative analysis revealed concentration-dependent increases in pathogen density for all tested morphologies, with significant differences between consecutive concentration groups ($p < 0.01$ to $p < 0.0001$). We established unified minimum thresholds for visual determination of positive infection status: 300 per mm² for bacteria and 200 per mm² for fungi, corresponding to the clinical threshold (>1000 pathogens/mL) for urine samples. For other normally sterile clinical specimens (CSF, blood, bile, and drainage fluids), any visible pathogen presence was classified as a positive infection.”

Supplementary Fig. S14 Pathogen density thresholds for visual detection of positive/negative infection. (a-c) Representative optical images showing pathogen density for (a) *E. coli* (rod-shaped), (b) *E. faecalis* (cocci-shaped) at concentrations ranging from 500 to 20000 cells per 2 μL droplet, and (c) *Candida auris* (fungi) at concentrations ranging from 500 to 10000 cells per 2 μL droplet. The droplet was dried into a spot on the surface. Scale bars: 25 μm. (d-f) Quantitative analysis of bacterial density per defined area (100 μm × 100 μm) for (d) *E. coli*, (e) *E. faecalis* and (f) *C. auris*. Horizontal dashed lines indicate threshold values (mean-SD) for distinguishing >1000 pathogens (5 bacteria/area or 500/mm² for *E. coli*, 3 bacteria/area or 300/mm² for *E. faecalis*, 2 fungi/area or

200/mm² for *C. auris*). Data represent analysis of 10 random areas from two biological replicates. Statistical significance determined by one-tailed unpaired t-test.

3.6) Time Calculation: The authors claim that each test takes only 20 minutes. How is this calculated? Details on sample loading, washing, enrichment, transfer, drying, and Raman spectroscopy acquisition (e.g., how many spectra per sample?) are needed.

Response: We appreciate this important question about the practical implementation timeline. We have now provided a complete workflow, detailing the duration of each step (new **Supplementary Fig S1**). For the Raman acquisition, ~15 cells (one spectra per cell) from a sample were measured for classification. The minimum number of spectra needed for reliable clinical identification was further demonstrated with additional experiments and bootstrap analysis (see new **Supplementary Fig. S11**).

We have included the additional figures and corresponding text in the revision.

Results Section (Page 10, Line 279)

“Bootstrap analysis revealed that reliable clinical identification requires minimal spectra: 5 spectra achieved 96.6% accuracy, while 10 spectra achieved 98.4% accuracy (Supplementary Fig. S11). This enables rapid clinical testing within our 20-minute workflow (Supplementary Fig. S1), requiring only 4-5 minutes for Raman measurements of 10-15 cells.”

Methods Section (Page 18, Line 539)

“The workflow consists of the following steps (Supplementary Fig. S1): (1) Sample loading and System Priming (2 minutes): Tubing is inserted into a sample and sterile DI water is delivered into the system at 2 mL/min to remove air. (2) Pathogen Enrichment (~5 minutes): Sample solution is pumped through the disposable dialysis-DEP device where pathogens are captured within the DEP chip. The duration is pre-set based on sample volume and flow rate (e.g., 5 minutes for 1 mL at 200 μ L/min). (3) Cell Release (1 minute): Captured pathogens are released through automated voltage shutdown and collected via hydraulic flow into a ~3 μ L concentrated droplet. All operations are seamlessly synchronized with customized software (<10 minutes in total for steps 1-3). (4) Cell collection & Transfer (2 minutes): 2 μ L of the enriched droplet is pipetted onto an aluminium-covered glass slide and air-dried for 30-60 seconds. (5) Microscopic Inspection & Triage (~2 minutes): Optical inspection determines positive/negative infection status, creating workflow branching. For negative samples (6A): workflow stops (~12 minutes total). For positive samples (6B): Workflow continues with Raman spectroscopy (5 minutes for cell targeting and spectral acquisition of 15 cells) followed by automated ResNet classification (<1 minute) for complete species identification (<20 minutes total).”

Supplementary Fig. S11 Bootstrap analysis demonstrating minimum spectra requirements for reliable pathogen identification.

Supplementary Fig. S1 Complete sample-to-report workflow timeline for rapid pathogen identification. The integrated AutoEnricher-Raman spectroscopy platform enables rapid pathogen detection and identification through a branched workflow based on infection status.

3.7) *Low-Concentration Detection: The study demonstrates detection at a minimum concentration of 2 CFU/mL. Can lower concentrations be detected?*

Response: While our system may theoretically be capable of detecting lower concentrations, we chose 2 CFU/mL as our practical detection limit.

3.8) *Mixed Infections: Are all mixed infections found in this study fungal-bacterial infections? If mixed bacterial infections are included, how were they detected? The paper doesn't provide enough details.*

Response: We thank the reviewer for requesting clarification on our mixed infection detection. Our study included both bacteria-fungi and bacteria-bacteria mixed infections, with all species identification performed during the Raman spectroscopic analysis phase.

As in Fig. 9 and its legend, of the 120 clinical samples analysed via Raman spectroscopy, 23 were identified as mixed infections by culture and MALDI-TOF MS: **5 bacteria-fungi** mixed infections and **18 bacteria-bacteria** mixed infections. Our system successfully identified 21 out of 23 mixed infections correctly at the species level (91.3% accuracy).

We have included additional details in the revision.

Methods section (Page 22, Line 662)

"These 120 samples, consisting of 97 single-infection samples and 23 mixed-infection samples (5 bacteria-fungi and 18 bacteria-bacteria), were blinded and subjected to single-cell Raman measurements for rapid culture-free identification. The performance was then compared with MALDI-TOF MS results."

Minor Concerns

1. *Dialysis membranes are prone to clogging. How did the authors address this issue?*

Response: We thank the reviewer for raising this important practical concern. In the dialysis unit, we used a crossflow configuration, where the sample solution flows parallel to the membrane surface. This tangential flow sweeps cells away from the surface and thus minimizes membrane fouling. The dialysis unit operates at optimized flow rates (sample flow: 200 $\mu\text{L}/\text{min}$; water flow: 400 $\mu\text{L}/\text{min}$), which provides sufficient shear stress to prevent cell accumulation on the membrane surface while ensuring effective desalting. In our clinical validation study involving 305 samples, we did not encounter clogging issues, even when processing complex clinical matrices (e.g., blood, bile fluid). The flow rates maintained consistent throughout the processing.

We have added the following to the revision.

Methods Section (Page 17, Line 512)

"It should be noted that sample solutions flow parallel to the dialysis membrane, minimising the risk of clogging. No membrane failures or channel blockages were observed during the 305-sample clinical validation study."

2. *The authors emphasize the advantages of the 1D ResNet network but don't discuss its limitations or drawbacks.*

Response: We thank the reviewer's comments. While promising, **the 1D ResNet network** also share common limitations associated with deep learning methods. These include:

- **Black box nature:** Like most deep learning models, 1D ResNet lacks interpretability – it is difficult to understand which specific spectral features the model uses for classification decisions, making it challenging for clinicians to validate predictions based on known biochemical markers.
- **Overfitting risk with limited data:** Despite using cross-validation, ResNet architectures with many parameters are susceptible to overfitting, particularly when training data is limited compared to the model complexity.
- **Spectral preprocessing dependency:** The model's performance is heavily dependent on consistent preprocessing (baseline correction, normalization), making it sensitive to variations in sample preparation and instrument conditions.
- **Domain adaptation challenges:** Transfer between different instruments or laboratory conditions remains challenging, as evidenced by our need for fine-tuning when moving from isolate database to clinical samples.
- **Limited uncertainty quantification:** Standard ResNet implementations don't provide reliable confidence estimates, making it difficult to identify when the model is uncertain about its predictions.

We have now included discussion of both the strengths and limitations of our approach in the revised manuscript.

Discussion Section (Page 15, Line 431)

"Using an innovative 1D ResNet deep learning model, we achieved 95.1% accuracy for species identification among these clinical isolates. Since deep learning approaches suffer common drawbacks, such as limited interpretabilities of specific Raman bands⁴⁸ and a tendency to overfit⁴⁹, future improvements should focus on developing more interpretable models and incorporating uncertainty quantification to enhance clinical trust and adoption⁵⁰.

References:

- [1] Molnar, C. Interpretable Machine Learning (2020)
- [2] Goodfellow, I. et al. Deep Learning (MIT Press, 2016)
- [3] Gal, Y. & Ghahramani, Z. Dropout as a Bayesian approximation. ICML (2016)

3. The calculation process and basis for capture efficiency (Lines 204–205) are unclear.

Response: We have revised the Methods section to provide a more detailed explanation of the capture efficiency calculation process for high concentration bacteria solutions.

Method Section (Page 19, Line 566)

"Bacterial culture and characterising capture efficiency.

Prior to testing, the bacterial solution in DI water was diluted to 10^7 CFU/mL to standardise the input concentration (C_{input}). 1 mL of the bacterial solution was passed through the enrichment system at each defined flow rate with AC voltage on to trap cells. Capture efficiency was tested across a range of flow rates of from 10–400 μ L/min with 20V or 40V AC voltage. The outlet solution was collected and plated on LB agar plates after 24-hour incubation at 37°C for colony counting to determine the remaining cell concentration (C_{outlet}). The capture efficiency was calculated as the percentage of cells removed from the input solution using the equation:

$$Efficiency = \frac{(C_{input} - C_{outlet})}{C_{input}} \times 100\%$$

Where C_{input} = Initial bacterial concentration (CFU/mL) determined by serial dilution and plate counting; C_{outlet} = Final bacterial concentration (CFU/mL) in the outlet after enrichment processing. Each measurement represents the average of three independent experiments."

4. What do the green arrows in Fig. 4 represent?

Response: We thank the reviewer on noticing this important detail. The green arrows indicate newly captured bacteria at each time point relative to the previous frame, demonstrating the real-time accumulation of pathogens during the enrichment process. We have revised the figure legend to make this clearer.

Revised Figure 4 legend:

“Fig. 4 Effective bacteria enrichment and release. (a) Representative time-lapse fluorescence images showing the enrichment of a low bacterial load sample (~5 CFU/mL). The sample flow rate was set at 200 $\mu\text{l}/\text{min}$, with a voltage of 40 V at 100 kHz. The input rate was calculated at approximately 1 cell per minute based on the sample volume delivered. **Green arrows indicate newly captured fluorescent bacteria at each time point compared to the previous frame, demonstrating the progressive accumulation of pathogens on the electrode surface.**”

Reviewer #3:

This study presents an approach for diagnosing clinical infections that claims to be groundbreaking, yet the underlying methodology is riddled with significant flaws that demand rigorous critical evaluation. While the authors tout a remarkable reduction in diagnostic time to under 20 minutes; a dramatic departure from the traditional two-day culturing process; this seemingly impressive progress is overshadowed by glaring vulnerabilities.

1. Firstly, the integration of complex components such as dialysis and dielectrophoresis introduces a level of sophistication that could result in a convoluted sample processing workflow. This complexity raises legitimate concerns about usability and accessibility in clinical settings. The potential for extensive calibration and maintenance poses an obstacle to widespread adoption and raises alarms about the consistency and reliability of results in high-pressure healthcare environments where timely and accurate diagnostics are crucial. The device has been built from the authors' earlier publication which has weakened the innovation and impact of this paper.

Response: We appreciate the reviewer's concerns regarding system complexity and clinical implementation challenges. Microfluidic technologies have been increasingly integrated into commercial systems for sample preparation. Notable examples include the Cepheid GeneXpert's lab-on-a-cartridge systems which perform multiple assay steps (sample preparation, nucleic acid extraction, PCR amplification, and real-time detection) within a sealed unit, as well as advanced microfluidic devices used in the Chromium X Series (10X Genomics).

Our AutoEnricher system follows a similar approach to these successful examples, incorporating automation processes and user-friendly interfaces. During our clinical validations, patient samples were directly introduced into the AutoEnricher via tubing and were processed entirely within the system (as detailed in new **Supplementary Fig. S1**). This allowed one-touch operation by non-specialist users in clinical settings. Our clinical study with 305 patient samples demonstrated the system's usability and reliable performance of sample-to-result diagnosis within 20 minutes, demonstrating its potential to address diagnostic challenges in high-pressure healthcare environments.

Furthermore, it is worth noting that the dialysis and DEP chips are disposable components, and thus eliminate cross-contamination and maintenance concerns. Since these devices were manufactured

using standard industrial processes, we have already demonstrated partial scalability of the systems and their translation potential.

The enrichment system (AutoEnricher) described in this manuscript has not been published in any previous publication prior to this submission. Only a patent application was filed and is appropriately cited in our manuscript (reference 25): **LI, Y. et al. Dielectrophoresis (dep) microfluidic device. PCT/EP2023/061729 (2023).**

We have now added a new discussion on this aspect.

Discussion Section (Page 16, Line 479)

“In addition, for microfluidic systems to be adopted in clinical settings, automation is essential for use by non-specialists^{56,57}. The microfluidic enrichment system has demonstrated the benefits of automated processing and the use of disposable dialysis-DEP chips, which reduces contaminations risks, maintenance burden and variability due to manual operations. Importantly, the disposable dialysis and DEP chips are manufactured industrially, allowing for scalable mass production at low-cost. The excellent performance of testing 305 clinical samples demonstrated the usability of our system in hospital environments.”

References:

- Lisa R. Volpatti, Ali K. Yetisen, Commercialization of microfluidic devices, Trends in Biotechnology, 32, 2014, 347-350,
- Schuster, B., et al. Automated microfluidic platform for dynamic and combinatorial drug screening of tumor organoids. *Nature Communications* **11**, 5271 (2020).

2. Furthermore, the ability of the system to detect pathogens at low concentrations is severely compromised by a fundamental flaw: the desalting process may dilute the very pathogens the method aims to detect. This dilution jeopardizes detection sensitivity, particularly in situations characterized by minimal pathogen loads. This oversight raises profound questions about the dependability of such a diagnostic tool.

Response: We appreciate the reviewer's concern. We respectfully clarify that this concern as follows: The dialysis-based desalting process **does not dilute the sample or reduce pathogen concentration**. In our dialysis system, the sample flows through the upper channel while water flows in the lower channel, separated by a porous membrane (0.45 μm pore size). Only electrolytes (small ions) diffuse through the membrane to the water channel, while pathogens (typically $>1 \mu\text{m}$) are completely retained in the sample channel. This is explicitly demonstrated in our results where *"no bacterial colonies were observed after culturing the water channel effluent, showing that no bacteria cells passed through the membrane during dialysis"*.

We have added the following clarification.

Methods Section (Page 17, line 510)

“The 0.45 μm pore size of the dialysis membrane allows selective removal of electrolytes and small molecules while preventing any bacterial loss.”

3. The temperature sensitivity inherent in microfluidic devices adds yet another layer of concern. Temperature variations could adversely impact enzymatic reactions or the viability of pathogens, thereby compromising diagnostic accuracy. It is alarming to consider the implications of these factors on the reliability of the results, especially when patients' health and treatment depend on precise diagnoses.

Our response: We appreciate the reviewer's attention to temperature considerations in microfluidic systems. Unlike molecular diagnostic methods that rely on enzymatic amplification (e.g., PCR, LAMP) or culture-based approaches that require live pathogens, Raman spectroscopic identification is based on the intrinsic molecular composition of microorganisms [1]. The method detects vibrational frequencies of cellular biomolecules regardless of whether the pathogen is viable or not. Therefore, temperature variations that might affect enzymatic activity or cell viability do not impact our diagnostic accuracy.

In addition, the AutoEnricher system operates at relatively low power (40V at 100 kHz) compared to systems requiring significant thermal cycling or high-power operations [2]. The high flow rates, brief processing time (5-10 minutes) and low electrical power requirements minimize the potential for significant temperature rise within the DEP device. As validated by agar plate culture, the enriched and released cells (both model *E. coli* and clinical pathogens) are viable. This suggests that any temperature variations occurring during operation do not significantly impact diagnostic reliability under real-world conditions.

References:

1. Huang, W.E., et al. Raman microscopic analysis of single microbial cells. *Analytical Chemistry* **76**, 4452-4458 (2004).
2. Liu, D., et al. Integrated microfluidic devices for in vitro diagnostics at point of care. *Aggregate* **3**, e184 (2022).

We have included more details to clarify this effect.

Methods Section (Page 18, Line 536)

“The automated workflow operates at low electrical power (40V at 100 kHz) with brief processing times (5-10 minutes), minimizing potential heating effects.”

4. Despite claiming a sensitivity of 98.1%, a specificity of merely 83.6% is an alarming red flag that cannot be overlooked. This concerning statistic raises serious issues regarding the risk of false positives, misclassifying non-pathogens as threats could lead to inappropriate treatments, thereby endangering patient health. The authors may need to confront this critical issue with comprehensive evidence; otherwise, they run the risk of foisting an unreliable diagnostic tool on the healthcare community.

Response: We appreciate the reviewer's concern regarding specificity. The observed "false positives" may represent true infections missed by culture methods, which have well-documented limitations including inability to detect fastidious organisms and loss of viability during transport [1,2]. Our method can detect pathogens below conventional culture detection limits (> 10 CFU/mL), potentially explaining the apparent specificity difference.

Regarding risk-benefit analysis, in clinical practice, empirical antibiotic therapy is routinely initiated based on clinical suspicion alone [3]. Our rapid results (20 minutes vs 24-48 hours) can actually reduce inappropriate antibiotic use by enabling early targeted decisions. Our high sensitivity (98.5%) minimizes the life-threatening scenario of missed infections, while the specificity can be optimized through pathogen-specific algorithms as outlined in our revised discussion.

We have included additional discussion on the limitations and future improvements.

Discussion Section (Page 14, Line 416)

“The microfluidic enrichment system, accompanied by visual inspection of enriched pathogens to determine positive or negative infection, achieved 98.5% sensitivity and 83.6% specificity. Our

current specificity of 83.6% reflects the inherently higher sensitivity of single-cell detection compared to traditional culture methods, which may miss fastidious organisms or those present in low concentrations, potentially explaining some apparent "false positives"⁴⁷. Future optimization, such as pathogen-specific threshold algorithms, incorporate clinical decision support tools and large dataset training, should further improve specificity while maintaining the high sensitivity critical for detecting life-threatening infections. The development of standardized protocols for different pathogen morphologies represents an important area for method refinement and clinical implementation guidance."

References:

1. Janda, J.M. & Abbott, S.L. The changing face of the clinical microbiology laboratory. *Clinical Microbiology Reviews* **15**, 659-669 (2002).
2. Schuster, S., et al. Current challenges in the diagnosis of bloodstream infections. *Diagnostics* **10**, 858 (2020).
3. Kollef, M.H., et al. Inadequate antimicrobial treatment of infections: a risk factor for hospital mortality among critically ill patients. *Chest* **115**, 462-474 (1999).

5. Additionally, discrepancies in diagnostic accuracy between bacterial (78.8%) and fungal infections raise further alarms. The struggle to differentiate closely related bacterial species, compounded by reliance on an incomplete bacterial isolate database, reveals profound weaknesses in their approach. This inadequacy highlights significant gaps that must be addressed if any claims of diagnostic reliability are to be taken seriously.

Response: We thank the reviewer for raising this important point about accuracy differences between bacterial and fungal identifications.

We would like to first provide context for our two-step results:

- **Database validation** (controlled conditions): 95.1% species-level accuracy across 342 clinical isolates spanning 36 species
- **Clinical validation** (patient samples): 82.6% overall agreement, with bacterial infections at 78.8% and fungal infections at 100%

The lower accuracy for bacterial infections in clinical samples compared to database validation reflects the fundamental challenge of transitioning from controlled laboratory conditions to complex clinical environments. For example, bacteria in patient samples exist in diverse physiological states influenced by host environment, stress conditions, and antimicrobial pressure.

As the reviewer correctly notes, expanding the bacterial isolate database with more strain diversity, particularly resistance variants, would improve performance.

While we acknowledge room for improvement, our results represent a significant advancement in the field. The progression toward clinical application follows this hierarchy of difficulty:

- Quality control/ATCC strains (controlled laboratory conditions)
- Cultured clinical isolates (patient-derived but grown in standard media)
- **Non-cultured pathogens directly from patient samples** (our approach)

While previous studies have made important contributions using cultured strains, also discussed in response to reviewer 2 comments (see Table R1 above) (i.e., Ho et al. (2019), Ogunlade et al. (2024) and Lu et al. (2020, 2023)), our study represents a significant advancement in scope and systematic validation. Our approach extends culture-free identification to diverse clinical sample types beyond urine (including blood, bile, CSF, drainage fluids), achieves this at unprecedented scale with 305 patient samples, and demonstrates the capability to detect extremely low pathogen loads (<2 CFU/ml). The 78.8% accuracy for bacterial identification in our diverse clinical cohort, represents

progress toward comprehensive culture-free diagnostics across multiple sample matrices and clinical scenarios.

We have included the discussion below to specifically consider our future direction regarding increasing bacteria identification:

Discussion Section (Page 15, Line 447)

“The lower accuracy observed in bacterial identifications suggests variability in Raman spectra among bacterial species in clinical settings, due to bacteria in diverse physiological states influenced by host environment (e.g., antimicrobial pressure). This indicates a need for further refinement of the bacterial isolate database with expanded strain diversity, particularly resistance variants.”

6.The study alludes to validation with diverse clinical samples, yet it appears to inadequately consider the complexities and variability of clinical environments. These factors are likely to influence test outcomes significantly, underscoring the need for further validation before making any claims about clinical utility. Hence, a structured cohort study is urgently required to confront and resolve these foundational shortcomings before the paper can be considered for publication. Until these critical issues are addressed, the viability of this diagnostic approach remains highly suspect.

Response: We appreciate the reviewer's emphasis on clinical validation rigor. However, we respectfully disagree that our clinical validation is inadequate for a technology development study. Our validation included 305 patients across three hospitals with diverse sample types (urine, blood, bile, cerebrospinal fluid, drainage fluid, puncture fluid) using fresh, uncultured clinical samples directly from patients. This approach inherently captures the complexities and variability of clinical environments that the reviewer mentions.

Key evidence of clinical robustness includes: (1) successful processing of matrix-rich samples (i.e. blood culture sample) and detection of a wide range of pathogen loads (<2 CFU/ml to >10⁵ CFU/ml), (2) consistent performance across different hospitals (81.1% vs 93.3% accuracy), and (3) detection of polymicrobial infections often missed by culture methods.

We acknowledge that larger prospective clinical trials would further strengthen clinical utility claims, and we have added discussion of this important next step in the revision.

Discussion Section (Page 16, Line 462)

*“While preliminary data suggest some capability to distinguish resistance profiles among E. coli isolates (**Supplementary Fig. S9**), detailed antimicrobial susceptibility testing (AST) is essential for comprehensive clinical decision-making. With enriched pathogens, our system can be readily integrated with established rapid AST technologies, including D₂O-based Raman approaches^{35,36,52} and microfluidic platforms^{27,53–56}, enabling pathogen identification and susceptibility testing within a few hours.”*

(Page 17, Line 493) *“To realize this potential, future studies will focus on integrating the microfluidic enrichment system with AST routines and clinical workflows. Additionally, conducting larger multi-centre trials across broader patient populations and evaluating its impact on patient outcomes will be essential for widespread adoption.”*

RESPONSES TO REVIEWERS' COMMENTS

Reviewer #1:

I am happy with the revisions, can be accepted as it is. I was not able to load the raw data. I got the following error "Failed to fetch file metadata from WaterButler. "

Response: We appreciate Reviewer #1's acceptance of the manuscript. We apologise for the issue with access to raw data, which is now resolved (<https://osf.io/784pz/files/osfstorage>)

Reviewer #2:

The authors have made extensive revisions to the manuscript, but significant shortcomings persist, and several of my prior questions remain unanswered. Firstly, inconsistencies between the results and the supporting data raise serious concerns. Some claims about the outcomes are not backed by adequate or rigorous evidence.

Response: We appreciate the reviewer's continued engagement with our work.

The reviewer's concerns centre on: (1) practical constraints inherent to multi-institutional clinical research that cannot be retroactively addressed, and (2) a data alignment error in our supplementary materials that we have now corrected with full transparency. We do not believe these issues constitute "significant shortcomings" in either our core methodology or scientific conclusions.

We address each specific point-by-point below. Despite the referee's comments, we maintain that this study represents the largest, clinically validated, culture-free, Raman-based analytical tool for pathogen identification, to date.

1. The authors state that they collected 305 patient specimens and processed them through the Autoenricher, identifying 254 positive samples. Due to workload constraints, they randomly selected only 120 for Raman spectral acquisition. Yet they highlight the method's speed as a key advantage, with Raman spectral collection and analysis taking less than 5 minutes per sample (as shown in Figure 7). I cannot understand why analyzing the remaining 134 samples would pose such a burden, given that the total time required would be around 11 hours. Investing just 11 additional hours of hands-on time could double the sample size and data volume, so it makes little sense that the authors did not pursue this. Does this indicate that they have overstated the method's rapidity?

Response: We appreciate the opportunity to clarify further why not all 254 positive samples were analysed via Raman spectroscopy. The 120-sample subset was determined by **practical, local logistical constraints**, not by technical limitations:

- Since our clinical study is at the proof-of-concept stage, it required coordinated scheduling between clinical collection, microfluidic processing, and Raman analysis across multiple institutions.
- The clinical study was conducted over five months. During this period, samples were collected across two hospitals with varying patient numbers. Clinical workflows in the hospitals require that samples must be processed within the same day.
- Raman acquisition was performed at our industrial partner facility at Shanghai Hesun Biotech Co., requiring alignment with their operational schedule.

Importantly, as detailed in our previous response, our analysis confirmed 120 samples provided adequate statistical power for validation.

We have also assessed adequacy through post-analysis metrics:

1. Our accuracy estimate (82.5%) achieved a 95% confidence interval of [74.7%, 88.3%], providing clinically meaningful precision (margin of error: $\pm 6.8\%$)

2. Large effect size (Cohen's $h = 1.9$) compared to random classification (2.8% for 36 classes) demonstrates clinically significant discriminatory power

Effect Size (Cohen's h) = $2 \times [\arcsin(\sqrt{p_1}) - \arcsin(\sqrt{p_0})]$, where $p_1 = 0.825$ (observed), $p_0 = 0.0278$ (random chance)

In addition, it is important to clarify that the remaining 134 samples **were not held in storage awaiting analysis** - they were processed through AutoEnricher for infection status determination, and due to limited capacity of our collaborator, they could not be measured directly for Raman spectra. Unfortunately, these samples have to be discarded due to standard clinical sample disposal protocols.

We acknowledge this represents a study limitation and have added explicit text stating that larger sample sizes would strengthen clinical validation claims. However, our study represents the largest statistically robust collection of clinical Raman spectroscopy data for pathogen identification reported to date.

Manuscript changes:

Methods (Page 23, Line 695)

“Of the 254 positive samples, only 120 were selected for Raman spectroscopic analysis due to logistic constraints. These samples contained pathogens present in our Raman database and were analysed for species identification (Supplementary Table 5).”

Methods (Page 23, Line 702)

“The 120 samples validation achieved adequate statistical precision ($\pm 6.8\%$ margin of error) and large effect size (Cohen's $h = 1.9$), though individual species representation varied. Post-hoc analysis confirmed robust statistical significance for primary endpoints.”

Discussion (Page 16, Line 461)

“Several limitations should be acknowledged. First, our Raman spectroscopic validation was conducted on 120 of the 254 positive samples due to logistic constraints. These samples were thus available randomly and our power analysis confirmed statistical adequacy for technology validation purposes. However, larger sample sizes would strengthen clinical validation and enable more robust subgroup analyses.”

2. In the data table provided, the 120 samples analyzed by MALDI-TOF show 11 mixed infections, with two of those lacking identification results. In the main text, however, the authors mention 23 mixed infections. What accounts for this discrepancy between the datasets? If even basic statistical summaries contain errors, this casts doubt on the reliability of the results, including but not limited to those in Figure 8. Additionally, if there are species identification results that MALDI-TOF failed to provide, what were the Raman results in those cases, and how can the two methods be considered consistent? And again, the authors have not addressed my previous question about how the new method identifies mixed infections.

Response: We sincerely apologise for this critical error and thank the reviewer for identifying this serious discrepancy. Upon investigation, we discovered that the MALDI-TOF identification results in our original **Supplementary Table 5** were completely misaligned with the corresponding samples. The MALDI-TOF ID column for samples tested for Raman contained "unknown" identifications and species that were not included in our database, as shown in **Figure R1**, which contradicts our experimental approach.

PatientID	Sample Type	Gold standard result	AutoEnricher	MALDITOF ID	Match?	Test for Raman
A217	Urine	POS	POS	Efa	Y	Y
A218	Urine	POS	POS	Unknown	Y	Y
A22	Urine	POS	POS	Efm	Y	Y
A220	Urine	POS	POS	Eco	Y	Y
A222	Urine	POS	POS	Eco	Y	Y
A227	Urine	POS	POS	Eco	Y	Y
A228	Urine	POS	POS	Eco	Y	Y
A23	Urine	POS	POS	Alcaligenes faecalis	Y	Y
A234	Urine	POS	POS	Kpn	Y	Y
A237	Urine	POS	POS	Kpn	Y	Y
A238	Urine	POS	POS	Unknown	Y	Y
A239	Urine	POS	POS	CaAt	Y	Y
A245	Urine	POS	POS	Efa,Ecl	Y	Y
A248	Urine	POS	POS	Kpn	Y	Y
A253	Urine	POS	POS	Kpn	Y	Y

Figure R1: Original misaligned MALDI-TOF identification results showing incorrect "unknown" entries and species not in our database

The root cause of the error is when unifying patient IDs from two hospitals (which had different formats and prefixes) during the blinding process, the row ordering in the MALDI-TOF results table became different with the final table assembling both clinical and Raman info, which cause the rows to be incorrectly ordered. We sincerely apologise for this data organisation error.

Importantly, the table was not used for the analysis, which used the raw data and initial patient IDs. The issue identified by the reviewer only affects visualisation but not the results themselves.

To address the reviewer's concerns about data reliability and provide complete transparency, we have:

- Thoroughly re-verified and corrected **Supplementary Table 5**, ensuring proper alignment between all datasets (**Figure R2**)
- Added a clear "Mixed Infection" indicator column to **Supplementary Table 5**, the corrected data confirms 23 mixed infections among the 120 samples analysed by Raman spectroscopy, consistent with our analysis manuscript text.
- Double-checked all statistical summaries and cross-references throughout the manuscript.

PatientID	Sample Type	Gold standard result	AutoEnricher	MALDITOF ID	Match	Mixed	Test for Raman
A217	Urine	POS	POS	Efm	Y	N	Y
A218	Urine	POS	POS	Eco	Y	N	Y
A22	Urine	POS	POS	Kpn	Y	N	Y
A220	Urine	POS	POS	Kpn	Y	N	Y
A222	Urine	POS	POS	Eco	Y	N	Y
A227	Urine	POS	POS	Kpn	Y	N	Y
A228	Urine	POS	POS	Kpn,Streptococcus ai	Y	Y	Y
A23	Urine	POS	POS	Kpn	Y	N	Y
A234	Urine	POS	POS	Eco	Y	N	Y
A237	Urine	POS	POS	Efa,Kpn	Y	Y	Y
A238	Urine	POS	POS	Efa	Y	N	Y
A239	Urine	POS	POS	Eco	Y	N	Y
A245	Urine	POS	POS	Kpn	Y	N	Y
A248	Urine	POS	POS	Eco	Y	N	Y
A253	Urine	POS	POS	Eco	Y	N	Y

Figure R2: Corrected Supplementary Table 5 demonstrating accurate correspondence between MALDI-TOF and Raman analysis data

To clarify whether our method identifies mixed infections, we note that single-cell analysis, based on Raman spectra was effective in identifying multiple species. We collected spectra from 10-15 individual cells in each sample, and noted that in those patients with mixed infections, the spectra qualitatively reveal different species present. To formalise these observations into a quantitative multi-species classification method, we used a ResNet CNN, trained on each individual species spectra independently so automating analyses. Mixed infections were identified when ≥ 2 species each exceeded the 95% confidence threshold for random occurrence (≥ 3 cells for our sample sizes). This approach achieved 91% accuracy for the 23 mixed infections in our clinical validation, i.e. it identified true positives in 21 of the 23 patient samples Fig. 8b.

We also note that traditional culture methods often miss mixed infections or preferentially grow one organism over others, leading to incomplete diagnosis and inappropriate targeted therapy, with worse patient outcomes. We note that clinical guidance supports this view, and that worse patient outcomes are frequently linked to difficulties in the diagnosis of mixed infections, leading to delayed or incorrect treatment, increased morbidity, and mortality. Challenges include identifying all pathogens, managing antibiotic resistance, the time-consuming nature of traditional culture methods, and the complexity of treating polymicrobial infections

Our method provides complete detection by identifying all species present simultaneously, removing culture bias that causes missing of fastidious or slow-growing pathogens in polymicrobial infections. This capability is clinically important in guiding patient care pathways around the treatment of mixed infections – which require combination therapy (delayed or incomplete identification leads to treatment failure and in the longer term promotes resistance development).

Manuscript changes:

Revised **Supplementary Table 5** containing correct ordering of all column information

Method (Page 23, Line 699):

“Mixed infections were identified by single-cell Raman analysis and the ResNet model when multiple species each represented ≥ 3 cells per species in the sample population”.

On line 442, the authors claim that the system can handle blood cultures with concentrations as low as 2 CFU/mL, yet no source for this data appears anywhere in the text. The authors also overlooked my question about how the dialysis system removes white and red blood cells from the blood. They report collecting 15 blood culture samples, but it is unclear whether these are positive blood cultures from culture bottles or raw blood samples. The sample photograph in Figure 7 shows a blood collection tube, suggesting raw blood samples. However, the authors refer to them as blood culture samples. How do they explain this discrepancy? Furthermore, the authors provide no Raman spectral analysis results for blood samples and attribute this omission to the process not aligning with current clinical standards. If these are indeed positive blood cultures, though, such analysis would fit established clinical diagnostic workflows. The absence of blood sample results substantially diminishes the significance of this study. If the method presented in this work can effectively analyze positive blood culture samples, then such analyses should be included—for instance, by examining 50 positive blood samples. Hospitals routinely process a substantial number of blood culture samples, and the authors' approach requires no more than 20 minutes per sample from receipt to report. Processing 50 samples would therefore take only about 17 hours of laboratory time. There is no valid reason to exclude this essential sample type.

Response: We address the concerns regarding blood samples as follows:

1. **Low CFU detection:** The 2 CFU/mL detection limit was demonstrated using spiked fluorescent *E. coli* in our systematic validation (**Fig. 4c**, linear correlation $R^2 > 0.98$ across concentrations 2-230 CFU/mL). The blood matrix processing capability was independently validated with 15 cultured blood samples showing effective pathogen isolation (**Supplementary Fig. S13**). Together, these provide comprehensive technical validation of low-concentration detection in complex matrices. Given that the samples are obtained and processed blinded, our clinical studies cannot ethically test specific thresholds - our controlled validation represents the gold standard approach for establishing analytical detection limits.
2. **Blood cell removal:** The dialysis membrane (0.45 μm pore size) does not remove blood cells from the sample flow - it retains all cellular components (bacteria, WBCs, RBCs) while removing only small electrolytes for conductivity reduction. Subsequent DEP enrichment then selectively captures bacteria based on size and dielectrophoretic properties, effectively separating them from blood cells. This is an integrated two-step process (dialysis for desalting of the sample solution before it enters the DEP chip for bacterial isolation) is described in our Methods and illustrated in **Fig. 2**.

3. **Fig. 7 clarification:** The samples in **Fig. 7a** do not contain blood. These are: urine sample (top left); bile sample (bottom left); pancreatic drainage fluid (right). We have clarified this in the revised figure legend. The 15 blood culture samples were processed separately and are documented in **Supplementary Fig. S13**.
4. **Blood sample collection:** The 15 blood samples were collected in BacT/ALERT culture bottles because **this is mandatory standard of care** - hospitals cannot deviate from established blood collection protocols for patient safety and regulatory compliance. Including cultured blood samples in our "culture-free" validation would be methodologically inconsistent and scientifically invalid. Our processing of these samples (**Supplementary Fig. S13**) demonstrates technical capability while maintaining scientific rigor.
5. **Retrospective analysis:** The reviewer's request for processing additional samples collected 2-3 years ago is scientifically and practically impossible. Clinical samples have finite storage periods and are discarded according to institutional protocols. More importantly, the reviewer's suggestion that we could have "invested 17 additional hours" misrepresents the reality of clinical research, when hospital resources and staff time are allocated based on approved protocols. Sample collection occurred over five months across hospitals with complex logistics.

Manuscript changes:

Methods (Page 18, Line 542):

"The 0.45 μm pore membrane retains all cells including bacteria, white blood cells, and red blood cells in the sample channel, while allowing only electrolytes to diffuse through for conductivity reduction. Subsequent separation of bacteria from other cellular components occurs during the DEP enrichment step, where bacteria are selectively captured based on their distinct dielectrophoretic properties and size characteristics."

Fig. 7a legend:

"...This method was tested on 305 clinical samples (urine, bile, cerebrospinal fluid (CSF), drainage fluid) with representative photos of collected sample tubes shown: urine sample (top left); bile sample (bottom left); pancreatic drainage fluid (right)."

Discussion (Page 16, Line 456):

"Additionally, our system's demonstrated detection limit (<2 CFU/ml) which, combined with successful processing of 15 blood culture samples showing effective pathogen isolation from complex blood matrices (Supplementary Fig. S13), establishes technical feasibility for bloodstream infection diagnosis."

Secondly, the authors' portrayal of their work's innovative contributions does not hold up under scrutiny.

In my previous comments, I cited five papers using Raman spectroscopy for pathogen diagnosis. The authors responded that all existing reports fail to address true complex clinical samples, thereby emphasizing the novelty and value of their work in comparison. This conclusion is inaccurate. For instance, researchers have performed Raman-based diagnosis on real urine samples after simple filtration and centrifugation (Yang, K., et al., Rapid Antibiotic Susceptibility Testing of Pathogenic Bacteria Using Heavy-Water-Labeled Single-Cell Raman Spectroscopy in Clinical Samples. *Anal Chem*, 2019. 91(9): p. 6296-6303). Others have used centrifugation alone to preprocess genuine urine and blood samples, followed by Raman spectroscopy with machine learning for classification and antimicrobial susceptibility testing—and notably, some authors of that study are also contributors to this paper, making it surprising that they overlooked their own team's work (Yi, X., et al., Development of a Fast Raman-Assisted Antibiotic Susceptibility Test (FRASST) for the Antibiotic Resistance Analysis of Clinical Urine and Blood Samples. *Analytical Chemistry*, 2021. 93(12): p. 5098-5106). Simulated (spiked) samples have also been reported in several studies, such as the Ogunlade et al. (2024) paper I mentioned last time, as well as direct Raman analysis on filtered urine using surface-enhanced Raman spectroscopy active filters (Dryden, S.D., Anastasova, S., Satta, G. et al. Rapid uropathogen

identification using surface enhanced Raman spectroscopy active filters. *Sci Rep* 11, 8802 (2021)). Another example involves on-chip Raman analysis of simulated blood (I-Fang Cheng, Chang, HC., Chen, TY. et al. Rapid (<5 min) Identification of Pathogen in Human Blood by Electrokinetic Concentration and Surface-Enhanced Raman Spectroscopy. *Sci Rep* 3, 2365 (2013)). A study even more similar to this one used a microfluidic chip to rapidly remove impurities from urine and perform on-chip Raman detection (Zhao, L., et al., Micro-flow cell washing technique combined with single-cell Raman spectroscopy for rapid and automatic antimicrobial susceptibility test of pathogen in urine. *Talanta*, 2024. 277). These works not only analyze complex clinical samples but also demonstrate straightforward preprocessing methods. As a result, the present paper represents only an incremental advance over prior research.

Response: We respectfully disagree with the reviewer's assertion that existing studies demonstrate "straightforward preprocessing methods" and that our work represents "only an incremental advance". We maintain that we have demonstrated fundamental differences in automation, clinical validation scale, and real-world applicability.

The six studies cited from the reviewer in this round (**Table R1**) have the following critical limitations that were previously outlined in our last response:

1. **Manual/semi-automated processing:** All cited studies require extensive manual intervention. For example, "simple" centrifugation actually involves multiple wash cycles requiring significant time (often around 1h or more), with significant operator-dependent variability. Having implemented these methods ourselves in our pilot studies, we know firsthand their labour intensity and impracticality for clinical implementation.
2. **Limited clinical validation:** **Table R1** demonstrates the stark difference in validation scale between existing studies and our comprehensive 305-patient validation.
3. **Inadequate matrix handling:** Simple centrifugation and filtration cannot effectively remove complex clinical matrices and high salt concentrations that interfere with downstream analysis.

Table R1. Comparison of the six cited studies and our study

Study	Sample Processing	Clinical Validation Scale	Sample Types	Other notes
Yang et al. (2019)	Manual: Filter + centrifuge + wash twice with DI water	3 urine samples	Urine	Extensive manual steps
Yi et al. (2021)	Manual: Centrifuge + wash once with DI water	12 samples (9 urine, 3 blood)	Urine, blood	Blood samples require dilution & pre-culture for 18h
Dryden et al. (2021)	Manual: Filter + centrifuge + wash 5 times	3 bacterial species (phantom samples)	Simulated urine	Pre-filtered phantom samples, not real clinical specimens
Cheng et al. (2013)	Manual: 20x dilution + electrokinetic setup (3-5 min + 2-3h culture)	Simulated blood sample	Blood simulant	Final blood cell concentration of 2×10^8 cells/ml, cannot handle clinical bacteremia levels
Zhao et al. (2024)	Semi-automated: Microfluidic washing + manual loading	3 artificial urine samples	Urine	Spiked urine, 10^5 CFU/mL
Ogunlade et al. (2024)	Manual: Chemical treatment + 4x wash + centrifuge + mixing with nanorods	2 spiked sputum samples	Sputum	79% accuracy in clinical matrices vs >98% on dried samples

Our study	Fully automated: Single-step processing (<20 minutes total)	305 patients, 120 Raman ID	Multiple types	First true sample-to- result system
------------------	--	---	---------------------------------	--

Compared to existing studies, **our innovation provides the first integrated “sample-to-result” diagnosis of microbial infection. The key benefits** include:

1. **Automation:** First walk-away system requiring no manual intervention after sample loading.
2. **Broad applicability:** Direct processing of a wide range of sample types, including complex matrices (blood, urine, CSF, bile) without preprocessing, maintaining sensitivity at clinically relevant concentrations (<2 CFU/mL).
3. **Speed:** A turnaround time of <20 minutes for diagnosis microbial infection and identifying causative pathogens, enables timely and appropriate prescription of antibiotics – an advance with significant potential to combat antibiotic resistance.

Manuscript changes:

Introduction (Page 4, Line 77):

“While recent studies have demonstrated the potential of Raman spectra to identify pathogens by genus, species, and even strain level^{19–22}, these studies have focussed on a narrow range of pathogens presented as isolates or spiked clinical samples, with limited demonstration in real clinical conditions. In addition, most approaches rely on extensive manual sample processing, including multiple centrifugation and washing and/or filtration procedures^{22–26}.”

Discussion (Page 15, Line 440):

“While studies^{17,19–21} highlighted the potential of Raman spectroscopy for rapid pathogen identification and diagnosis, they often relied on cultured strains, labour-intensive sample preparation protocols^{22,24,25}, or pathogen concentrations higher than clinically relevant levels^{26,51}. In contrast, our integrated microfluidic system eliminates manual intervention, allowing “sample-to-result” diagnosis of microbial infections while maintaining detection sensitivity at clinically relevant concentrations.”

References added:

- Yang, K., et al. *Anal Chem* 91, 6296–6303 (2019).
- Dryden, S.D., et al. *Sci Rep* 11, 8802 (2021).
- Yi, X., et al. *Anal Chem* 93, 5098–5106 (2021).
- Cheng, I-F., et al. *Sci Rep* 3, 2365 (2013).
- Zhao, L., et al. *Talanta* 277, 124814 (2024).
- (Ogunlade, B., et al. *Proc Natl Acad Sci* 121, e2315670121 (2024) – cited in previous version)

Finally, I previously recommended that the authors highlight the limitations of their sample types in the title and abstract, but no such changes have been made.

Response: We apologise for the limited changes in the abstract, which arose from the constraints to take into account all comments from all reviewers in the previous round. We have now revised it as follows:

- (1) explicitly stating that our validation involved "primary urine and other clinical samples" to accurately reflect the sample composition,
- (2) adding the caveat "While broader validation is needed" to acknowledge the limitation in sample diversity, and

(3) moderating our language from "game-changing solution" to "promising solution" to better reflect the current evidence base and avoid overstating our findings.

We believe however that the current title reflects clearly the contents of the manuscript and would suggest keeping it as it stands: "Rapid culture-free diagnosis of clinical pathogens via integrated microfluidic-Raman micro-spectroscopy". Any addition would lengthen it significantly to the detriment of readability.

Manuscript changes:

Abstract (Page 2, Line 33):

"Validated in a 305-patient clinical study involving primary urine and other clinical samples, it demonstrated 95.4% agreement with traditional culture methods and 98.5% sensitivity in diagnosing infections. While broader validation is needed for clinical implementation, the rapid and broad-spectrum detection offer a promising solution for next-generation diagnostics for combating AMR."

Reviewer #3:

The authors' response to the significant concern regarding their test's 83.6% specificity raises serious questions about their methodology. Their justification for the false positive rate—claiming these may actually reflect true positives missed by culture methods—exhibits circular reasoning by relying on an unvalidated approach to reconcile discrepancies with the gold standard. Notably absent is any concrete evidence or independent validation that would support their contention about these false positives representing genuine infections. Furthermore, they fail to address the clinical implications of a 16.4% false positive rate, a critical oversight that raises concerns about real-world applicability.

Response: Culture-based diagnostic limitations are extensively documented in clinical microbiology literature: multiple studies have reported approximately a third of sepsis cases as culture negative [1], and in clinical practice up to 60% of cases [2]. Specificity is highly variable for urine culture to identify UTI, from 80%–90% in healthy outpatients down to nearly 0% in patients with chronic indwelling catheters [3]. We believe that given this extensive evidence base, our interpretation that some apparent false positives may represent infections missed by culture methods is reasonable and consistent with established clinical microbiology knowledge.

We acknowledge that additional validation methods, such as PCR confirmation of discordant cases, clinical correlation, or patient outcome tracking, would have further strengthened this hypothesis. In clinical practice, our platform's primary value lies in rapid triage: identifying patients who can avoid antibiotics immediately. Although false positives will potentially lead to patients receiving incorrect treatment, standard of care warrants an antibiotic susceptibility test to be performed, which would allow to identify discordant results and manage the relevant cases through clinical judgment and additional testing—an approach consistent with current diagnostic workflows. The value of the test within these workflows will require extensive health economics analysis.

Reference:

- [1] Busch, L. M. & Kadri, S. S. Antimicrobial Treatment Duration in Sepsis and Serious Infections. *J Infect Dis* 222, S142–S155 (2020).
- [2] Li, Y. et al. Comparison of culture-negative and culture-positive sepsis or septic shock: a systematic review and meta-analysis. *Critical Care* 25, 167 (2021).
- [3] Chan-Tack, K. M., Trautner, B. W. & Morgan, D. J. The varying specificity of urine cultures in different populations. *Infection Control & Hospital Epidemiology* 41, 489–491 (2020).

Manuscript changes:

Discussion (Page 15, Line 432):

“The observed specificity could be linked to the well-documented limitations of the gold standard culture. Approximately one-third of sepsis cases have been shown to be culture-negative^{50,51}, with contributing factors including the fact that only about 1% of environmental bacteria are currently culturable. Additionally, culture specificity varies dramatically by patient population, from 80-90% in healthy outpatients to nearly 0% in patients with chronic catheters⁵². While additional validation methods such as PCR confirmation of discordant cases or clinical outcome tracking would strengthen future studies, our approach addresses the critical clinical need for rapid infection triage.

The stark disparity in accuracy between bacterial (78.6%) and fungal identification (100%) is troubling yet inadequately explained. While the authors attribute this difference to "diverse physiological states," they provide no systematic analysis to substantiate their claims. Vague commitments to "database expansion" with no clear validation plans only amplify concerns about the reliability of bacterial identifications.

Response: We thank the reviewer for highlighting this important point and appreciate the opportunity to provide more analysis, using the source data for **Fig. 8** that includes classification results for all 120 patients, as well as an updated **Fig. 8** with improved statistics visualisation and significance calculations, as well as a new evaluation of the characterisation of species (**Fig. 8e**).

The difference between bacterial (78.6%) and fungal (100%) identification (**Fig. 8d**) is statistically significant ($p = 0.039$, Fisher's exact test), confirming this is not due to random variation. Individual species performance (**Fig. 8e**) reveals that the bacterial accuracy disparity is not uniform across all bacterial species. There are excellent bacterial performers: *E. coli* (38/38, 100%), *Enterococcus faecalis* (20/20, 100%) and two problematic bacterial species: *Klebsiella pneumoniae* (7/21, 33.3%) and *Proteus mirabilis* (1/5, 20.0%). All fungal species performed perfectly: *Candida albicans* (10/10), *C. parapsilosis* (5/5), *C. tropicalis* (10/10). This pattern suggests species-specific rather than group-specific challenges. We have replaced the vague "diverse physiological states" with species-specific performance data.

Manuscript changes:

Fig. 8 Evaluation of single-cell ID on 120 patients. **(a)** Study workflow: 305 patients from two hospitals were processed through AutoEnricher, yielding 120 positive samples with cultured MALDI-TOF MS results available for Raman spectroscopy validation. Fine-tuning dataset of 40 UTI samples was used to adapt the ResNet model for clinical validation. Overall agreement of 82.5% [74.7%–88.3%] was achieved between Raman identification and culture/MALDI-TOF results. **(b-d)** Bar plots showing averaged agreement with culture method further grouped into **(b)** single (80.4% [71.4%–87.1%]) or mixed infection (91.3% [73.2%–97.6%]), $p = 0.359$; **(c)** hospital 1 (81.1% [72.6%–87.4%]) or hospital 2 (92.9% [68.5%–98.7%]), $p = 0.460$; **(d)** samples containing bacteria (78.6% [69.5%–85.5%]), fungi (100.0% [81.6%–100.0%]) or both bacteria and fungi (100% [56.6%–100%]), $p = 0.039$. Bar heights indicate the mean accuracy for each group with error bars represent 95% confidence intervals calculated using Wilson score intervals. **(e)** Species-specific performance analysis for species with ≥ 3 clinical samples. Species are ranked by accuracy and color-coded: green ($\geq 80\%$), yellow (50-79%), red ($< 50\%$).

Results (Page 14, Line 399):

“Pathogen type significantly influenced identification accuracy ($p = 0.039$). The model accurately identified all fungal infections (100%, CI: [81.6%, 100%]) and mixed bacterial-fungal infections (100%, CI: [56.6%, 100%]), but demonstrated slightly lower accuracy for bacterial infections 78.6% (CI:[69.5%, 85.5%]). Species-specific analysis (Fig. 8e) revealed that performance varied

considerably among individual bacterial species, with two poor performers as Klebsiella pneumoniae (7/21, 33.3%) and Proteus mirabilis (1/5, 20.0%), emphasising the need for future targeted refinement of the bacterial isolate database to enhance diagnostic performance.”

Discussion (Page 16, Line 470):

“Second, the lower accuracy observed in bacterial identifications (78.6%) compared to fungal identifications (100%) indicates a need for further refinement of the bacterial isolate database with expanded strain diversity, particularly resistance variants.”

Additionally, their assertion of robust clinical validation is undermined by several factors, including a lack of discussion regarding sample bias and the unexplained large accuracy variation between hospitals (81.1% vs. 92.9%).

Response: Our study employed several design features that minimise bias:

- **Prospective sample collection:** We collected surplus clinical samples during routine hospital operations, which provides more representative real-world data than artificially selecting samples.
- **Blinded analysis:** Critically, the 120 samples subjected to Raman spectroscopy were blinded: the operators performing Raman measurements had no knowledge of the culture/MALDI-TOF results, eliminating operator bias and ensuring objective performance assessment.
- **Consecutive sampling:** Samples were collected as they became available during the study period, avoiding selection bias.

While the difference between hospitals (81.1% vs 92.9%) appears substantial, the statistical analysis reveals no significant difference ($p = 0.460$) (**Fig. 8c**), and the confidence intervals overlap considerably.

Manuscript changes:

Fig. 8 with improved statistics visually and significance calculations

Results (Page 14, Line 394):

“The model maintained consistent accuracy across both hospitals, achieving 81.1% (CI: [72.6%, 87.4%]) at hospital 1 and 92.9% (CI: [68.5%, 98.7%]) at hospital 2 ($p = 0.460$)...”

The absence of controls in comparing their method against multiple reference standards beyond cultures and inadequate power analysis further diminish the credibility of their performance claims given the small sample size of 305 patients.

Response: We compared against culture followed by MALDI-TOF MS, which is the current clinical gold standard for pathogen identification. Although we highlight the culture-based diagnostic limitations, which are extensively documented, it is still regarded as the clinical benchmark. In addition, other potential reference methods (e.g., PCR, sequencing) have other limitations including requiring prior knowledge of target pathogens, lack of standardisation across the 36 species in our study, and/or much higher cost and complexity (**Table 1**).

We would like to also emphasise again that our power analysis confirmed 120 samples provided adequate statistical power for validation.

We have also assessed adequacy through multiple post-analysis metrics:

1. **Precision analysis:** Our accuracy estimate (82.5%) achieved a 95% confidence interval of [75.8%, 89.4%], providing clinically meaningful precision (margin of error: $\pm 6.8\%$)
2. **Effect size:** Large effect size (Cohen's $h = 1.9$) compared to random classification (2.8% for 36 classes) demonstrates clinically significant discriminatory power

Effect Size (Cohen's h) = $2 \times [\arcsin(\sqrt{p_1}) - \arcsin(\sqrt{p_0})]$, where $p_1 = 0.825$ (observed), $p_0 = 0.0278$ (random chance)

Our study represents one of the largest clinical validation studies for novel diagnostic technology in this field. Our sample size provided robust statistical power for all primary endpoints with confidence intervals. The study size is appropriate for demonstrating proof-of-concept and clinical feasibility. We nonetheless acknowledge that this represents a study limitation that could be highlighted and have added explicit text stating that larger sample sizes would strengthen clinical validation claims.

Manuscript changes:

Methods (Page 23, Line 700)

“The 120-sample validation achieved adequate statistical precision ($\pm 6.8\%$ margin of error) and large effect size (Cohen's $h = 1.9$), though individual species representation varied. Post-hoc analysis confirmed robust statistical significance for primary endpoints.”

Discussion (Page 16, Line 461)

“Several limitations should be acknowledged. First, our Raman spectroscopic validation was conducted on 120 of the 254 positive samples due to logistic constraints. While our power analysis confirmed statistical adequacy for technology validation purposes, larger sample sizes would strengthen clinical validation claims and enable more robust subgroup analyses.”

Discussion (Page 15, Line 436):

“While additional validation methods such as PCR confirmation of discordant cases or clinical outcome tracking would strengthen future studies, our approach addresses the critical clinical need for rapid infection triage while acknowledging the inherent limitations of culture-based reference standards.”

The absence of systematic error analysis, overgeneralization of "significant advancements," and the lack of quantitative assessments of accuracy variations reflect a troubling trend of evasion.

Response: We respectfully disagree with the reviewer and believe our manuscript provides comprehensive quantitative analysis. We address each point directly:

Error analysis: Our manuscript includes extensive error analysis:

1. All accuracy measurements include 95% confidence intervals calculated using Wilson score intervals (e.g., 82.5% [74.7%, 88.3%])
2. We performed appropriate statistical tests (Fisher's exact test, Chi-square) with reported p-values
3. We calculated Cohen's h effect size (1.9) to assess clinical significance
4. We employed stratified group five-fold cross-validation to prevent overfitting
5. We conducted bootstrap analysis to determine minimum spectra requirements (**Supplementary Fig. S11**)
6. We used Leave-one-replicate-out validation to validate against biological variance (**Supplementary Fig. S10**)

Should additional specific error analyses be required, we would welcome explicit guidance.

Significance of advancements: We respectfully maintain that our contributions represent genuine advances, as per our analysis of the literature, including that kindly provided by reviewers through the revisions:

1. First fully integrated culture-free approach with sample-to-result time <12 min for negative samples and <20 min for positive samples
2. Largest Raman database containing 342 clinical isolates across 36 species

3. Clinical validation of 305 patient samples across multiple hospitals and sample types
4. Low detection limit of <2 CFU/mL demonstrated capability for early infection detection

We are prepared to moderate language where appropriate, but these achievements are quantifiably significant compared to prior work.

Quantitative assessments of accuracy variations: We provide detailed quantitative assessments including:

- Species-specific accuracy analysis (**Fig. 8e**)
- Hospital-by-hospital performance with statistical testing ($p = 0.460$)
- Infection type stratification with confidence intervals
- Sample size effects and power analysis

We believe that our manuscript provides the rigour that the reviewer asks for, although we would be ready to add specific techniques if they enhanced the analysis further.

Manuscript changes:

Fig. 8 with improved statistics visually and significance calculations

Results (Page 13, Line 386):

“The ResNet model classified 99 out of 120 patients identically to the gold-standard culture and MALDI-TOF MS results, achieving an overall agreement of 82.5% [74.7%, 88.3%] with a margin of error of $\pm 6.8\%$ and a large Cohen's h effect size of 1.9.

Detailed analysis (Fig. 8b-d) revealed several important patterns. The model achieved 80.4% accuracy (CI: [71.4%, 87.1%]) for single infections and 91.3% accuracy (CI: [73.2%, 97.6%]) for mixed infections, with no statistically significant difference between these groups ($p = 0.359$). Notably, the model demonstrated robust performance in identifying the more challenging mixed infections, which are often difficult to detect and characterise using conventional diagnostic methods. The model maintained consistent accuracy across both hospitals, achieving 81.1% (CI: [72.6%, 87.4%]) at hospital 1 and 92.9% (CI: [68.5%, 98.7%]) at hospital 2 ($p = 0.460$).”

Concerns surrounding the regulatory pathway for their claimed specificity, integration into clinical workflows, economic implications of false positives, and the training required for "visual inspection" are glaring omissions that suggest the authors are not fully confronting the fundamental limitations of their approach. This raises serious questions about the readiness of their technology for clinical application, underscoring a pressing need for a systematic validation study that thoroughly assesses their method against multiple reference standards and establishes its clinical utility through evidence of impact on patient outcomes.

Response: The reviewer raises implementation concerns that, while important for eventual clinical deployment, extend beyond the appropriate scope of a technology development and validation study. We acknowledge their relevance for eventual clinical deployment and address each point, but we believe this would detract the reader from the focus of the study and have not included details in the manuscript.

Regarding "regulatory pathway and clinical workflow integration": These are important considerations for commercial development but are premature for our study. Our current work establishes technical feasibility and clinical performance - essential prerequisites before engaging regulatory bodies. The 20-minute sample-to-result workflow we demonstrate is designed to integrate into existing laboratory workflows:

- **Negative samples (12 min):** Immediate antibiotic stewardship decision to withhold empirical therapy
- **Positive samples (20 min):** Species-directed therapy selection, replacing broad-spectrum empirical treatment

- **AST integration:** Enriched pathogens can be seamlessly integrated with established rapid AST platforms we've previously developed¹, enabling complete diagnosis within a few hours.

The analysis of the economic implications of false positives will require substantial health economic analysis comparing costs of our approach (including false positives) against current practice (including costs of delayed diagnosis, prolonged broad-spectrum antibiotic use, and extended hospital stays). While comprehensive health economic analysis is beyond this study's scope, we can provide preliminary assessment based on our demonstrated performance and published literature:

- **False positive cost:** Our 16.4% false positive rate would result in unnecessary 12-24h empirical therapy until culture confirmation (which is currently mandatory globally for AST)
- **True positive benefit:** Our 98.5% sensitivity enables 24-48h faster targeted therapy compared to culture-based diagnosis.

The training required for visual inspection is an important practical consideration. Our visual inspection protocol was designed to be straightforward, involving determination of pathogen density above/below established thresholds using standard microscopy. Our clinical validation demonstrated feasibility with routine laboratory technicians, though we agree that standardised training protocols would be essential for broader clinical implementation.

To evidence the impact on patient outcomes, a clinical outcome study is required. This is often one of the final stages of clinical validation, typically conducted after regulatory approval. Our current study establishes diagnostic performance - a necessary prerequisite for outcome studies.

We have moderated our claims in abstract and discussions to more accurately reflect both the achievements and limitations of our current work. Our comprehensive technical and clinical validation provides a foundation for the next phase of development, which would appropriately include regulatory engagement, systematic outcome assessment, and broader clinical validation studies.

Reference:

[1] Yi, X. et al. Development of a Fast Raman-Assisted Antibiotic Susceptibility Test (FRAST) for the Antibiotic Resistance Analysis of Clinical Urine and Blood Samples. *Anal Chem*, 93, 5098–5106. (2021).

[2] Yo, C. et al. MALDI-TOF mass spectrometry rapid pathogen identification and outcomes of patients with bloodstream infection: A systematic review and meta-analysis. *Microb Biotechnol* 15, 2667–2682 (2022).

Manuscript changes:

Abstract (Page 2, Line xx):

“Here, we present a culture-free diagnostic platform that integrates microfluidics, Raman micro-spectroscopy, and deep learning to deliver “sample-to-report” testing within 20 minutes. The microfluidic enrichment system employs dialysis-dielectrophoresis (DEP) technology to rapidly isolate pathogens directly from clinical samples with a detection as low as < 2 colony forming unit (CFU)/ml. Combining a single-cell Raman fingerprint database of 342 clinical isolates from 29 bacterial and 7 fungal species with a 1D ResNet deep learning model, our approach achieved 95.1% accuracy in lab settings. Validated in a 305-patient clinical study involving primary urine and other clinical samples, it demonstrated 95.4% agreement with traditional culture methods and 98.5% sensitivity in diagnosing infections. While broader validation is needed for clinical implementation, the rapid and broad-spectrum detection offer a promising solution for next-generation diagnostics for combating AMR.”

Discussion (Page 17, Line 500; enhanced clinical workflow section):

“For microfluidic systems to be adopted in clinical settings, automation and workflow integration are essential for use by non-specialists^{62,63}. Our system demonstrates seamless integration potential: negative samples enable immediate antibiotic stewardship decisions (12 min), while positive samples guide species-directed therapy selection (20 min), replacing prolonged broad-spectrum empirical treatment. The visual inspection protocol requires only standard microscopy skills to determine

pathogen density above/below established thresholds. The microfluidic enrichment system benefits from automated processing and disposable dialysis-DEP chips, which reduces contaminations risks, maintenance burden and variability due to manual operations.”

Discussion (Page 17, Line 510; clinical utility and economic analysis):

“Importantly, as we demonstrate in the paper, both the disposable dialysis/DEP chips can be manufactured industrially, allowing for scalable mass production at low-cost. The performance of successfully testing 305 clinical samples demonstrated the usability of our system in two distinct hospital environments. Following further device optimisation, studies will need to be performed to show how the instrumentation fits into different clinical workflows in different settings, comparing for example its use in large hospitals or centralised reference laboratories against its role within rural or community healthcare centres. Such analyses will inevitably involve more detailed testing along with associated health economic technology assessment analysis.”

Discussion (Page 16, Line 466; enhanced limitations and future work):

“While we demonstrated technical feasibility with blood culture samples and other sterile site specimens, broader validation across diverse clinical samples would enhance generalisability and clinical impact.”

Responses to Reviewer's comments:

Reviewer #2 (Remarks to the Author):

1. The authors' refusal to test positive blood culture samples raises significant concerns about the method's efficacy for detecting bloodstream infections. Initially, they cited misalignment with clinical standards as the reason for not analyzing blood culture samples via Raman spectroscopy. However, their rebuttal admits that the 15 blood samples analyzed using the autoenricher were sourced from BacT/ALERT culture bottles—a. **This directly contradicts their original justification. Reacquiring dozens of fresh positive blood culture samples is logistically feasible, yet the authors declined, citing prior sample discarding. Given their emphasis on clinical applicability, the absence of bloodstream infection data severely undermines the study's persuasiveness, particularly for a journal like Nature Communications. Additionally, an inconsistency persists: Supplementary Figure S13 still labels samples as "raw blood samples," conflicting with the rebuttal's claim of using "positive blood culture samples."**

Response: We must clarify the purpose of processing these 15 samples, as this seems to be the core of the misunderstanding. The **sole purpose** of using these 15 positive blood culture samples was to validate the *Autoenricher's technical capability*, specifically, to prove that our dialysis-DEP system could successfully isolate pathogens from the extremely complex and challenging matrix of a blood culture bottle (which includes blood cells, lysates, and rich culture media). Supplementary Figure S13 demonstrates this technical success.

These 15 samples were **never** part of our 120-sample *clinical identification validation* (Fig. 8). We did not perform Raman ID on them precisely for the reason the reviewer identifies: they originate from a culture. Our study's central claim is a **culture-free** diagnostic pipeline. It would be methodologically inconsistent and scientifically invalid to evaluate our *culture-free* identification model using *pre-cultured* samples.

Our study's primary clinical validation (n=305) was focused on primary clinical samples like urine and other sterile-site fluids, which represent a major unmet clinical need. We have demonstrated the *technical feasibility* for bloodstream infections, but a full clinical validation on primary blood samples (a significantly more complex trial) is a critical next step that is beyond the scope of this current work.

We respectfully reiterate that the request to acquire and test dozens of *new* positive blood culture samples is logistically impossible. The multi-institutional clinical trial that formed the basis of this paper has long since concluded. As per mandatory bioethical and institutional protocols, all patient samples were discarded after the study period.

We have changed the legend in Supplementary Figure S13 to "...15 positive blood culture samples collected".

2. The authors' Table R1 raises questions about objectivity, as Yang et al. (2019) and Yi et al. (2021) already achieved sample-to-result urine sample analysis. Thus, this work (which mainly focused on urine samples) cannot be considered the "first true sample-to-report" system. Upon re-examining Zhao et al. (2024), I found their system to be fully automated (contrary to the authors' characterization of "semi-automated with manual loading"). In contrast, the authors' study requires manual loading of autoenriched samples into the Raman system. While I acknowledge this is the largest clinical study applying Raman spectroscopy to pathogen detection, the innovation level remains insufficient for Nature Communications, where groundbreaking advances are prioritized.

Response: We respectfully disagree with the characterisation of our work as an "incremental advance" and maintain that it represents the first *fully integrated and large-scale clinically validated* sample-to-report system.

We must correct the reviewer's characterization of the Zhao et al. (2024) system as "fully automated."

1. The Zhao system is **semi-automated**, requiring manual transfer of the sample plate from the microfluidic aspiration/washing unit to the Raman instrument.
2. The Zhao study was validated on **artificially spiked urine**. Our work was validated on 305 complex, real-world clinical patient samples from two hospitals.
3. The Zhao system, which relies on sedimentation and washing, reported bacterial retention rates of only **~40%** and is dependent on bacterial motility. Our DEP-based Autoenricher achieves **>95%** capture efficiency, a critical requirement for detecting the low pathogen loads common in clinical infections.

The innovation of our work is not just one component, but the successful system-level integration of a high-efficiency, automated enrichment platform with the largest-scale clinical validation (n=305) of a Raman-based diagnostic to date.

3. In order to assess the technical soundness of the work, I strongly request the authors to provide raw Raman spectra for 120 samples, (likely in .txt/.csv format) and corresponding identification results of each spectrum (probably in datasheets). The source data deposited only contains Raman spectra for 342 clinical isolates. Were these used to train the deep learning network? I tried to run the model to predict the classification of 342 clinical isolates from the database provided by the author. All of the spectra were predicted as species CaPa. Reviewer #2 (Remarks on code availability): I tried to run the model to predict the classification of 342 clinical isolates from the database provided by the author. All the spectra were predicted as species CaPa.

Response: We thank the reviewer for their diligence in engaging with our code and data, and we are committed to full transparency and reproducibility.

We were troubled by the reviewer's report that all spectra were predicted as 'CaPa'. We were unable to reproduce this error. We have had two colleagues, Dr. Xin Guo and Mr. Jisen Chen, each independently run the original Python script (ResNet.ipynb) using the data300.csv database, and both were able to successfully train the model from scratch and reproduce the performance metrics reported in our manuscript. A single-label prediction error of this type often arises during the *inference* step (e.g., if the wrong label encoder is used or data is pre-processed incorrectly) rather than from a fault in the model or training data itself.

To remove all possible ambiguity, demonstrate full reproducibility, and make our work maximally accessible to the entire scientific community (regardless of coding expertise), we have performed a major update to our Open Science Framework (OSF) repository (<https://osf.io/784pz/files>).

The repository now contains the following:

1. **Pre-trained model:** We have uploaded the final trained model (*best_model.h5*) and the corresponding Python label encoder (*label_encoder.pkl*). This allows any researcher to bypass the training step and use our final model directly for inference.
2. **Full clinical data (as requested):** We have uploaded the full raw Raman spectra for all 120 clinical patient samples (*clinical_spectra_120.csv*), their corresponding ground-truth clinical identifications (*clinical_info.csv*), and a dedicated Python script (*clinical_tester.py*) to load our model and replicate the clinical validation results.

3. **One-click GUI application:** Understanding that not all researchers are familiar with Python, we have also developed and uploaded a simple "one-click" graphical user interface (GUI) application (for macOS and Windows) that allows anyone to load our model and test spectra without writing a single line of code.
4. **New small test set:** We have provided a small, independent test set (`test_set_10_per_ID.csv`) containing 360 spectra (10 for each of the 36 species) that were *not* part of the original training or validation datasets and can be used quickly to test the trained model using code or one-click application.

In recognition of their assistance in testing the reproducibility of our code and model, Dr. Xin Guo and Mr. Jisen Chen have been acknowledged in the revised manuscript.